# Global oceanic diazotroph database version 2 and elevated estimate of global oceanic N₂ fixation

Zhibo Shao[1,#], Yangchun Xu[1,#], Hua Wang[1], Weicheng Luo[1], Lice Wang[1], Yuhong Huang[1], Nona Sheila R. Agawin[2], Ayaz Ahmed[3], Mar Benavides[4,5], Mikkel Bentzon-Tilia[6], Ilana Berman-Frank[7], Hugo Berthelot[8], Isabelle C. Biegala[4], Mariana B. Bif[9], Antonio Bode[10], Sophie Bonnet[4], Deborah A. Bronk[11], Mark V. Brown[12], Lisa Campbell[13], Douglas G. Capone[14], Edward J. Carpenter[15], Nicolas Cassar[16,17], Bonnie X. Chang[18], Dreux Chappell[19], Yuh-ling Lee Chen[20], Matthew J. Church[21], Francisco M. Cornejo-Castillo[22], Amália Maria Sacilotto Detoni[23], Scott C. Doney[24], Cecile Dupouy[4], Marta Estrada[22], Camila Fernandez[25,26], B. Fernández-Castro[27], Debany Fonseca-Batista[28], Rachel A. Foster[29], Ken Furuya[30], Nicole Garcia[4], Kanji Goto[31], Jesús Gago[32], Mary Rose Gradoville[33], M. Robert Hamersley[34], Britt A. Henke[35], Cora Hörstmann[4], Amal Jayakumar[36], Zhibing Jiang[37], Shuh-Ji Kao[1], Dave Karl[38], Leila R. Kittu[39], Angela N. Knapp[40], Sanjeev Kumar[41], Julie LaRoche[42], Hongbin Liu[43], Jiaxing Liu[44], Caroline Lory[45], Carolin R. Löscher[46], Emilio Marañón[47], Lauren F. Messer[48], Matthew M. Mills[49], Wiebke Mohr[50], Pia H. Moisander[51], Claire Mahaffey[52], Robert Moore[53], Beatriz Mouriño-Carballido[47], Margaret R. Mulholland[54], Shin-ichiro Nakaoka[55], Joseph A. Needoba[56], Eric J. Raes[57], Eyal Rahav[58], Teodoro Ramírez-Cárdenas[59], Christian Furbo Reeder[4], Lasse Riemann[60], Virginie Riou[61], Julie C. Robidart[62], V.V.S.S. Sarma[63], Takuya Sato[64], Himanshu Saxena[41], Corday Selden[65], Justin R. Seymour[66], Dalin Shi[1], Takuhei Shiozaki[67], Arvind Singh[41], Rachel E. Sipler[12], Jun Sun[68,69], Koji Suzuki[70], Kazutaka Takahashi[71], Yehui Tan[44], Weiyi Tang[36], Jean-Éric Tremblay[72], Kendra Turk-Kubo[35], Zuozhu Wen[1], Angelicque E. White[38], Samuel T. Wilson[73], Takashi Yoshida[65], Jonathan P. Zehr[35], Run Zhang[1], Yao Zhang[1], Ya-Wei Luo[1]

[1] State Key Laboratory of Marine Environmental Science and College of Ocean and Earth Sciences, Xiamen University, Xiamen, Fujian, China

[2] Marine Ecology and Systematics (MarES) research group, University of the Balearic Islands, Palma de Mallorca, Spain

[3] Environment and life Science Research Centre, Kuwait Institute for Scientific Research, Salmiya, Kuwait

[4] Aix Marseille Univ, Université de Toulon, CNRS, IRD, MIO, UM 110, 13288, Marseille, France

[5] Turing Center for Living Systems, Aix-Marseille University, 13009 Marseille, France

[6] Department for Biotechnology and Biomedicine, Technical University of Denmark, Lyngby, Denmark

[7] Department of Marine Biology, Leon H. Charney School of Marine Sciences, University of Haifa, Haifa, Israel

[8] Ifremer, DYNECO, Plouzané, France

[9] Monterey Bay Aquarium Research Institute, Moss Landing, California, USA

[10] Oceanographic Center of A Coruña, Spanish Institute of Oceanography (IEO-CSIC), A Coruña, Spain

[11] Bigelow Laboratory for Ocean Sciences, East Boothbay, Maine, USA

[12] Climate Change Cluster, University of Technology Sydney, Sydney NSW, Australia

[13] Department of Oceanography, Texas A&M University, College Station, Texas, USA

[14] Department of Biological Sciences, Marine and Environmental Biology Section, University of Southern California, Los Angeles, California, USA

[15] College of Science and Engineering, San Francisco State University, San Francisco, California, USA

[16] Division of Earth and Ocean Sciences, Nicholas School of the Environment, Duke University, Durham, North Carolina, USA

[17] CNRS, Université de Brest, IRD, Ifremer, LEMAR, Plouzané, France

[18] Vesta, PBC, Southampton, New York, USA

[19] College of Marine Science, University of South Florida, Florida, USA

[20] Department of Oceanography, National Sun Yat-sen University, Kaohsiung, Taiwan

[21] Flathead Lake Biological Station, University of Montana, Polson, Montana, USA

[22] Institute of Marine Sciences (ICM-CSIC), Barcelona, Spain

[23] Institute of Marine Sciences of Andalucía (ICMAN), Consejo Superior de Investigaciones Científicas (CSIC), Campus Río San Pedro, Puerto Real, Spain

[24] Department of Environmental Sciences, University of Virginia, Charlottesville, Virginia, USA

[25] CNRS Observatoire océanologique, Banyuls sur mer, France

[26] Center for Oceanographic Research COPAS Coastal, Universidad de Concepción, Chile

[27] Ocean and Earth Science, National Oceanography Centre, University of Southampton, Southampton, UK

[28] Department of Oceanography, Dalhousie University, Halifax, Nova Scotia, Canada

[29] Department of Ecology, Environment, and Plant Sciences, Stockholm University, Stockholm, Sweden

[30] Institute of Plankton Eco-engineering, Soka University, Hachioji, Tokyo, Japan

[31] Graduate School of Environmental Science, Hokkaido University, Kita-Ku, Sapporo, Japan

[32] Spanish Institute of Oceanography (IEO-CSIC), Centro Oceanografico de Vigo, Spain

[33] Columbia River Inter-Tribal Fish Commission, Portland, Oregon, USA

[34] Environmental Studies, Soka University of America, Aliso Viejo, California, USA

[35] Ocean Sciences Department, University of California at Santa Cruz, Santa Cruz, California, USA

[36] Department of Geosciences, Princeton University, Princeton, New Jersey, USA

[37] Second Institute of Oceanography, Ministry of Natural Resources, Hangzhou, Zhejiang, China

[38] Department of Oceanography, University of Hawai'i at Mānoa, Honolulu, Hawaii, USA

[39] Marine Biogeochemistry, GEOMAR Helmholtz Centre for Ocean Research Kiel, Düstern, Kiel, Germany

[40] Department of Earth, Ocean, & Atmospheric Science, Florida State University, Tallahassee, Florida, USA

[41] Geosciences Division, Physical Research Laboratory, Ahmedabad, India

[42] Department of Biology, Dalhousie University, Halifax, Nova Scotia, Canada

[43] Department of Ocean Science, The Hong Kong University of Science and Technology, Hong Kong, China

[44] Key Laboratory of Tropical Marine Bio-resources and Ecology, Guangdong Provincial Key Laboratory of Applied Marine Biology, South China Sea Institute of Oceanology, Chinese Academy of Sciences, Guangzhou, Guangdong, China

[45] French National Research Institute for Sustainable Development, IRD, Marseille, France

[46] Department of Biology, DIAS, University of Southern Denmark, Odense, Denmark

[47] Centro de Investigación Mariña da Universidade de Vigo (CIM-UVigo), Departamento de Ecoloxía e Bioloxía Animal, Universidade de Vigo, Campus Lagoas-Marcosende, Vigo, Spain

[48] Division of Biological and Environmental Sciences, Faculty of Natural Sciences, University of Stirling, Stirling, Scotland, UK

[49] Earth System Science, Stanford University, Stanford, California, USA

[50] Max Planck Institute for Marine Microbiology, Bremen, Germany

[51] Department of Biology, University of Massachusetts Dartmouth, Dartmouth, Massachusetts, USA

[52] Department of Earth, Ocean and Ecological Sciences, University of Liverpool, Liverpool, UK

[53] Department of Oceanography, Dalhousie University, Halifax, Nova Scotia, Canada

[54] Department of Ocean and Atmospheric Sciences, Old Dominion University, Norfolk, Virginia, USA

[55] Center for Global Environmental Research, National Institute for Environmental Studies, Tsukuba, Japan

[56] OHSU-PSU School of Public Health, Oregon Health and Science University Portland, Portland, Oregon, USA

[57] Flourishing Oceans, Minderoo Foundation, Broadway, Nedlands, WA, Australia

[58] Israel Oceanographic and Limnological Research, National Institute of Oceanography, Haifa, Israel

[59] Centro Oceanográfico de Málaga, Instituto Español de Oceanografía (IEO, CSIC), Fuengirola, Spain

[60] Department of Biology, University of Copenhagen, Helsingør, Denmark

[61] Analytical, Environmental and Geo-Chemistry & Earth System Sciences, Vrije Universiteit Brussel, Brussels, Belgium

[62] National Oceanography Centre, Southampton, UK

[63] CSIR-National Institute of Oceanography, Regional Cente Waltair, Visakhapatnam, India

[64] Institute for Chemical Research, Kyoto University, Kyoto, Japan

[65] Department of Marine and Coastal Sciences, Rutgers University, New Brunswick New Jersey, USA

[66] Climate Change Cluster, University of Technology Sydney, Sydney, New South Wales, Australia

[67] Atmosphere and Ocean Research Institute, The University of Tokyo, Chiba, Japan

[68] Research Centre for Indian Ocean Ecosystem, Tianjin University of Science and Technology, Tianjin, China

[69] College of Marine Science and Technology, China University of Geosciences (Wuhan), Wuhan, Hubei, China

[70] Faculty of Environmental Earth Science, Hokkaido University, Sapporo, Japan

[71] Graduate School of Agricultural and Life Sciences, The University of Tokyo, Tokyo, Japan

[72] Québec-Océan and Takuvik, Department of Biology, Laval University, Québec, Canada

[73] School of Natural and Environmental Sciences, Newcastle University, Newcastle upon Tyne, UK

[#] These authors contributed equally.

*Correspondence to*: Ya-Wei Luo (ywluo@xmu.edu.cn)

**Abstract.** Marine diazotrophs convert dinitrogen ($N_2$) gas into bioavailable nitrogen (N), supporting life in the global ocean. In 2012, the first version of the global oceanic diazotroph database (version 1) was published. Here, we present an updated version of the database (version 2), significantly increasing the number of *in situ* diazotrophic measurements from 13,565 to 55, 286. Data points for $N_2$ fixation rates, diazotrophic cell abundance, and *nifH* gene copy abundance have increased by 184%,

86%, and 809%, respectively. Version 2 includes two new datasheets for the *nifH* gene copy abundance of non-cyanobacterial diazotrophs and cell-specific $N_2$ fixation rates. The measurements of $N_2$ fixation rates approximately follow a log-normal distribution in both version 1 and version 2. However, version 2 considerably extends both the left and right tails of the distribution. Consequently, when estimating global oceanic $N_2$ fixation rates using the geometric means of different ocean basins, version 1 and version 2 yield similar rates (43–57 versus 45–63 Tg N yr$^{-1}$; ranges based on one geometric standard

error). In contrast, when using arithmetic means, version 2 suggests a significantly higher rate of 223±30 Tg N yr$^{-1}$ (mean±standard error; same hereafter) compared to version 1 (74±7 Tg N yr$^{-1}$). Specifically, substantial rate increases are estimated for the South Pacific Ocean (88±23 versus 20±2 Tg N yr$^{-1}$), primarily driven by measurements in the southwestern subtropic, and the North Atlantic Ocean (40±9 versus 10±2 Tg N yr$^{-1}$). Moreover, version 2 estimates the $N_2$ fixation rate in the Indian Ocean to be 35±14 Tg N yr$^{-1}$, which could not be estimated using version 1 due to limited data availability.

Furthermore, a comparison of $N_2$ fixation rates obtained through different measurement methods at the same months, locations, and depths reveals that the conventional $^{15}N_2$ bubble method yields lower rates in 69% cases compared to the new $^{15}N_2$ dissolution method. This updated version of the database can facilitate future studies in marine ecology and biogeochemistry. The database is stored at the Figshare repository (https://doi.org/10.6084/m9.figshare.21677687) (Shao et al., 2022).

## 1 Introduction

Dinitrogen ($N_2$) fixation is a process carried out by select prokaryotes (diazotrophs) capable of converting $N_2$ gas, which is not usable by most organisms, into bioavailable nitrogen (N). In the sunlit surface ocean, where dissolved inorganic forms of N such as nitrate ($NO_3^-$) and ammonium ($NH_4^+$) are scarce, $N_2$ fixation plays an important role in providing N that can contribute to primary production, particularly in oligotrophic regions (Wang et al., 2019; Gruber, 2008). Globally, $N_2$ fixation serves to compensate, at least partially, for fixed N removed via denitrification and anammox (Deutsch et al., 2007; Gruber, 2019).

Marine diazotrophs include three main types of cyanobacteria (Zehr, 2011): (1) nonheterocystous filamentous cyanobacteria (e.g., *Trichodesmium*); (2) heterocystous cyanobacteria like *Richelia*, which may form diatom-diazotroph associations (DDAs); and (3) unicellular cyanobacteria (UCYNs). Non-cyanobacterial diazotrophs (NCDs) have also been widely detected in the ocean (Bombar et al., 2016; Delmont et al., 2021; Moisander et al., 2017). However, the contribution of NCDs to marine $N_2$ fixation has not been directly quantified, despite a few studies that have reported $N_2$ fixation by putative

NCDs at the cellular level (Harding et al., 2022; Bentzon-Tilia et al., 2015a).

Diazotroph abundance has been estimated from *nifH* gene copies using qPCR assays (Church et al., 2005b) or droplet digital PCR (ddPCR) (Gradoville et al., 2017). The abundance of some cyanobacterial diazotrophs can also be obtained by counting them directly using microscopy-based techniques and in some cases flow cytometry. A recent work combined an image recognition pipeline with molecular mapping of the *nifH* gene to quantify diazotrophs in the Tara Oceans dataset

(Karlusich et al. 2021). *NifH* gene copies have been more frequently measured than microscopy-based cell counts and can be more useful when evaluating the abundance of different diazotrophic groups. Caution must be taken because there can be discrepancies between cell-count-based and *nifH*-based diazotrophic abundances (Luo et al., 2012), a finding largely attributed to large variations in the number of *nifH* copies per diazotroph cell, thus far observed particularly in *Trichodesmium* and heterocystous cyanobacteria (Sargent et al., 2016; White et al., 2018; Karlusich et al., 2021). However, a recent regional study

spanning over 200 km in the North Pacific Subtropical Gyre has found a statistically significant linear correlation between the abundances of the *nifH* gene and cell counts in the UCYN-B (i.e., *Crocosphaera*) (linear slope = 1.82) and heterocystous cyanobacteria (*Richelia* and *Calothrix*; linear slope from 1.51-2.58) but not in *Trichodesmium* (Gradoville et al., 2022). A recent discussion highlighted the influence of the uncertainty in gene copy conversion to biomass and the need for further investigations on how to best take advantage of gene copy data for global diazotroph biogeography modelling purposes (Meiler

et al., 2022; Zehr and Riemann, 2023); however, there is an agreement that quantifying gene counts is a powerful tool for studying marine diazotroph distributions (Meiler et al., 2023; Zehr and Riemann, 2023). Meiler et al., (2023) proposed a number of topics of study for this field moving forward; Gradoville et al. (2022) concluded that "we hope that future studies report *nifH*:cell and explore the mechanisms controlling this ratio." Both gene based and microscopy cell counts have innate biases, which should be elucidated in future studies.

Given the importance of $N_2$ fixation to ocean ecology and biogeochemistry, it is imperative that a database of up-to-date $N_2$ fixation and diazotrophic abundance measurements be maintained. Currently, global estimates of marine fixed N inputs via $N_2$ fixation rate mostly ranges from 100 to 170 Tg N $yr^{-1}$ (see summary in Zhang et al., 2020). This value, together with other bioavailable N sources to the ocean including riverine input and atmospheric deposition, is considerably lower than estimates of N losses from the ocean such as denitrification, anammox and sediment burial (Zhang et al., 2020; Gruber, 2008; Zehr and

Capone, 2021). While the overestimation of the N losses cannot be ruled out, one of possible reasons for this imbalance is the inaccurate estimation of global marine $N_2$ fixation due to limited spatio-temporal coverage of rate measurements and the different methods employed in $N_2$ fixation assays (White et al., 2020). Another possible reason is the limited knowledge of ecological niches of $N_2$ fixing organisms. Over the last decade, the realm of marine $N_2$ fixation has been expanded to include numerous non-paradigmatic habitats. Coastal (Mulholland et al., 2012; Bentzon-Tilia et al., 2015b; Mulholland et al., 2019;

Tang et al., 2020; Turk-Kubo et al., 2021), subpolar (Sato et al., 2021; Shiozaki et al., 2018c), and even polar ocean regions (Blais et al., 2012; Sipler et al., 2017; Harding et al., 2018; Shiozaki et al., 2020) have demonstrated $N_2$ fixation. Notably, $N_2$ fixation in aphotic waters remains debated (Bonnet et al., 2013; Farnelid et al., 2013; Selden et al., 2021a; Rahav et al., 2013b; Hamersley et al., 2011; Benavides et al., 2018a; Moisander et al., 2017). Other studies have also suggested that NCDs may be significant contributors to marine $N_2$ fixation (Shiozaki et al., 2014b; Turk-Kubo et al., 2022; Geisler et al., 2020; Delmont et

al., 2021; Karlusich et al., 2021; Bombar et al., 2016; Moisander et al., 2017) and may occupy different niches from cyanobacterial diazotrophs (Shao and Luo, 2022).

Luo et al. (2012) compiled the first global oceanic diazotrophic database including *in situ* measurements of $N_2$ fixation rates and cell-count-based and *nifH*-based diazotrophic abundance. Several years later, two studies supplemented the database with a collection of some newly reported diazotrophic data (Tang and Cassar, 2019; Tang et al., 2019), although a substantial

amount of additional data remained to be included. Here, we present an updated version of the global oceanic diazotrophic database with data not yet compiled. We describe the database information, a summary of the data updates, measurement methods and data distribution. Furthermore, we conduct a first-order estimation of the global oceanic $N_2$ fixation rate using the updated version of the database. In light of the aforementioned concerns of *nifH*:cell and various $N_2$ fixation methods (*see* **Section 2.3**), we also discuss the significance of employing different methodological approaches to estimate $N_2$ fixation rates

and abundance metrics. We use the data available in the database to analyze the discrepancies between $N_2$ fixation rates using $^{15}N_2$ bubble and dissolution methods, and compare the observed ranges of *nifH* gene copies and diazotrophic cell abundance.

## 2 Data and methods

### 2.1 Database summary

This study updated the original global oceanic diazotrophic database of Luo et al. (2012) (version 1 hereafter) with new

*in situ* measurements of $N_2$ fixation rates and abundances of diazotrophic cells and *nifH* gene copies. Together there were 55, 286 diazotrophic data points in the updated database (version 2 hereafter) (**Tables 1–3**), including 13,565 data points from version 1 (Luo et al., 2012), 6,736 measured in 2012 – 2018 and compiled by two previous studies (Tang et al., 2019; Tang and Cassar, 2019), 26,597 data points measured in 1979 – 2023 and compiled by this study, and 8,388 NCD data mostly from Turk-Kubo et al. (2022) (see below). In version 2, some errors in the datasets of Tang et al. (2019) (mostly caused by unit

conversions) were also corrected.

Version 2 was composed of six main sub-databases: (1) 9,231 volumetric $N_2$ fixation rates (5,853 new data points) (**Tables 1 & 4**); (2) 2,590 depth-integrated $N_2$ fixation rates (1,805 new data points) (**Tables 1 & 4**); (3) 9,040 volumetric cell abundances (4,154 new data points) (**Tables 2 & 5**); (4) 1,784 depth-integrated cell abundances (859 new data points) (**Tables 2 & 5**); (5) 29,655 volumetric *nifH* gene copy abundances (26,506 new data points) (**Tables 3 & 6**); and (6) 2,986 depth-

integrated *nifH* gene copy abundances (2,544 new data points) (**Tables 3 & 6**). Please be aware that 2,416 $N_2$ fixation rates were measured with incubation periods less than 24 hours; they were listed in separate spreadsheets in the database for reasons discussed in **Section 2.3**. Additionally we included a compiled NCD dataset (Turk-Kubo et al., 2022) in the database, which contained 7,919 *nifH* gene copy abundances of primarily the most studied phylotype NCD Gamma A (Shao and Luo, 2022; Langlois et al., 2015), also referred to as 24774A11 (Moisander et al., 2012) and UMB (Bird et al., 2005), as well as other

phylotypes, and updated that compilation with 469 additional *nifH* gene copy abundances of NCDs published more recently

(Turk-Kubo et al., 2021; Sato et al., 2022; Moore et al., 2018; Reeder et al., 2022; Wen et al., 2022; Bonnet et al., 2023). We also collected 468 cell-specific *in situ* $N_2$ fixation rates and added them to version 2 (**Table 7**).

Depth-integrated data were either provided directly in published papers or calculated as part of this study for those vertical profiles with at least 3 volumetric data points in each profile. The measurements within a profile were first interpolated linearly with depth, with the shallowest datum representing the level between the sea surface and the depth of that datum. The profile was then integrated from the sea surface to the deepest recorded measurement. Most vertical profiles of $N_2$ fixation rates were measured within the euphotic zone, with a few studies extending measurements to several hundred meters or deeper. In these cases, we only integrated to the deepest data point above 200 m, taking into account the scarcity of aphotic $N_2$ fixation measurements in the global ocean and their controversial contribution to the global budget (Benavides et al., 2018a). As a result, it was possible that certain measurements below the euphotic zone but above 200 m were included in the integration. However, these measurements would typically have minimal impact on the depth-integrated $N_2$ fixation rates due to their low rates and limited vertical extent in this range.

$N_2$ fixation rates were measured for whole seawater samples, for different size fractions (> 10 μm and < 10 μm), or specifically for *Trichodesmium* and heterocystous cyanobacteria. When whole-water $N_2$ fixation rates were not reported, total $N_2$ fixation rates were calculated as the sum of the $N_2$ fixation rates of available groups.

The cyanobacterial diazotrophic abundance data in version 2 were grouped into three taxonomic categories: *Trichodesmium*, UCYN, and heterocystous cyanobacteria. The UCYN abundance data were further grouped into UCYN-A, UCYN-B, and UCYN-C. Four sublineages of UCYN-A, including UCYN-A1, UCYN-A2, UCYN-A3, and UCYN-A4, have been identified (Thompson et al., 2014; Farnelid et al., 2016). UCYN-A1 and UCYN-A2 have significant distinctions in the sizes and species of their symbiotic hosts, with the former living in relatively smaller hosts (Thompson et al., 2014; Martínez-Pérez et al., 2016; Cornejo-Castillo et al., 2016). Hence, in addition to recording the total *nifH* gene copy abundance of UCYN-A in our database, the *nifH* gene copy abundances of its sublineages were also included if reported. Heterocystous cyanobacterial abundance was grouped into *Richelia intracelluaris* (het-1 and het-2, associated with *Hemiaulus* and *Rhizosolenia*, respectively) and *Richelia rhizosoleniae* (het-3, named *Calothrix* sp. before, associated with *Chaetoceros*) (Foster et al., 2022b).

Sampling information (latitude, longitude, depth and time) was provided for each data point. Physical, chemical and biological parameters, including temperature, salinity, and concentrations of nitrate, phosphate, iron and chlorophyll *a*, were also included when available.

**Table 1.** Summary of number of data points for $N_2$ fixation rates by category. Measurements with incubation periods of 24 hours or of shorter than 24 hours are summarized separately.

| | Original database | | New data added in version 2 | | | | sum | |
| --- | --- | --- | --- | --- | --- | --- | --- | --- |
| | | | Tang et al., 2019 | | This study | | | |
| **Volumetric $N_2$ fixation rate** | | | | | | | | |
| | 24 h | < 24 h | 24 h | < 24 h | 24 h | < 24 h | 24 h | < 24 h |
| *Trichodesmium* | | 677 | | | 6 | | 6 | 677 |
| Heterocystous | | 185 | | | | | | 185 |
| < 10 μm | 228 | 28 | 75 | | 265 | 6 | 568 | 34 |
| > 10 μm | 54 | 36 | 9 | 21 | 51 | 6 | 114 | 63 |
| Whole seawater | 1,743 | 427 | 1,169 | 171 | 3,782 | 292 | 6,694 | 890 |
| Total | 2,025 | 1,353 | 1,253 | 192 | 4,104 | 304 | 7,382 | 1,849 |
| Proportion in version 2 | 21.9% | 14.6% | 13.6% | 2.1% | 44.5% | 3.3% | | |
| **Depth-integrated $N_2$ fixation rate** | | | | | | | | |
| | 24 h | < 24 h | 24 h | < 24 h | 24 h | < 24 h | 24 h | < 24 h |
| *Trichodesmium* | 40 | 206 | 81 | 8 | | 9 | 121 | 223 |
| Heterocystous | 1 | 65 | 80 | 12 | | | 81 | 77 |
| < 10 μm | 28 | 18 | 7 | 12 | 21 | 2 | 56 | 32 |
| > 10 μm | 3 | 32 | | | 21 | 2 | 24 | 34 |
| Whole seawater | 285 | 107 | 500 | 53 | 956 | 41 | 1,741 | 201 |
| Total | 357 | 428 | 668 | 85 | 998 | 54 | 2,023 | 567 |
| Proportion in version 2 | 13.8% | 16.5% | 25.8% | 3.3% | 38.5% | 2.1% | | |

**Table 2.** Summary of number of data points for diazotrophic cell abundances. UCYNs include UCYN-A, UCYN-B and unclassified UCYNs. Heterocystous cyanobacteria include Het-1, Het-2 and Het-3.

| | Original database | New data added to version 2 | Sum |
| --- | --- | --- | --- |
| **Volumetric cell abundances** | | | |
| *Trichodesmium* | 3,274 | 2,812 | 6,086 |
| UCYN | | 139 | 139 |
| Heterocystous cyanobacteria | 1,612 | 1203 | 2,815 |
| Total | 4,886 | 4,154 | 9,040 |
| Proportion in version 2 | 54.1% | 45.9% | |
| **Depth-integrated cell abundances** | | | |
| *Trichodesmium* | 620 | 692 | 1,312 |
| UCYN | | 19 | 19 |
| Heterocystous | 305 | 148 | 453 |

| | | | | |
|---|---|---|---|---|
| Total | 925 | 859 | 1,784 | |
| Proportion in version 2 | 51.9% | 48.1% | | |

**Table 3.** Summary of number of data points for *nifH* gene copy abundances. UCYNs include UCYN-A1, UCYN-A2, UCYN-B and UCYN-C. Heterocystous cyanobacteria include Het-1, Het-2 and Het-3.

| | Original database | New data added to version 2 | | Sum |
|---|---|---|---|---|
| | | Tang & Cassar, 2019 | This study | |
| **Volumetric *nifH* gene copy abundances** | | | | |
| *Trichodesmium* | 758 | 770 | 3,165 | 4,693 |
| UCYN | 1,792 | 2,640 | 6,903 | 11,309 |
| Heterocystous cyanobacteria | 599 | 505 | 4,135 | 5,239 |
| NCDs | | | 8,388 | 8,388 |
| Total | 3,149 | 3915 | 22,591 | 29,655 |
| Proportion in version 2 | 10.6% | 13.2% | 76.2% | |
| **Depth-integrated *nifH* gene copy abundances** | | | | |
| *Trichodesmium* | 105 | 123 | 408 | 636 |
| UCYN | 263 | 418 | 871 | 1552 |
| Heterocystous | 74 | 82 | 642 | 798 |
| Total | 442 | 623 | 1,921 | 2,986 |
| Proportion in version 2 | 14.8% | 20.9% | 64.3% | |

235

**Table 4.** Summary of new data points of $N_2$ fixation rates added to the version 2 of the database.

| Reference | Region | *Trichodesmium* | Heterocystous | < 10 μm Diazotrophs | > 10 μm Diazotrophs | Whole Seawater | Depth-integrated data |
|---|---|---|---|---|---|---|---|
| **Part 1. Incubation periods of 24 hours** | | | | | | | |
| Ahmed et al. (2017) | E Arabian Sea | | | | | 19 | 5[a] |
| Benavides et al. (2016a) | Mediterranean Sea | | | | | 10 | |
| Benavides et al. (2018a) | Tropical SW Pacific | | | | | 59 | |
| Benavides et al. (2022a) | Tropical SW Pacific | | | | | 38 | |
| Benavides et al. (2017) | SW Pacific | | | | | 2 | |

| Reference | Region | *Tricho-desmium* | Hetero-cystous | < 10 μm Diazotrophs | > 10 μm Diazotrophs | Whole Seawater | Depth-integrated data |
|---|---|---|---|---|---|---|---|
| Benavides et al. (2021) | S Pacific | | | | | 41 | |
| Benavides et al. (2022b) | S Pacific | 6 | | | | 6 | 2 |
| Bentzon-Tilia et al. (2015b) | Baltic Sea | | | | | 23 | 23[a] |
| Berthelot et al. (2017) | Tropical W Pacific | | | | | 48 | 12[a] |
| Biegala and Raimbault (2008) | SW Pacific | | | 12 | 12 | 12 | 9 |
| Blais et al. (2012) | Arctic Ocean | | | | | 18 | 12 |
| Bombar et al. (2015) | Subtropical N Pacific | | | | | 20 | 2 |
| Bonnet et al. (2013) | Tropical SE Pacific | | | | | | 8[a] |
| Bonnet et al. (2018) | Tropical SW Pacific | | | | | 102 | 14 |
| Bonnet et al. (2015) | SW Pacific | | | 126 | | 128 | 30[a] |
| Bonnet et al. (2023) | Subtropical S Pacific | | | | | 84 | 14 |
| Böttjer et al. (2017) | Subtropical N Pacific | | | | | 243 | 108[a] |
| Cerdan-Garcia et al. (2021) | subtropical N Atlantic | | | | | 15 | |
| Chang et al. (2019) | Tropical SE Pacific | | | | | 37 | |
| Chen et al. (2019) | W Pacific Ocean | | | | | 95 | 16 |
| Dekaezemacker et al. (2013) | Tropical SE Pacific | | | | | 43 | 10 |
| Dugenne et al. (2023) | Subtropical N Pacific | | | | | 30 | 5 |
| Fernandez et al. (2015) | Central Chile Upwelling System | | | | | 55 | 14[a] |
| Fernández-Castro et al. (2015) | Atlantic, Pacific and Indian Oceans | | | | | 177 | 43[a] |
| Fonseca-Batista et al. (2017) | E Atlantic | | | | | 56 | 14 |
| Fonseca-Batista et al. (2019) | Temperate NE Atlantic | | | | | 46 | 10[a] |
| Foster et al. (2009) | Red Sea | | | | | 26 | |
| Foster et al. (unpublished data) | E tropical S Pacific | | | | | 23 | 5 |

| Reference | Region | *Tricho-desmium* | Hetero-cystous | < 10 μm Diazotrophs | > 10 μm Diazotrophs | Whole Seawater | Depth-integrated data |
|---|---|---|---|---|---|---|---|
| Garcia et al. (2007) | SW Pacific | | | | | | 1[a] |
| Gradoville et al. (2020) | N Pacific | | | | | 20 | |
| Gradoville et al. (2017) | S Pacific; N Pacific | | | | | 30 | 5 |
| Großkopf et al. (2012) | Atlantic Ocean | | | | | 39 | 17 |
| Hallstrøm et al. (2022) | NE Atlantic | | | | | 59 | 11[a] |
| Harding et al. (2018) | Arctic Ocean | | | | | 38 | |
| Harding et al. (2022) | Subtropical N Pacific | | | | | 7 | |
| Hörstmann et al. (2021) | S Indian Ocean; Southern Ocean | | | | | 13 | |
| Ibello et al. (2010) | Mediterranean Sea | | | | | 21 | 14[a] |
| Jayakumar et al. (2017) | Tropical NE Pacific | | | | | 32 | 7 |
| Jiang et al. (2023) | East China Sea and Southern Yellow Sea | | | | | 97 | 29[a] |
| Kittu et al. (2023) | Tropical SE Pacific | | | | | 103 | 21 |
| Knapp et al. (2016) | Tropical SE Pacific | | | | | | 6[a] |
| Konno et al. (2010) | NW Pacific | | | | | | 16[a] |
| Krupke et al. (2015) | Subtropical NE Atlantic | | | | | 1 | |
| Kumari et al. (2022) | Bay of Bengal | | | | | 97 | 18[a] |
| Landou et al. (2023) | Red Sea | | | | | 72 | 22[a] |
| Li et al. (2020) | N South China Sea; East China Sea | | | | | 68 | 15[a] |
| Liu et al. (2020) | South China Sea | | | | | 25 | 5[a] |
| Loescher et al. (2014) | Tropical SE Pacific | | | | | 30 | 5[a] |
| Loick-Wilde et al. (2015) | Amazon River | | | | | | 54[a] |
| Loick-Wilde et al. (2019) | Tropical W Pacific | | | | | 8 | |
| Lory et al. (2022) | Tropical SE Pacific | | | | | 5 | |
| Löscher et al. (2016) | Tropical SW Pacific | | | | | 225 | 31+4[a] |
| Löscher et al. (2020) | Bay of Bengal | | | | | 18 | |
| Lu et al. (2018) | Equatorial W Pacific | | | | | 3 | 3[a] |
| Martínez-Pérez et al. (2016) | Tropical N Atlantic | | | | | 84 | 14 |

| Reference | Region | *Trichodesmium* | Hetero-cystous | < 10 μm Diazotrophs | > 10 μm Diazotrophs | Whole Seawater | Depth-integrated data |
|---|---|---|---|---|---|---|---|
| Messer et al. (2016) | S Pacific | | | | | 27 | |
| Messer et al. (2021) | S Australian Gulf System | | | 10 | | 10 | |
| Mills et al. (2020) | California Current System | | | | | 4 | |
| Moreira-Coello et al. (2017) | the coastal NW Iberian upwelling | | | 30 | | | 10[a] |
| Mulholland et al. (2019) | NW Atlantic | | | | | 402 | 242[a] |
| Needoba et al. (2007) | Temperate N Pacific | | | | | 2 | |
| Palter et al. (2020) | Gulf stream | | | | | 7 | |
| Raes et al. (2014) | E Indian | | | | | 31 | |
| Raes et al. (2020) | S Pacific | | | | | 118 | |
| Rahav et al. (2013b); Rahav et al. (2015) | Red Sea and E Mediterranean Sea | | | | | 62 | 10 |
| Rahav et al. (2013a) | Mediterranean Sea | | | | | 8 | |
| Rahav et al. (2016) | Mediterranean Sea | | | | | | 3[a] |
| Reeder et al. (2022) | S Baltic Sea | | | | | 15 | 5 |
| Riou et al. (2016) | N Atlantic | | | | | 24 | 6 |
| Sarma et al. (2020) | Bay of Bengal | | | | | 2 | |
| Sato et al. (2021) | Subarctic Sea of Japan; Sea of Okhotsk | | | | | 31 | 3 |
| Sato et al. (2022) | E Indian | | | | | 73 | 18[a] |
| Saulia et al. (2020) | Tropical SW Pacific | | | | | 71 | 71[a] |
| Selden et al. (2019) | Tropical NE Pacific | | | | | 8 | 16[a] |
| Selden et al. (2021b) | NW Atlantic | | | | | 93 | 26[a] |
| Selden et al. (2021a) | Tropical SE Pacific | | | | | 125 | 19 |
| Shiozaki et al. (2013) | W Pacific | | | | | 50 | 10 |
| Shiozaki et al. (2014a) | SW Pacific | | | 40 | | 42 | |
| Shiozaki et al. (2014b) | Indian Ocean | | | 26 | | 26 | 6[a] |
| Shiozaki et al. (2015a) | NW Pacific | | | | | 73 | 11 |
| Shiozaki et al. (2015b) | N Pacific | | | | | 112 | 22[a] |
| Shiozaki et al. (2017) | N Pacific | | | | | 74 | 15[a] |
| Shiozaki et al. (2018b) | W Arctic | | | | | 84 | 21[a] |

| Reference | Region | Tricho-desmium | Hetero-cystous | < 10 μm Diazotrophs | > 10 μm Diazotrophs | Whole Seawater | Depth-integrated data |
|---|---|---|---|---|---|---|---|
| Shiozaki et al. (2018c) | S Pacific | | | | | 65 | 15[a] |
| Shiozaki et al. (2020) | Antarctic Coast | | | | | 53 | 15 |
| Singh et al. (2017) | Tropical NE Atlantic | | | | | 52 | 13 |
| Sipler et al. (2017) | Arctic Ocean | | | | | 8 | |
| Sohm et al. (2011) | S Atlantic | | | | | 12 | 3[a] |
| Subramaniam et al. (2008) | Tropical N Atlantic | | | | | | 242[a] |
| Subramaniam et al. (2013) | Atlantic Ocean | | | | | 96 | 24[a] |
| Tang et al. (2020) | N Atlantic | | | | | 15 | |
| Turk-Kubo et al. (2012) | Tropical N Atlantic | | | 27 | | | 7 |
| Turk-Kubo et al. (2021) | Southern California Current System | | | 21 | | 64 | 14 |
| Wasmund et al. (2015) | S Atlantic | | | | | | 66[a] |
| Watkins-Brandt et al. (2011) | N Pacific | | | | | | 1[a] |
| Wen et al. (2022) | Tropical NW Pacific | | | | | 143 | 22[a] |
| White et al. (2018) | Subtropical N Pacific | | | | | 43 | 13[a] |
| Wilson et al. (2012) | N Pacific | | | | | 9 | 4[a] |
| Wilson et al. (2017) | Subtropical N Pacific | | | | | 33 | |
| Wu et al. (2021) | Eastern Indian Ocean | | | 48 | 48 | 48 | 7 |
| Yogev et al. (2011)[b] | E Mediterranean Sea | | | | | 16 | 32[a] |
| Zhang et al. (2015) | South China Sea | | | | | 82 | 11 |
| Zhang et al. (2019) | Tropical NW Pacific | | | | | 87 | 9[a] |
| **Part 2. Incubation period less than 24 hours** | | | | | | | |
| Agawin et al. (2013) | Subtropical Atlantic | | | | 21 | 17 | |
| Benavides et al. (2013b) | subtropical N Atlantic | | | | | 38 | |
| Benavides et al. (2014) | the coastal Namibian upwelling system | | | | | 14 | 3 |
| Bhavya et al. (2016) | Arabian Sea | | | | | 4 | |
| Biegala and Raimbault (2008) | SW Pacific | | | 6 | 6 | 6 | 6 |
| Bombar et al. (2011) | South China Sea | | | | | 15 | |

| Reference | Region | *Tricho-desmium* | Hetero-cystous | < 10 μm Diazotrophs | > 10 μm Diazotrophs | Whole Seawater | Depth-integrated data |
|---|---|---|---|---|---|---|---|
| Fernandez et al. (2015) | Central Chile Upwelling System | | | | | 29 | |
| Foster et al. (2013) | Subtropical N Pacific | | | | | 3 | |
| Foster et al. (2022a) | Tropical NW Atlantic | | | | | 45 | 9 |
| Foster et al. (unpublished data) | N Atlantic | | | | | 24 | 5 |
| Gandhi et al. (2011) | E Arabian Sea | | | | | 28 | 7[a] |
| Halm et al. (2012) | S Pacific | | | | | 43 | 10[a] |
| Kromkamp et al. (1997) | Indian Ocean | | | | | | 9[a] |
| Krupke et al. (2013) | Subtropical N Atlantic | | | | | 6 | |
| Krupke et al. (2014) | N Atlantic | | | | | 42 | 44[a] |
| Kumar et al. (2017) | E Arabian Sea | | | | | 12 | 3 |
| Chen et al. (2014) | South China Sea | | | | | | 24[a] |
| Sahoo et al. (2021) | Bay of Bengal | | | | | | 6[a] |
| Saxena et al. (2020) | Bay of Bengal | | | | | 32 | 8[a] |
| Singh et al. (2019) | E Arabian Sea | | | | | 20 | 5[a] |
| Wang et al. (2021) | NW Atlantic | | | | | 85 | |
| Total | | 6 | 0 | 346 | 87 | 5414 | 1805 |

[a] Data are reported by data providers as depth-integrated $N_2$ fixation rates (unlabelled data computed by integrating profiles of volumetric $N_2$ fixation rate data).

[b] $N_2$ fixation rate incubation time during 24-30 hrs.

**Table 5.** Summary of new data points of cell-count-based abundances added to the version 2 of the database. The data were measured using microscopy-based method (method A), TSA/CARD-FISH (method B), flow cytometer (method C) or image recognition (method D). UCYNs include UCYN-A, UCYN-B and unclassified UCYNs. Heterocystous cyanobacteria include Het-1, Het-2 and Het-3.

| Reference | Region | Method | *Tricho-desmium* | UCYN | Heterocystous cyanobacteria | Depth-integrated data |
|---|---|---|---|---|---|---|
| Biegala and Raimbault (2008) | SW Pacific | B | | 15 | | |

| Reference | Region | Method | *Tricho-desmium* | UCYN | Heterocystous cyanobacteria | Depth-integrated data |
|---|---|---|---|---|---|---|
| Bif and Yunes (2017) | S Atlantic | A | 16 | | | |
| Campbell et al. (2005) | SW Pacific | A | 462 | | 259 | 33[a] |
| Detoni et al. (2016) | S Atlantic | A | 14 | | | |
| Dugenne et al. (2023); Gradoville et al. (2022) | N Pacific Subtropic Gyre | C | 4 | 4 | 7 | |
| Dupouy et al. (2011) | SW Pacific | A | 18 | | | |
| Estrada et al. (2016) | Global | A | 407 | | 407 | |
| Fernández et al. (2010) | Global | A | | | | 40[a] |
| Foster et al. (2022a) | W tropical N Atlantic | A | | | 37 | 9 |
| Foster et al. (unpublished data) | N Atlantic | A | | | 54 | |
| Hegde et al. (2008) | Bay of Bengal | A | 135 | | | |
| Holl et al. (2007) | N Atlantic | A | | | | 10[a] |
| Jiang et al. (2017) | E China Sea | A | 1174 | | | 252[a] |
| Jiang et al. (2023) | E China Sea | A | 39 | | 39 | 78[a] |
| Krupke et al. (2013) | N Atlantic | B | | 9 | | |
| Le Moal and Biegala (2009) | Mediterranean Sea | B | | 17 | | |
| Le Moal et al. (2011) | Mediterranean Sea | B | | 18 | | |
| Lory et al. (2022) | S Pacific | A | 3 | | | |
| Lu et al. (2018) | W Eq. Pacific | A | 2 | | | |
| Martínez-Pérez et al. (2016) | Tropical N Atlantic | A | | 56 | | 14 |
| Masotti et al. (2007) | SW Pacific | A | 20 | | | 5 |
| Mompeán et al. (2013) | N Atlantic | A | | | | 43[a] |
| Mompeán et al. (2016) | Global | A | | | | 141[a] |
| Karlusich et al. (2021) | Global | D | 46 | | 81 | |
| Riou et al. (2016) | N Atlantic | B | | 20 | | 5 |
| Sahu et al. (2017) | Bay of Bengal | A | 14 | | | |
| Shiozaki et al. (2013) | W Pacific | A | 10 | | 12 | |
| Shiozaki et al. (2015a) | NW Pacific | A | 60 | | | 10 |
| Subramaniam et al. (2008) | N Atlantic | A | | | | 162[a] |

| Reference | Region | Method | *Tricho-desmium* | UCYN | Heterocystous cyanobacteria | Depth-integrated data |
|---|---|---|---|---|---|---|
| Tenório et al. (2018) | SW Pacific | A | 81 | | | 19[a] |
| White et al. (2018) | N Pacific | A | 83 | | 83 | 38 |
| Wu et al. (2021) | Bay of Bengal | A | 224 | | 224 | |
| Total | | | 2812 | 139 | 1203 | 859 |

[a] Data are reported by data providers as depth-integrated cell abundance (unlabelled depth-integrated abundances
computed from volumetric data).

**Table 6.** Summary of new data points of *nifH* gene copy abundances added to the version 2 of the database. UCYNs include UCYN-A1, UCYN-A2, UCYN-B and UCYN-C. Heterocystous cyanobacteria include Het-1, Het-2 and Het-3.

| Reference | Region | *Trichodesmium* | UCYN | Heterocystous cyanobacteria | Depth-integrated data |
|---|---|---|---|---|---|
| Benavides et al. (2016a) | N Atlantic | 13 | 30 | 15 | |
| Bentzon-Tilia et al. (2015b) | Baltic Sea | | 20 | | |
| Berthelot et al. (2017) | W Tropical Pacific | 64 | 256 | 64 | 96 |
| Bombar et al. (2011) | South China Sea | 18 | 36 | 18 | |
| Bombar et al. (2015) | N Pacific | | | | 32 |
| Bonnet et al. (2015) | SW Pacific | 87 | 261 | 87 | 84 |
| Bonnet et al. (2023) | SW Pacific | 66 | 132 | | 44 |
| Cabello et al. (2020) | Monterey Bay | | 200 | | |
| Confesor et al. (2022)[b] | W Florida Shelf | 67 | | | |
| Cerdan-Garcia et al. (2021) | N Atlantic | 7 | 7 | | |
| Chen et al. (2019) | W Pacific | 103 | 381 | 177 | 123 |
| Cheung et al. (2020) | N Pacific | 519 | 519 | | |
| Cheung et al. (2022) | W Bering Sea | | 58 | 29 | |
| Church and Zehr (2020) | N Pacific | 968 | 1936 | 1936 | 605 |
| Church et al. (2008) | N Pacific | | | | 60 |
| Detoni et al. (2022) | SW Atlantic | 70 | 140 | 70 | 72 |
| Dugenne et al. (2023); Gradoville et al. (2022) | N Pacific Subtropic Gyre | 72 | 216 | 216 | 112 |
| Foster et al. (unpublished data) | South China Sea | 99 | 224 | 350 | 158 |
| Gradoville et al. (2020) | N Pacific | 43 | 85 | 28 | |
| Hallstrøm et al. (2022) | NE Atlantic | | | | 42[a] |
| Halm et al. (2012) | S Pacific Gyre | 8 | 16 | | |
| Hamersley et al. (2011) | S California Bight | 6 | 12 | 6 | |
| Harding et al. (2018) | Arctic Ocean | | 39 | | |
| Hashimoto et al. (2016) | Seto Inland Sea | | 176 | | |
| Henke et al. (2018) | W Tropical S Pacific | | 142 | | |
| Krupke et al. (2013) | N Atlantic | | 24 | | 3 |

| Reference | Region | *Trichodesmium* | UCYN | Heterocystous cyanobacteria | Depth-integrated data |
|---|---|---|---|---|---|
| Liu et al. (2020) | South China Sea | 49 | 98 | | 33 |
| Lory et al. (2022) | W Tropical S Pacific | 3 | 3 | | |
| Lu et al. (2018) | W Tropical Pacific | 3 | 6 | 3 | |
| Martínez-Pérez et al. (2016) | N Tropical Atlantic | 84 | 252 | 84 | 70 |
| Messer et al. (2021) | S Australian Gulf | | 20 | | |
| Mills et al. (2020) | Coast of S California | 4 | 12 | 4 | |
| Moisander et al. (2014) | S Pacific | 174 | 348 | 174 | 92 |
| Moore et al. (2018) | Tropical Atlantic | 104 | 312 | 208 | |
| Moreira-Coello et al. (2017) | the coastal NW Iberian upwelling | | 20 | | 20[a] |
| Palter et al. (2020) | Gulf stream | 24 | 24 | | |
| Ratten et al. (2015) | N Atlantic | 9 | 27 | 9 | 10 |
| Reeder et al. (2022) | Baltic Sea | | 15 | 15 | |
| Sato et al. (2021) | Subarctic Sea | | 31 | | 3 |
| Sato et al. (2022) | Eastern Indian Ocean | 73 | 73 | | 36 |
| Saulia et al. (2020) | SW Pacific | 71 | 213 | 143 | |
| Scavotto et al. (2015) | | | 2 | | |
| Selden et al. (2021b) | Atlantic Bight | 23 | 69 | 23 | |
| Selden et al. (2022) | Arctic Ocean | | 40 | | |
| Shiozaki et al. (2014b) | Arabian Sea | 26 | 52 | | 18 |
| Shiozaki et al. (2014c) | S China Sea | 171 | 342 | | 72[a] |
| Shiozaki et al. (2015a) | Temperate N Pacific | 73 | 146 | | 33 |
| Shiozaki et al. (2017) | N Pacific | 74 | 222 | 74 | 90 |
| Shiozaki et al. (2018a) | Kuroshio | 46 | 138 | 46 | |
| Shiozaki et al. (2018b) | W Arctic | | 84 | | 21 |
| Shiozaki et al. (2018c) | S Pacific | 94 | 285 | 95 | 95 |
| Shiozaki et al. (2020) | Antarctic sea ice | | 53 | | |
| Sohm et al. (2011) | S Atlantic Gyre | | 58 | | |
| Stenegren et al. (2017) | W Tropical N Atlantic | | | 235 | 61 |

| Reference | Region | Trichodesmium | UCYN | Heterocystous cyanobacteria | Depth-integrated data |
|---|---|---|---|---|---|
| Stenegren et al. (2018) | W Tropical S Pacific | 108 | 402 | 120 | 108 |
| Tang et al. (2020) | N Atlantic | 42 | 42 | | |
| Turk-Kubo et al. (2014) | E Tropical S Pacific | 60 | 159 | 57 | 53 |
| Turk-Kubo et al. (2021) | Coast of S California | 190 | 588 | 202 | 135 |
| Wen et al. (2017) | W Pacific | 22 | 44 | 22 | |
| Wen et al. (2022) | W Pacific | 130 | 390 | 130 | 110[a] |
| White et al. (2018) | N Pacific | | | | 34 |
| Wu et al. (2019) | Bay of Bengal | 68 | 63 | | 19 |
| Total | | 3935 | 9543 | 4640 | 2544 |

[a] Data are reported by data providers as depth-integrated *nifH* gene copy abundances (unlabelled depth-integrated abundances computed from volumetric data).

[b] *rnpB* gene copies were determined.**Table 7.** Summary of data points of cell-specific $N_2$ fixation rates added to the version 2 of the database. The rates were measured either by using the combination of CARD-FISH and nanoSIMS (method A), or via the measurements of bulk $N_2$ fixation rates incubated with known number of diazotrophic cells (method B) (*see* **Section 2.3**). Note that all the data were reported as $N_2$ fixation rates per cell, except for Filella et al. (2022) in which biomass-normalized rates in unit of $d^{-1}$ were reported.

| Reference | Region | Method | Trichode-smium | UCYN-A | UCYN-A1 | UCYN-A2 | UCYN-B | Riche-lia | Calo-thrix | Unclassified Cyano-bacteria | NCDs |
|---|---|---|---|---|---|---|---|---|---|---|---|
| Benavides et al. (2022a) | Tropical SW Pacific | A | 6 | | | | | | | | |
| Benavides et al. (2017) | SW Pacific | A | 2 | | | | | | | | |
| Bonnet et al. (2018) | Tropical SW Pacific | A | 3 | | | | 2 | | | | |
| Filella et al. (2022) | S Pacific Gyre | A | 12 | | | | 12 | | | | |
| Foster et al. (2011) | N Pacific | A | | | | | 2 | 18 | 2 | | |
| Foster et al. (2013) | N Pacific | A | | | | | 6 | | | | |

| Reference | Region | Method | *Trichode-smium* | UCYN-A | UCYN-A1 | UCYN-A2 | UCYN-B | *Riche-lia* | *Calo-thrix* | Unclassified Cyano-bacteria | NCDs |
|---|---|---|---|---|---|---|---|---|---|---|---|
| Foster et al. (2022a) | Tropical NW Atlantic | A | | | | | | 39 | | | |
| Gradoville et al. (2020) | N Pacific | A | | | 5 | | | | | | |
| Gradoville et al. (2021) | N Pacific | A | | | 17 | | | | | | |
| Harding et al. (2018) | Arctic Ocean | A | | | | 2 | | | | | |
| Harding et al. (2022) | Subtropical N Pacific | A | | | | | | | | 40 | 34 |
| Krupke et al. (2013) | Subtropical N Atlantic | A | | 4 | | | 2 | | | | |
| Krupke et al. (2015) | Subtropical NE Atlantic | A | | 1 | | | | | | | |
| Martínez-Pérez et al. (2016) | Tropical N Atlantic | A | 101 | | 57 | 10 | | | | | |
| Mccarthy and Carpenter (1979) | N Atlantic | B | 24 | | | | | | | | |
| Mills et al. (2020) | California Current System | A | | | 15 | 9 | | | | | |
| (Turk-Kubo et al., 2021) | Southern California Current System | A | | | 26 | 17 | | | | | |
| Total | | | 148 | 10 | 115 | 38 | 24 | 57 | 2 | 40 | 34 |

## 2.2 Quality control

The data of N₂ fixation rates and diazotrophic abundance in the database spanned over several orders of magnitude. Extremely high rate and abundance values of both usually occurred during algal blooms, and zero values indicated that diazotrophic activity was below detection or truly absent at the sampling time and stations. The positive-value data were first logarithmically transformed and then analyzed for outliers, considering that they were approximately log-normally distributed (**Fig. S1-S5**). For each parameter, we used Chauvenet's criterion to identify suspicious outliers whose probability of deviation from the

means is lower than $1/2n$, where $n$ is the number of data points (Glover et al., 2011). Because $N_2$ fixation rates and diazotroph abundances in the ocean can be extremely low, this filtering only applied to data on the high side. Although these outliers (labelled in the database) could be true values, we flagged them to remind users for caution.

## 2.3 Nitrogen fixation rate data

The commonly used methods for marine $N_2$ fixation rates include $^{15}N_2$ tracer methods and acetylene reduction assay (Mohr et al., 2010; Montoya et al., 1996; Capone, 1993). However, in the last decade, the community has turned largely to the use of $^{15}N_2$ tracer methods. The acetylene reduction assay estimates gross $N_2$ fixation rates indirectly from the reduction of acetylene to ethylene. Theoretical conversion factors of 3:1 or 4:1 have been used to convert acetylene reduction rates to $N_2$ fixation rates (Postgate, 1998; Capone, 1993; Wilson et al., 2012), although a wide range of conversion factors from 0.93 to 56 have been reported (e.g., Mague et al., 1974; Graham et al., 1980; Montoya et al., 1996; Capone et al., 2005; Mulholland et al., 2006; Wilson et al., 2012). When using the $^{15}N_2$ tracer method, samples are incubated in seawater with $^{15}N_2$ gas; the $^{15}N/^{14}N$ ratio of particulate nitrogen is measured at the beginning and at end of the incubation to calculate the $N_2$ fixation rate (Capone and Montoya, 2001). Most measurements using the $^{15}N_2$ tracer method only counted the fixed N in particulate forms and ignored the N that was fixed but then excreted by diazotrophs in form of dissolved organic N (DON) during incubation, which could theoretically be counted by the acetylene reduction assays (Mulholland, 2007). In some studies using the $^{15}N_2$ tracer method, this missing N was counted by also measuring the $^{15}N$ enrichment in DON (Berthelot et al., 2017; Benavides et al., 2013a; Berthelot et al., 2015; Benavides et al., 2013b).

Compared to the $^{15}N_2$ tracer method, the acetylene reduction assay needs a shorter incubation time. However, in addition to the uncertainty in converting ethylene production to $N_2$ fixation, the purity of acetylene gas, trace ethylene contamination, and the Bunsen gas solubility coefficient of produced ethylene can also affect the accuracy of estimated $N_2$ fixation rates (Hyman and Arp, 1987; Breitbarth et al., 2004; Kitajima et al., 2009). Acetylene used in the assay can even impact the metabolic activities of diazotrophs (Giller, 1987; Hardy et al., 1973; Flett et al., 1976; Staal et al., 2001). Moreover, the acetylene reduction assay needs to pre-concentrate cells for signal detection when diazotrophic biomass is low, which may lead to underestimated $N_2$ fixation rates by perturbing cells during concentration and filtration (e.g., Capone et al., 2005; Barthel et al., 1989; Staal et al., 2007). In recent years, the acetylene reduction assay has undergone significant advancements. The sensitivity of ethylene detection has been improved by utilizing a reduced gas analyzer (Wilson et al., 2012) and by using highly purified acetylene gas to minimize the ethylene background (Kitajima et al., 2009). However, the preparation of high-purity acetylene with low level of ethylene contamination remains a challenge. More recently, a new method named Flow-through incubation Acetylene Reduction Assays by Cavity ring-down laser Absorption Spectroscopy (FARACAS) has been introduced for high-frequency measurements of aquatic $N_2$ fixation (Cassar et al., 2018). This method involves continuous flow-through incubations and spectral monitoring of the acetylene reduction to ethylene. By employing short-duration flow-

through incubations without cell preconcentration, potential artifacts are minimized. This approach also allows for near real-time estimates, enabling adaptive sampling strategies.

The original $^{15}N_2$ tracer method involved addition of a known volume of $^{15}N_2$-labelled bubbles to the incubation bottle (named *original $^{15}N_2$ bubble method* hereafter). However, this method was later found to underestimate rates because $N_2$ gas solubility is low and tracer additions take a long time to equilibrate (Mohr et al., 2010; Großkopf et al., 2012; Jayakumar et al., 2017). To address this issue, the *$^{15}N_2$ dissolution method* has been employed, which involves pre-preparing $^{15}N_2$-enriched seawater to maintain a constant $^{15}N_2$ atom% enrichment throughout the incubation (Mohr et al., 2010), similar to the method described in Glibert and Bronk (1994). However, the $^{15}N_2$ dissolution method does not always yield higher $N_2$ fixation rates than the original $^{15}N_2$ bubble method (Table S4 in Großkopf et al., 2012; Saulia et al., 2020); it is still not conclusive what control the magnitude of the underestimation (if it exists) by the original $^{15}N_2$ bubble method. Compared to the original $^{15}N_2$ bubble method, the $^{15}N_2$ dissolution method is more susceptible to the introduction of contaminants (e.g., metals) during the preparation of the $^{15}N_2$ inoculum due to its more complex process, which can alter the diazotrophic activities and abundance, thereby impacting the accuracy of $N_2$ fixation measurements (Dabundo et al., 2014; Klawonn et al., 2015). For example, Needoba et al. (2007) reported that a low but detectable amount of $Fe^{3+}$ contamination can be measured when protecting the needle of the gas-tight syringe with a commercially available tubing. Additionally, pH and other chemical properties of the inoculum may be altered during its preparation, further affecting the measurements of $N_2$ fixation. Despite these limitations, the $^{15}N_2$ dissolution method remains the predominant assay for measuring $N_2$ fixation rate due to its ability to satisfy the fundamental assumption of constant $^{15}N_2$ atom% enrichment over the incubation period.

More recently, a modified $^{15}N_2$ bubble method, known as the *$^{15}N_2$ bubble release method*, has been proposed as an alternative to the $^{15}N_2$ dissolution method (Klawonn et al., 2015; Chang et al., 2019; Selden et al., 2019). This method involves adding $^{15}N_2$ gas to the incubation bottles and mixing for a brief period (~15 min) to facilitate $^{15}N_2$ equilibration, then removing the gas bubble. Compared to the original $^{15}N_2$ bubble method, the $^{15}N_2$ bubble release method ensures a uniform $^{15}N_2$ atom% enrichment throughout the incubation. Moreover, it causes less interference with the incubation matrix than the $^{15}N_2$ dissolution method. However, the mixing of incubation bottles required to stimulate gas dissolution has been suggested to negatively affect diazotrophs, although no robust studies have yet been performed to assess this criticism (Wannicke et al., 2018; White et al., 2020). Moreover, the $^{15}N_2$ bubble release method requires a handling step and additional costs for preparing tracers may be another challenge for researchers (White et al., 2020). Ultimately White et al. (2020) "advise employing either the dissolution or bubble release method, whichever is best suited to the specific research objectives and logistical constraints" with additional recommendations on the need for determination of detection limits for all rate measurements.

We compared volumetric $N_2$ fixation rates in the upper 50 m and depth-integrated $N_2$ fixation rates in the database measured using the acetylene reduction assays, the original $^{15}N_2$ bubble method and the $^{15}N_2$ dissolution method, and found that they span a similar range (**Fig. 1**). Meanwhile, in the analysis for volumetric $N_2$ fixation rates in upper 50 m, the peak of the log-normal distributions of the measurements using the $^{15}N_2$ dissolution method was approximately double that of the original $^{15}N_2$ bubble method (**Fig. 1a**). The measurements using the $^{15}N_2$ bubble release method were limited to several study

sites and their distribution was thus not presented in this study. A further analysis comparing the original $^{15}N_2$ bubble method

and the $^{15}N_2$ dissolution method will be presented later.

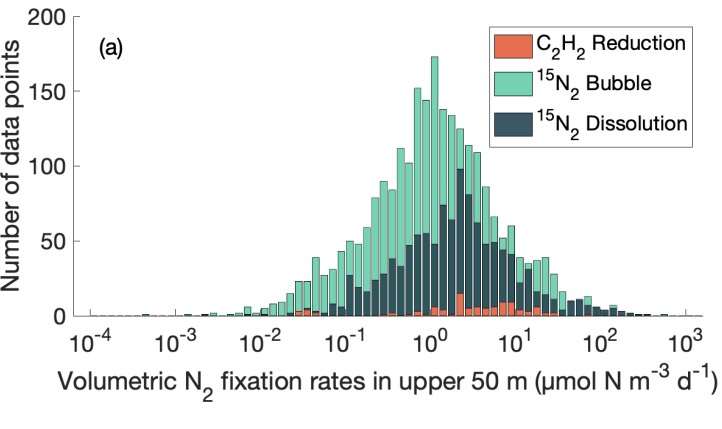

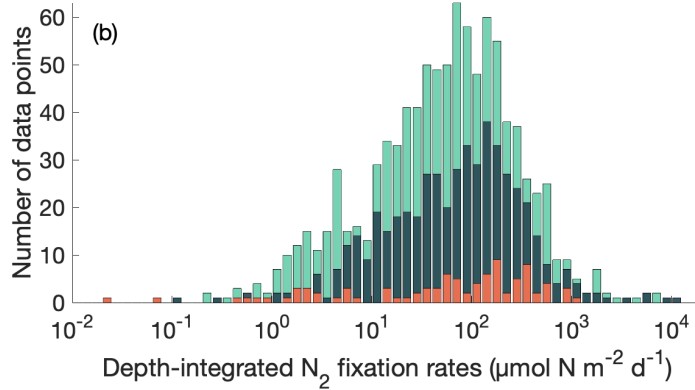

**Figure 1.** Distribution of $N_2$ fixation rates measured using the acetylene ($C_2H_2$) reduction assays, the original $^{15}N_2$ bubble method and the $^{15}N_2$ dissolution method. (a) Volumetric data in upper 50 m; (b) depth-integrated data. Only rates measured with incubation periods of 24 hours are shown. Please note that the bars in plot do not represent cumulative data.


The majority of $N_2$ fixation rates (9,405) were measured with incubation periods of 24 hours and were reported as daily rates. In contrast, 2,416 samples were incubated for less than 24 hours and hourly $N_2$ fixation rates were reported. Diel cycles of $N_2$ fixation vary among samples and/or diazotrophic groups, and substantial errors may be introduced when extrapolating $N_2$ fixation rates incubated for less than 24 hours to daily rates (White et al., 2020). Therefore, the $N_2$ fixation rates measured

with incubation periods of less than 24 hours were collected into separated datasheets in our database and were not used in further analyses within this study. Please note that the incubation periods of whole diurnal cycles (e.g., 24, 48, or 72 hours) were used in Konno et al. (2010). The samples in Yogev et al. (2011) were incubated between 24 to 30 hours. The reported

daily $N_2$ fixation rates by these two studies were also included in the 24-hour datasheets and were used in our estimation of the global marine $N_2$ fixation rate (see below).

Cell-specific $N_2$ fixation rates of diazotrophs (or symbioses) were mostly measured using catalyzed reporter deposition fluorescence in-situ hybridization (CARD-FISH) and nanoscale secondary ion mass spectrometry (nanoSIMS), in combination with $^{15}N_2$ addition experiments (Mills et al., 2020; Berthelot et al., 2019). Using specific oligonucleotide probes, CARD-FISH enables the visualization and location of the regions of interest in diazotrophs at a single-cell level under epifluorescence microscope. This is subsequently prepared for the secondary electron image in nanoSIMS analysis. Importantly, the handling,

fixation and processing of the samples with CARD-FISH, has been demonstrated to significantly impact the enrichment measured by nanoSIMS (see Musat et al. 2014; Woebken et al. 2015; Meyer et al. 2021). The nanoSIMS technique detects the enrichment of $^{15}N$ atoms in the targeted regions, allowing for the calculation of the cell-specific rate. Additionally, in one study, handpicked *Trichodesmium* colonies or trichomes were incubated, and the measured total $N_2$ fixation rates were normalized to number of cells (Mccarthy and Carpenter, 1979).

**2.4 Estimation of the global marine $N_2$ fixation rate**

      Using these data, we performed a first-order estimation of the global marine $N_2$ fixation rate. In a previous study (Luo et al., 2012), version 1 was utilized to estimate the global marine $N_2$ fixation rate, which included all the depth-integrated $N_2$ fixation rates. However, in this study, we employed more rigorous criteria to estimate the global rate using both version 1 and version 2, taking into account the reliability of different $N_2$ fixation rate data discussed in the preceding section. Specifically,

we exclusively used depth-integrated $N_2$ fixation rates that met the following criteria: (1) measurements were taken from whole seawater samples, (2) incubation periods of 24 hours were used, and (3) the three $^{15}N_2$-based methods were employed, although we acknowledged that the rates obtained using the original $^{15}N_2$ bubble method might be underestimated. $N_2$ fixation rates obtained through the acetylene reduction method were excluded from this estimate due to the significant uncertainties described above.

Applying these criteria, we selected 309 and 1,642 depth-integrated $N_2$ fixation rates from version 1 and version 2, respectively. The much more data in version 2 potentially provided more constraints on estimating global marine $N_2$ fixation. We applied Chauvenet's criterion to identify outliers, using the log-transformed values of the selected data (*see* **Section 2.2**). As a result, two high-value outliers were removed in version 1 (one in North Pacific and one in South Pacific) while no outliers were detected in version 2. This difference can be attributed to the larger number of data samples in version 2, which allowed

for a more relaxed threshold in identifying outliers.

      The estimation of the global marine $N_2$ fixation rate involved four steps. First, we calculated the arithmetic or geometric means of depth-integrated $N_2$ fixation rates within each 3° latitude × 3° longitude bin. Second, these mean values were further averaged using either arithmetic or geometric methods to determine the mean $N_2$ fixation rates for different ocean basins, which included the North Atlantic, South Atlantic, North Pacific, South Pacific, Indian, Arctic, Southern Oceans, and the

Mediterranean Sea. Third, we multiplied the arithmetic or geometric mean of each basin by its respective area to estimate the total $N_2$ fixation rate for that specific basin, except when there was insufficient spatial coverage available. Finally, we obtained the global marine $N_2$ fixation rate by summing up the individual rates calculated for each basin, with the errors associated with basin rates propagated properly (Glover et al., 2011).

In the first two steps, the geometric means were derived from positive $N_2$ fixation rates ($NF_+$): if $\mu$ and $SE$ represented the mean and standard error of $\ln(NF_+)$, respectively, the geometric mean was $e^\mu$. The confidence interval for the geometric mean, based on the standard error, ranged between $e^\mu/e^{SE}$ and $e^\mu \cdot e^{SE}$ (Thomas, 1979). To address the issue of not including zero-value $N_2$ fixation rates, we adjusted the geometric means by multiplying them with the percentage of zero-value data within each $3° \times 3°$ bin (in the first step) or within each basin (in the second step).

## 2.5 Diazotrophic abundance data

Diazotroph cell abundances were determined by using standard light microscopy, and in some cases by using epifluorescence microscopy. A recent study used machine learning techniques to detect and enumerate diazotrophs in a large dataset of microscopic images (Karlusich et al., 2021). In the original database, only the cell abundances of *Trichodesmium* and heterocystous cyanobacteria were recorded. Version 2 also included datasets of enumerated abundance of all UCYN groups detecting them by TSA (Tyramid Signal Amplification)-FISH using the specific DNA probe UCYN-238 (Biegala and Raimbault, 2008; Le Moal and Biegala 2009; Le Moal et al., 2011; Riou et al. 2016). This method is also called CARD-FISH and was used to specifically enumerated UCYN-A (Martínez-Pérez et al., 2016; Biegala and Raimbault, 2008; Le Moal et al., 2011). (**Table 5**).

Cell abundance of *Trichodesmium* was recorded as the number of trichomes per volume of water in our database, although it was also reported in some studies as the number of cells or colonies per volume of water. In the latter cases, the data were converted to trichomes per volume of water by using a commonly used factor of 200 (132–241) trichomes colony[-1] (Letelier and Karl, 1996), similar to the conversion used in the original database (Luo et al., 2012).

The abundance of heterocystous cyanobacterial cells was also recorded in this database. When the number of DDAs was reported in several studies, we assumed that 2 (reported range: 1–2) and 5 (reported range: 1–5) *Richelia* spp. filaments associated with each *Hemiaulus* and *Rhizosolenia* cell, respectively (Villareal et al., 2011; Caputo et al., 2019), and 5 (reported range: 3–10) *Richelia rhizosoleniae* filaments were associated with each *Chaetoceros* cell (Tuo et al., 2021; Caputo et al., 2019). *Richelia* have terminal heterocysts, and the number of vegetative cells varies depending on the host diatom. In *Hemiaulus* and *Chaetoceros* spp. diatoms, *Richelia* filaments are shorter (e.g., 3-4 vegetative cells), compared to in *Rhizosolenia*, the *Richelia* filaments are longer (e.g., 5-6 vegetative cells) (Foster et al., 2022b).

In measurements of *nifH* gene copy abundances, different qPCR or ddPCR assays were designed to target specific diazotrophic groups (Church et al., 2005a; Foster et al., 2007; Gradoville et al., 2017; Benavides et al., 2016a), mainly including *Trichodesmium*, UCYN subgroups (A1, A2, B, and C) and heterocystous groups (het-1, het-2, and het-3) (**Table 6**).

    All the uncertainties reported in this paper reflect one standard error of the means unless specified otherwise.

## 3. Results

**3.1 Data distribution**

The version 2 of the database significantly expanded $N_2$ fixation rate measurements, filling spatial gaps particularly in the Indian Ocean and the Southern Hemisphere (**Table 1; Figs. 2a–b, 3a–b**). The number of depth-integrated $N_2$ fixation rate measurements was tripled (**Table 1; Figs. 2b & 3b**). The largest fraction of new data derived from inclusion of *nifH* gene abundances, in particular data contributions from the Pacific and Atlantic Oceans (**Table 3; Figs. 2e–f, 3e–f**). Compared to

other parameters, the new database contained only a modest increase in new cell abundances, mostly from the subtropical oceans (**Table 2; Figs. 2c–d, 3c–d**). Overall, there remained more limited data on $N_2$ fixation and diazotrophic abundance in the Arctic and Southern Oceans, with a number of rate measurements reporting values below detection limits.

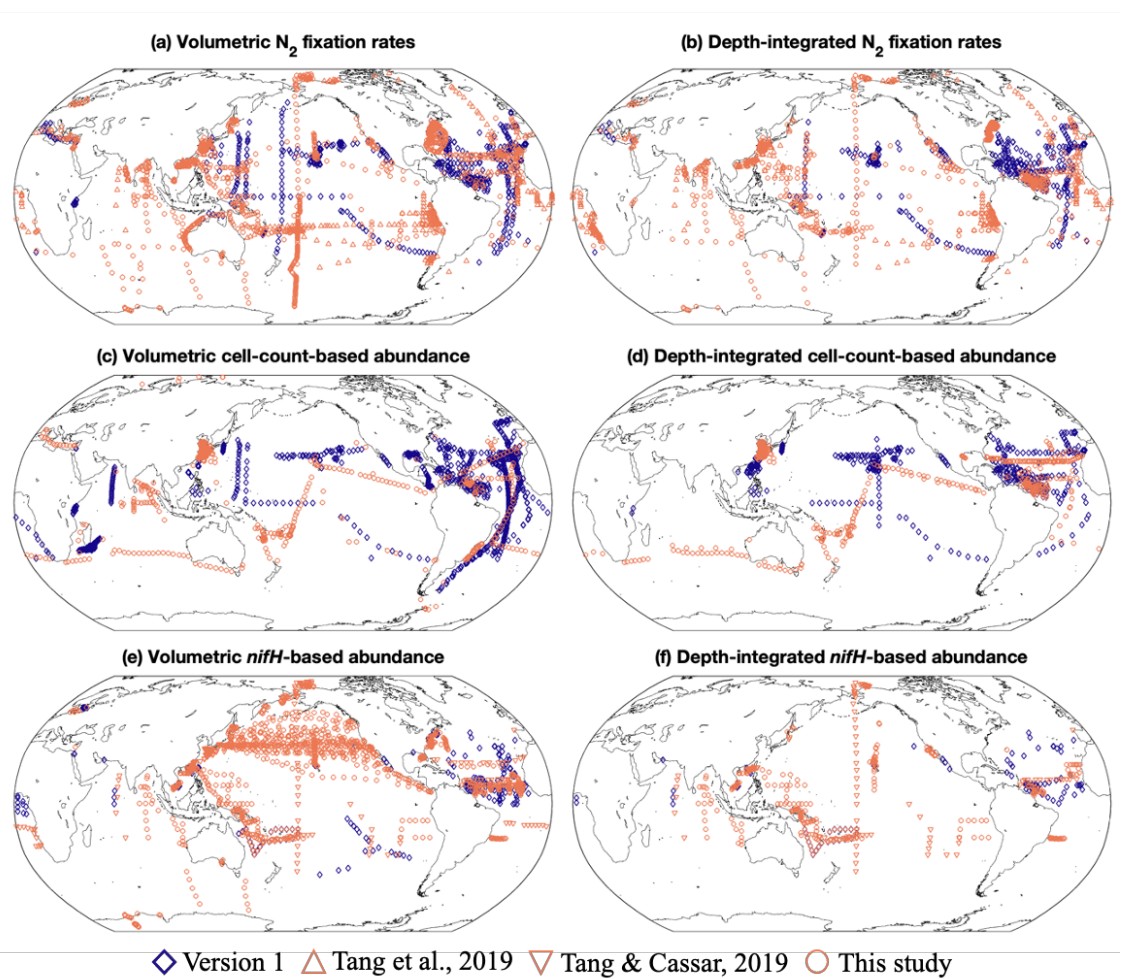

**Figure 2.** Spatial distribution of volumetric and depth-integrated number of data point in the version 2 of the diazotrophic database binned in 1° latitude × 1° longitude grids. (**a-b**) $N_2$ fixation rates, (**c-d**) cell abundance, and (**e-f**) *nifH* gene copy abundance. The data sources include the original version of this database (Luo et al., 2012) (blue diamonds), two compiled datasets (Tang et al., 2019; Tang and Cassar, 2019) (orange triangles) and this study (orange circles).

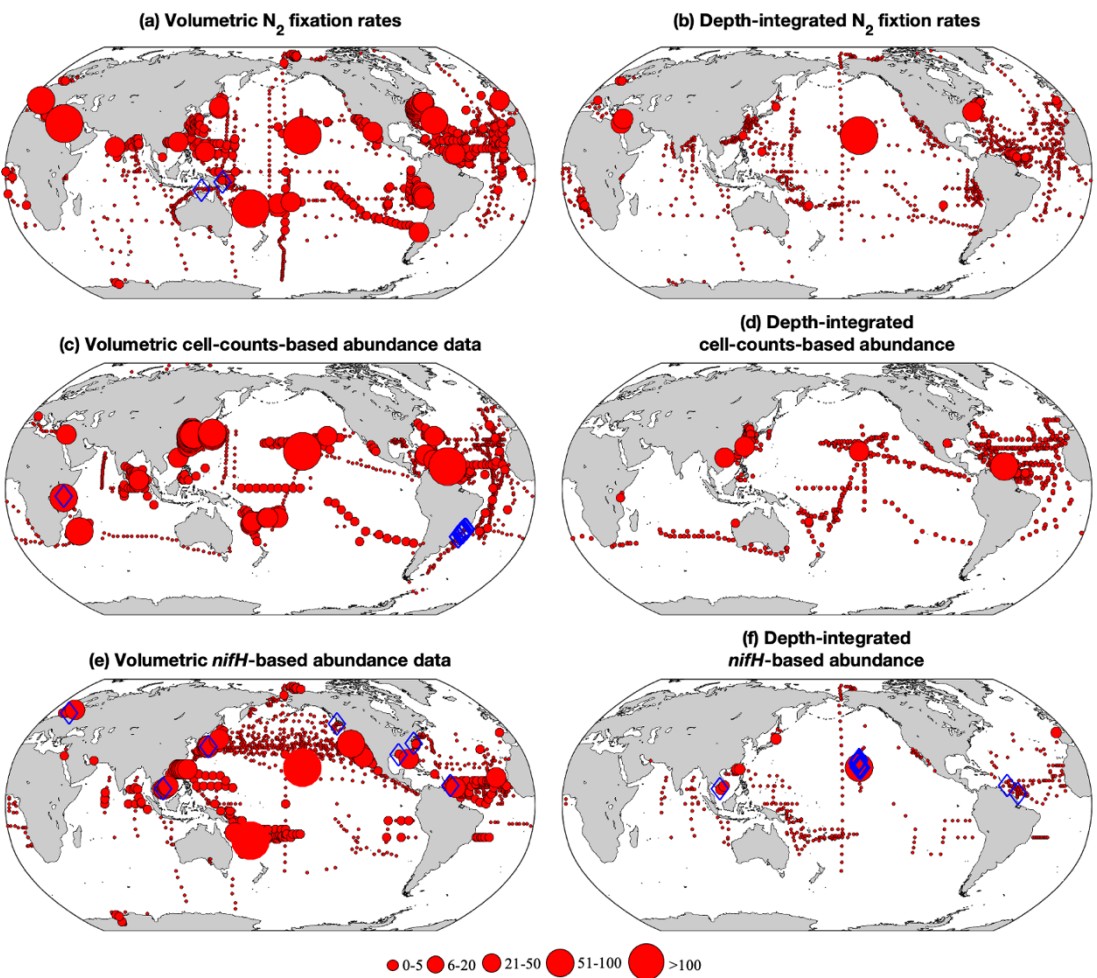

**Figure 3.** Spatial distribution of volumetric and depth-integrated number of data point binned in 1° latitude × 1° longitude grids for (**a-b**) N$_2$ fixation rates, (**c-d**) cell abundance, and (**e-f**) *nifH* gene copy abundance. The size of the circles represents the number of data points in each bin. The blue diamonds mark the location of outliers identified using Chauvenet's criterion.

Version 2 added data at all latitudinal ranges (**Fig. 4**). In particular, version 2 extended the range of data from tropical and subtropical areas to include polar regions in the Arctic Ocean (Harding et al., 2018) and Antarctic coast (Shiozaki et al., 2020).

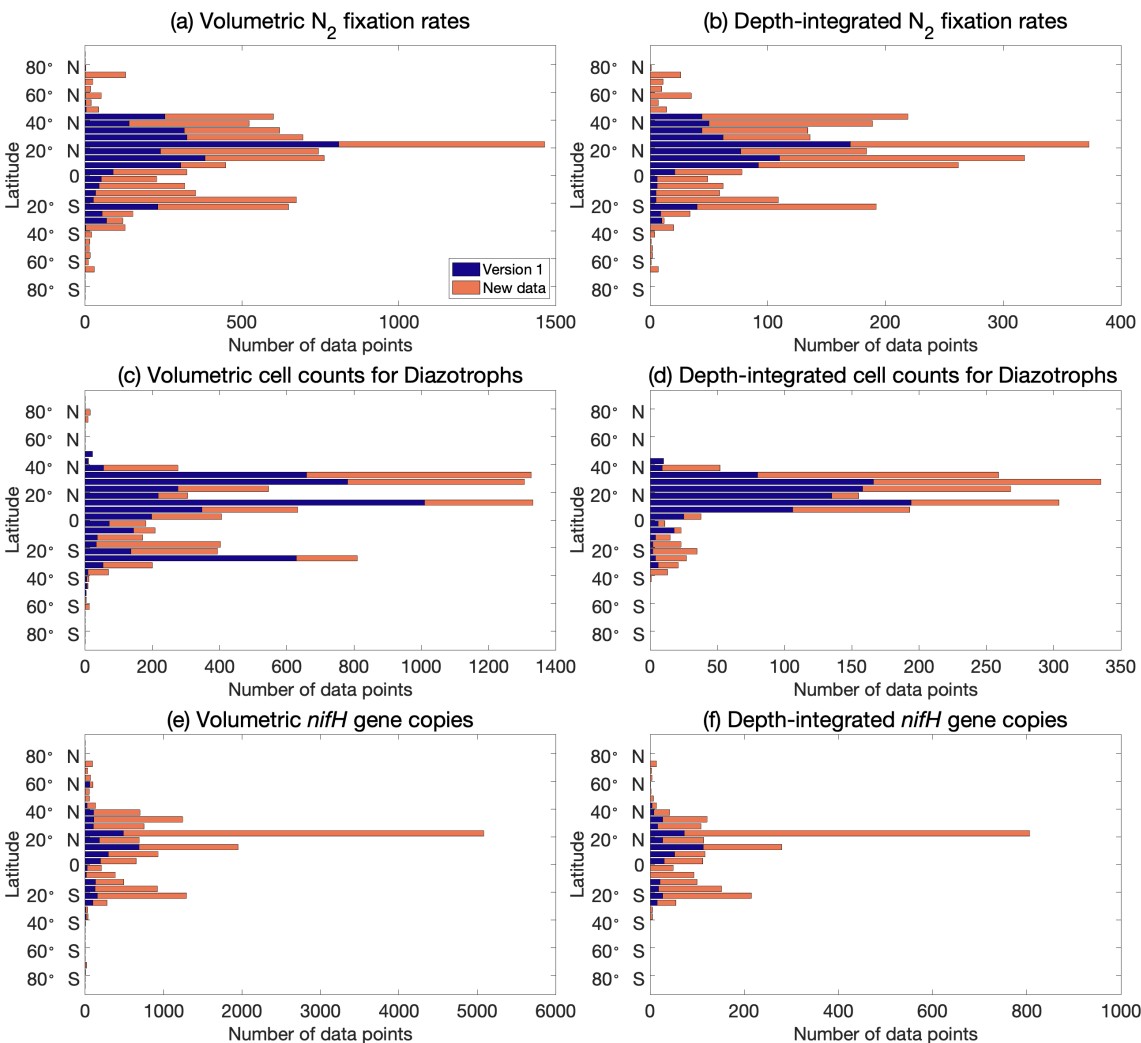

**Figure 4.** Latitudinal distribution of volumetric and depth-integrated (**a-b**) N$_2$ fixation rates, (**c-d**) cell abundance, and (**e-f**) *nifH* gene copy abundance, including the data in the version 1 of the database (blue) and the new data added to the version 2 of the database (orange).


The data in version 2 reduce the difference in number of data points across months, especially for *nifH* gene copies, in which substantially more samples were collected in January and February (**Fig. 5**). When considering seasons in both the Northern hemisphere and South Atlantic and Pacific the data were distributed more evenly (**Fig. 6**).

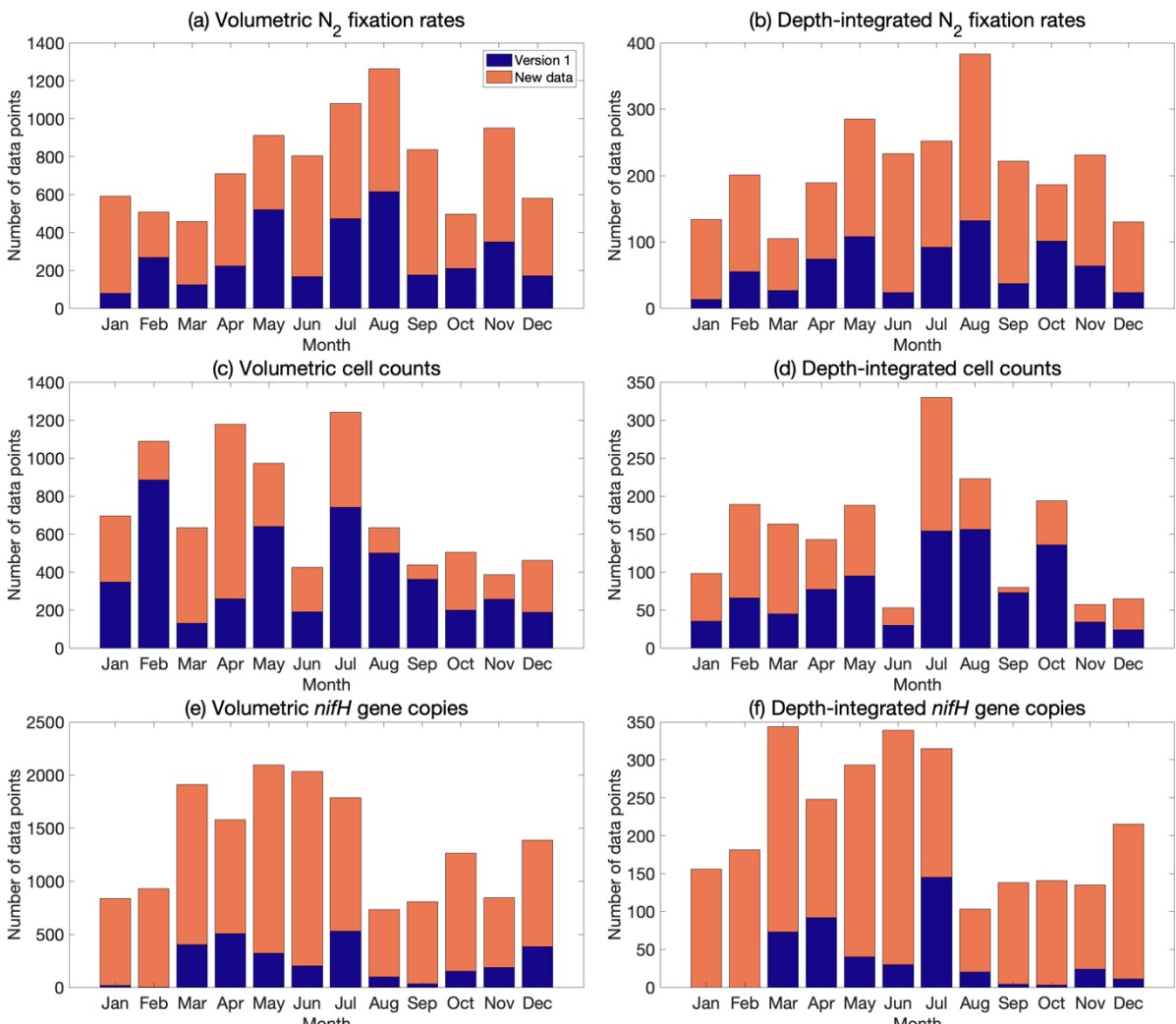

**Figure 5.** Monthly distribution of volumetric and depth-integrated (**a-b**) N₂ fixation rates, (**c-d**) cell abundance, and (**e-f**) *nifH* gene copy abundance, including the data in the version 1 of the database (blue) and the new data added to the version 2 of the database (orange).

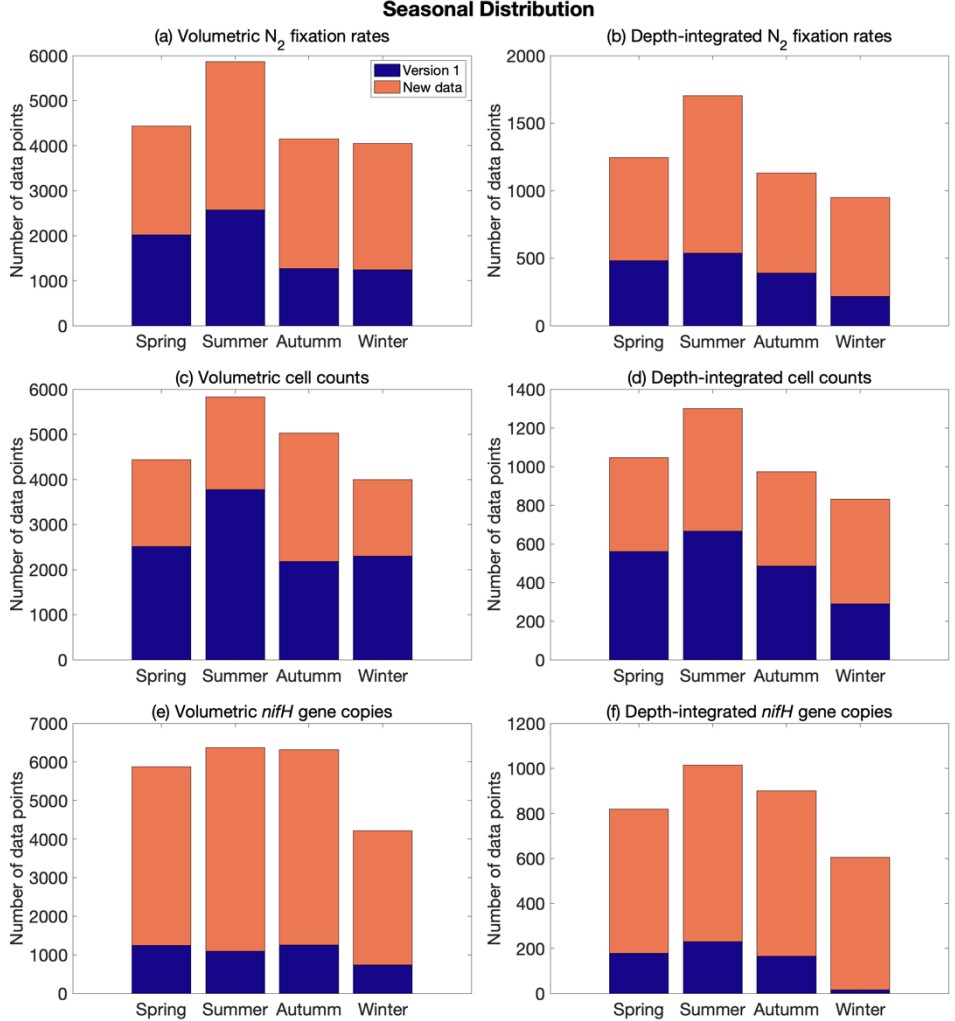

**Figure 6.** Seasonal distribution of volumetric and depth-integrated (**a-b**) $N_2$ fixation rates, (**c-d**) cell abundance, and (**e-f**) *nifH* gene copy abundance, including the data in the version 1 of the database (blue) and the new data added to the version 2 of the database (orange). Spring: March–May in the Northern Hemisphere and September–November in the Southern Hemisphere; summer: June–August in the Northern Hemisphere and December–February in the Southern Hemisphere; autumn: September–November in the Northern Hemisphere and March–May in the Southern Hemisphere; and winter: December–February in the Northern Hemisphere and June–August in the Southern Hemisphere.

Although most of the new data were measured in the near-surface waters, numerous *nifH* gene copy abundance data were also sampled in deeper layers in the euphotic zone (**Fig. 7**). Additionally, active $N_2$ fixation and the existence of diazotrophs were found below the euphotic zone (e.g., depth > 200 m) (Benavides et al., 2016b; Benavides et al., 2018b; Selden et al., 2019; Hamersley et al., 2011; Bonnet et al., 2013; Loescher et al., 2014; Benavides et al., 2015) (**Fig. 7**).

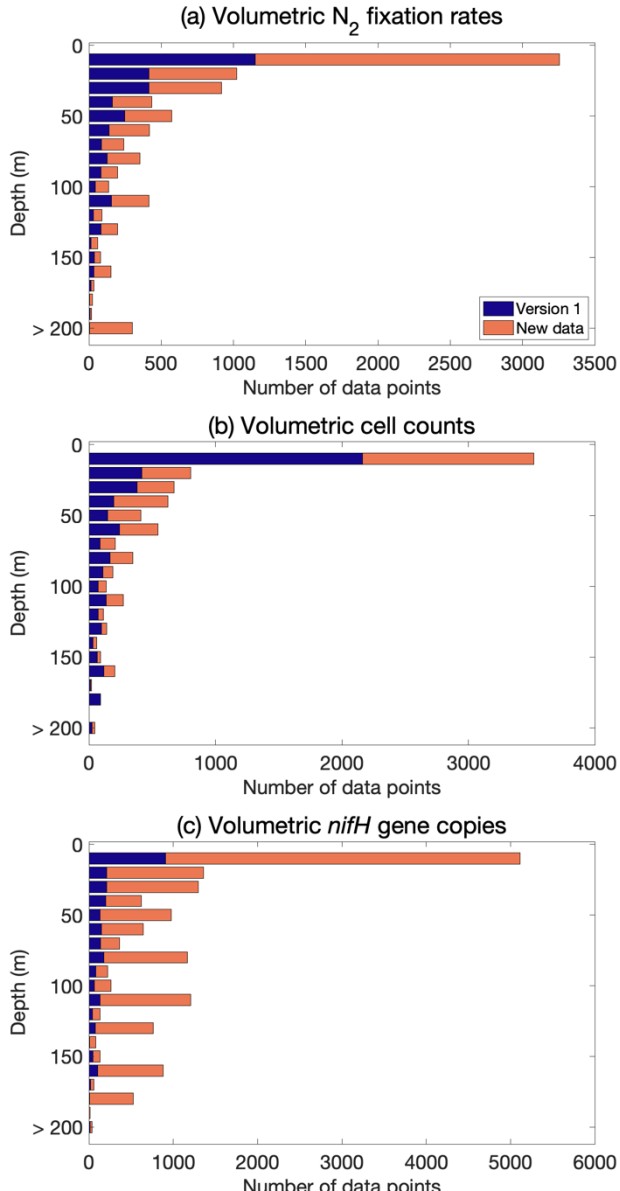

**Figure 7.** Vertical distribution number of (**a**) $N_2$ fixation rates, (**b**) cell abundance, and (**c**) *nifH* gene copy abundance data, including the data in the version 1 of the database (blue) and the new data added to the version 2 of the database (orange).

### 3.2 N₂ fixation rates

The volumetric $N_2$ fixation rates in 5 vertical layers and the depth-integrated $N_2$ fixation rates were binned in 3° latitude × 3° longitude bins, and the arithmetic means in each bin are displayed (**Fig. 8**). The depth-integrated $N_2$ fixation rates ranged orders of magnitude, from $10^{-4} – 10^3$ µmol N m⁻² d⁻¹ (mostly from 1 to $10^2$ µmol N m⁻² d⁻¹) (**Fig. 8a**). Some high rates (i.e., $10^2 – 10^3$

μmol N m$^{-2}$ d$^{-1}$) were found in the western Pacific Ocean, the regions near the Hawaiian Islands, and the western tropical Atlantic Ocean. Approximately 10% of the depth-integrated N$_2$ fixation rates were < 1 μmol N m$^{-2}$ d$^{-1}$, and were mainly from the North Atlantic and Indian Oceans. Within the water column, the N$_2$ fixation rates were highest in the upper 25 m (**Fig. 8b, c**), below which the rates rapidly decreased with depth (**Fig. 8d, e, f**). In the upper 25 m, volumetric N$_2$ fixation rates in the southwestern Pacific were higher than those in other areas, mostly ranging from 1 to 100 μmol N m$^{-3}$ d$^{-1}$. Undetectable N$_2$ fixation rates were reported mostly in subpolar regions, as well as in certain tropical and subtropical regions (**Fig. 8**).

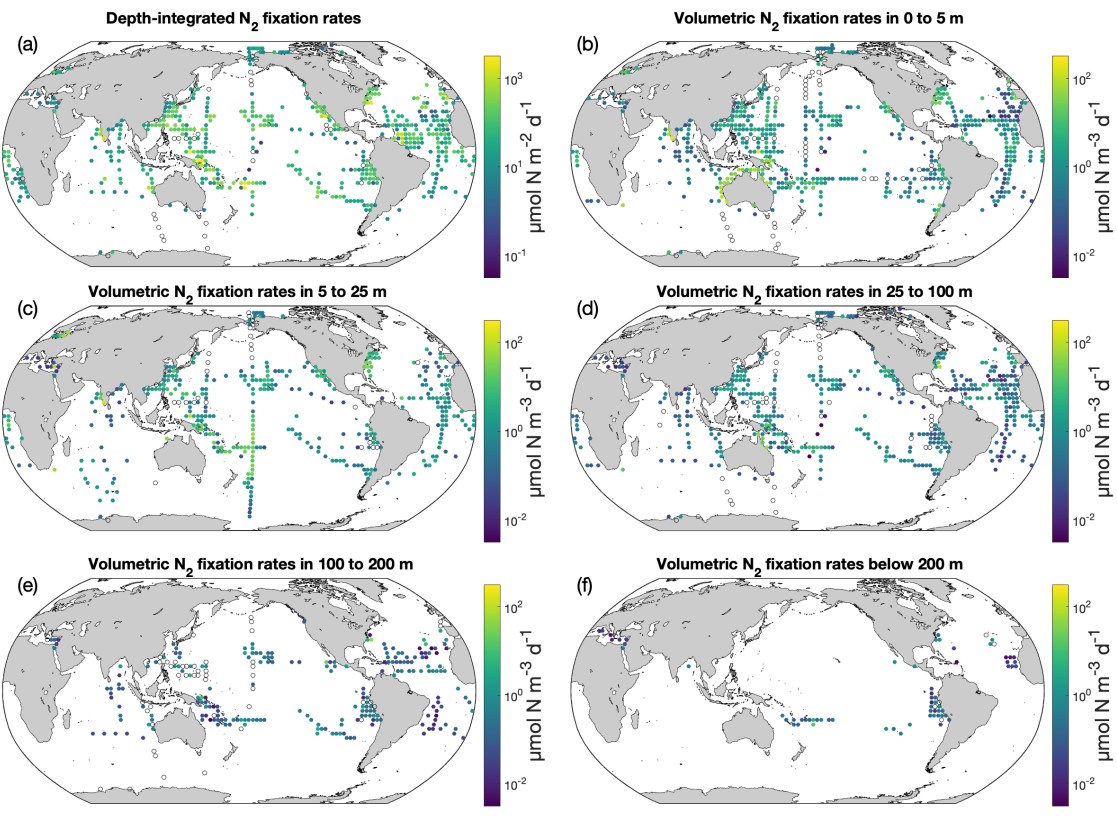

**Figure 8.** N$_2$ fixation rates in the version 2 of the database. The panels show (**a**) depth-integrated data and volumetric data in (**b**) 0–5 m, (**c**) 5–25 m, (**d**) 25–100 m, (**e**) 100-200 m, and (**f**) below 200 m. For a clear demonstration, arithmetic mean N$_2$ fixation rates in 3° latitude × 3° longitude bins are shown. Zero-value data are denoted as black empty circles. Only rates measured with incubation periods of 24 hours are included.

Cell-specific N$_2$ fixation rates span a range from 10$^{-4}$ to 10$^{3}$ fmol N cell$^{-1}$ d$^{-1}$, although mostly on the order of 10$^{-2}$ to 10$^{2}$ fmol N cell$^{-1}$ d$^{-1}$ (**Fig. 9**). The mean cell-specific N$_2$ fixation rates of *Trichodesmium*, UCYN-A2 and heterocystous cyanobacteria were one to two orders of magnitude higher than those of other diazotrophic groups (**Fig. 9 & Table S1**).

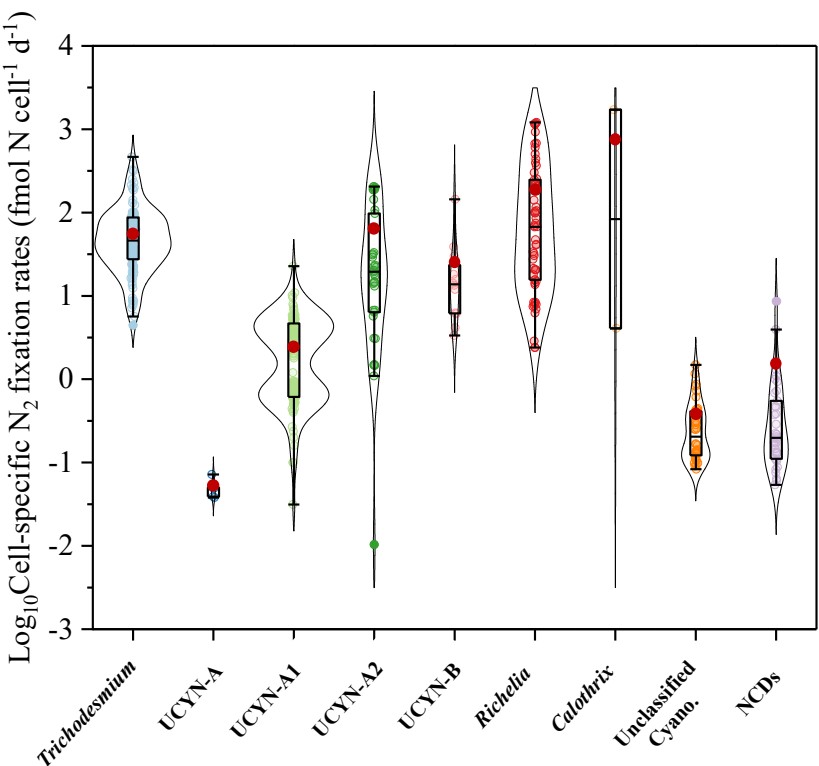

**Figure 9.** Violin plot of cell-specific N$_2$ fixation rates, including measurements for *Trichodesmium*, UCYN-A, UCYN-A1, UCYN-A2, UCYN-B, heterocystous cyanobacteria, unclassified cyanobacteria, and NCDs. The range of each box spans the 25th−75th percentile of data, the black line in each box is the median, and the red dot represents the arithmetic mean.

**3.3 Diazotrophic abundance**

The depth-integrated cell abundances and volumetric cell abundances in upper 25 m are also shown as the arithmetic means in 3° latitude × 3° longitude bins (**Fig. 10**). *Trichodesmium* abundance generally decreased from the west to the east in the Atlantic Ocean (**Fig. 10a–b**). In the Pacific Ocean, *Trichodesmium* appeared more abundant in the west. The abundance data of heterocystous diazotrophs were still scarce (**Fig. 10c, e**). The volumetric cell-count-based abundance data are also displayed
in three additional depth intervals (**Fig. S6**).

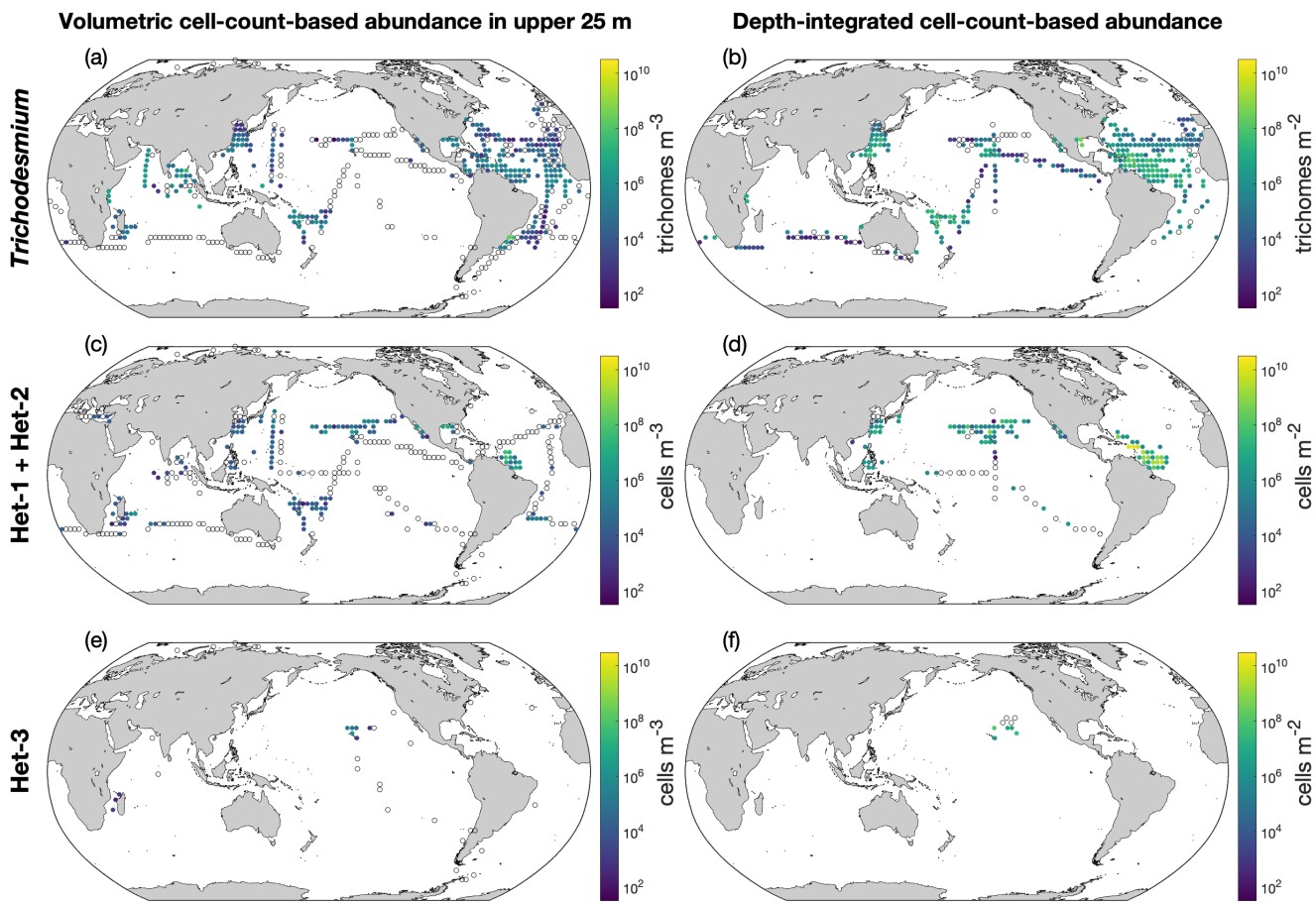

**Figure 10.** Depth-integrated cell abundances and volumetric cell abundances in upper 25 m in the version 2 of the database. The panels show (**a–b**) *Trichodesmium*, (**c–d**) het-1/2, and (**e–f**) het-3. For a clear demonstration, data are binned to 3° latitude × 3° longitude, and arithmetic means in each bin are shown. Zero-value data are denoted as open black circles.


Gene copies of *nifH* had better spatial coverage than the cell-count data (**Fig. 11**). Depth-integrated *Trichodesmium nifH* copies were also more abundant in the western Pacific and western Atlantic Oceans (**Fig. 11a**). Some high depth-integrated *nifH* abundance of UCYN-A and UCYN-B were also reported in the northwestern and southwestern Pacific Ocean (**Fig. 11c, e**). High *nifH* abundances of *Richelia* were found in the southwestern Pacific Ocean and western Atlantic Oceans (**Fig. 11i**).

The *nifH* abundance data for UCYN-C and het-3 were sparse. The volumetric *nifH* abundance data are displayed in three depth intervals (**Fig. 11 & Fig. S7**). Almost all diazotrophs were more abundant in the upper 25 m than in deeper water.

**Volumetric *nifH*-based abundance in upper 25 m**          **Depth-integrated *nifH*-based abundance**

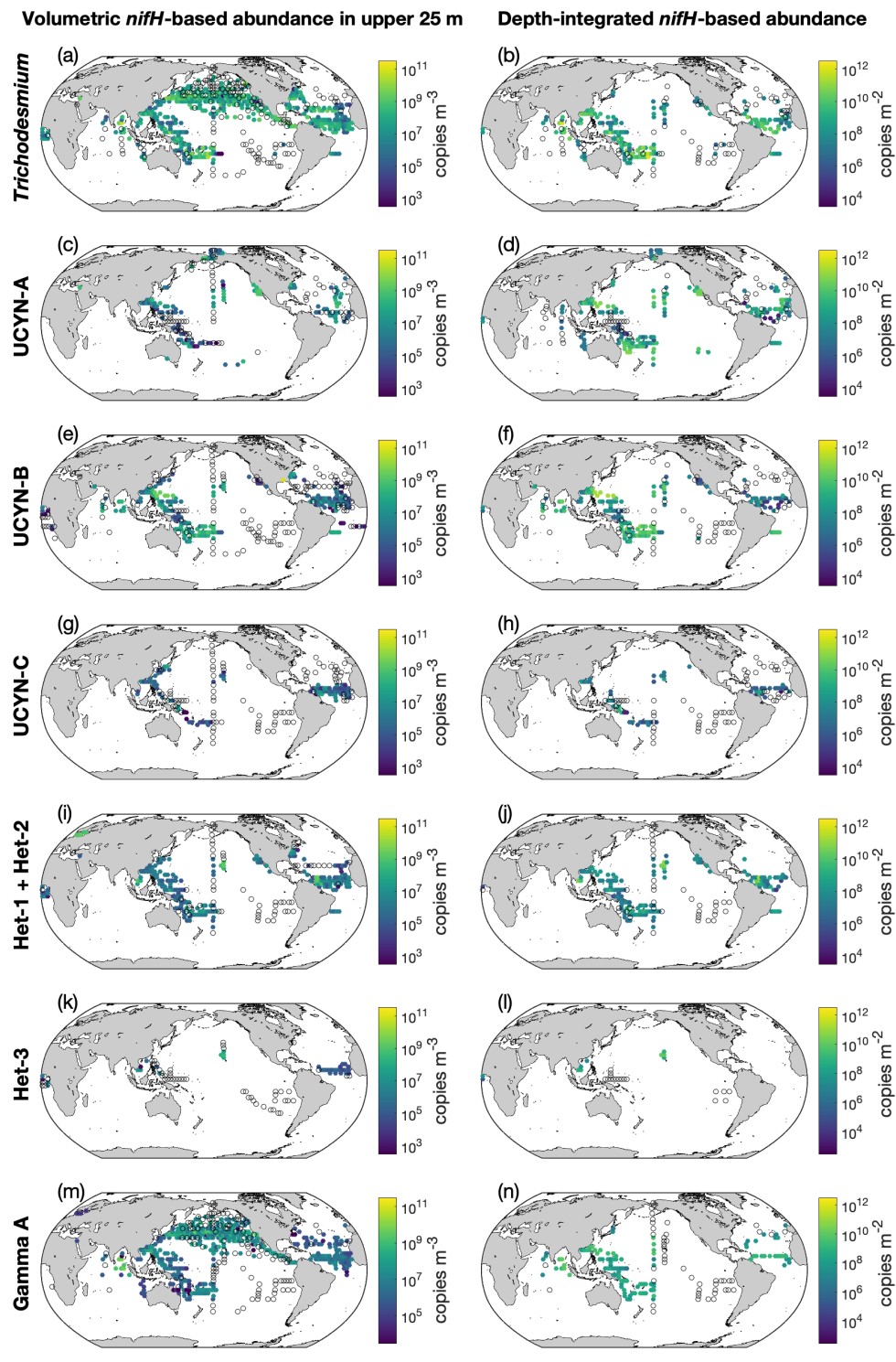

**Figure 11.** Volumetric and depth-integrated *nifH* gene copy abundances in the version 2 of the database. For volumetric abundances, only data in the upper 25 m are shown. The panels show gene copy abundances of (**a–b**) *Trichodesmium*, (**c–d**) UCYN-A, (**e–f**) UCYN-B, (**g–h**) UCYN-C, (**i–j**) het-1 + het-2, (**k–l**) het-3, and (**m–n**) Gamma A (an NCD phylotype). The depth-integrated data for Gamma A are not available. For a clear demonstration, data are binned to 3° latitude × 3° longitude and arithmetic means in each bin are shown. Zero-value data are denoted as open black circles.

### 3.4 First-order estimate of global oceanic N$_2$ fixation rate

Compared to version 1, the spatial coverage of data in version 2, in terms of the fraction of 3°×3° bins, was greatly increased in all ocean basins (**Table 8**). The spatial data coverage was very low in the Southern and Arctic Oceans (1% and 2% of total bins, respectively) (**Table 8**) and we therefore did not estimate total N$_2$ fixation rates for these two basins. Please note that the inaccurate areas of the North and South Pacific Oceans used in estimating global oceanic N$_2$ fixation rate by Luo et al. (2012) was corrected in this study (**Table 8**).

**Table 8.** First-order estimates of N$_2$ fixation rates based on their arithmetic means in different ocean basins. Data are first binned to 3° latitude ×3° longitude grids before being used to calculate arithmetic means in each basin. The arithmetic means are multiplied by the basin areas to calculate the N$_2$ fixation rates of each basin. NQ: not quantified due to limited data points. ND: no data. The percentages in parentheses are fraction of the 3° ×3° bins in each basin that have measurements. The reported uncertainties are one standard error of the mean.

| Region | Number of binned data | | Latitudinal range | | Ocean area (× 10$^{12}$ m$^2$) | | Arithmetic mean N$_2$ fixation rate (µmol N m$^{-2}$ d$^{-1}$) | | Areal sum of N$_2$ fixation rate (Tg N yr$^{-1}$) | |
|---|---|---|---|---|---|---|---|---|---|---|
| | Version 1 | Version 2 | Version 1 | Version 2 | Version 1 | Version 2 | Version 1 | Version 2 | Version 1 | Version 2 |
| North Atlantic | 47 (9%) | 116 (21%) | 0°−55°N | 0°−55°N | 37 | 37 | 55±9 | 213±46 | 10±2 | 40±9 |
| South Atlantic | 14 (4%) | 52 (15%) | 40°S −0° | 45°S −0° | 26 | 30 | 13±4 | 30±5 | 1.8±0.6 | 5±1 |
| North Pacific | 34 (4%) | 143 (17%) | 0°−55°N | 0°−55°N | 75 | 75 | 111±17 | 144±28 | 42±7 | 55±11 |
| South Pacific | 20 (2%) | 100 (12%) | 40°S−0° | 45°S−0° | 63 | 69 | 61±7 | 250±66 | 20±2 | 88±23 |
| Indian Ocean | ND | 47 (9%) | | 45°S−25°N | | 56 | ND | 123±50 | ND | 35±14 |
| Mediterranean Sea | 1 (3%) | 9 (23%) | 30°N−45°N | 30°N−45°N | 2.5 | 2.5 | NQ | 5±1 | NQ | 0.06±0.02 |
| Arctic Ocean | ND | 17 (2%) | | | | 11 | ND | 23±5 | ND | NQ |
| Southern Ocean | ND | 10 (1%) | | | | 60 | ND | 9±8 | ND | NQ |
| Global Ocean | | | | | | | | | 74±7 | 223±30 |

We first compared the N₂ fixation rates estimated based on arithmetic means between using version 1 and version 2 (**Table 8**). Using available data in version 2, the global N₂ fixation rate was determined to be 223±30 Tg N yr⁻¹, which was three times that obtained from version 1 (**Table 8**). The substantial increase was mostly driven by notable changes in the South Pacific, North Atlantic, and Indian Oceans. In the South Pacific Ocean, numerous high N₂ fixation rates were observed in the western subtropical region over the past decade (**Fig. 12**), resulting in a substantial increase of 68±23 Tg N yr⁻¹ in the estimated N₂ fixation rate for this basin (**Table 8**). It is worth noting that these newly recorded measurements in the western subtropics of the South Pacific Ocean might even be underestimated since most of them were obtained using the original ¹⁵N₂ bubble method. In the North Atlantic Ocean, the estimated N₂ fixation rate also experienced an increase of 30±9 Tg N yr⁻¹ for (**Table 8**), without any discernible pattern regarding the locations of the new high N₂ fixation measurements (**Fig. 13**). Furthermore, in the Indian Ocean, the improved data coverage in version 2 (**Fig. 8a**) supported the estimation of an N₂ fixation rate of 35±14 Tg N yr⁻¹ for this basin (**Table 8**), which was not possible to calculate using version 1 due to insufficient data availability.

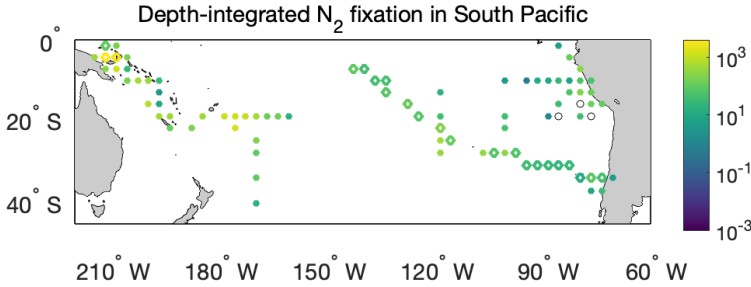

**Figure 12.** Depth-integrated N₂ fixation rates in the South Pacific Ocean (μmol N m⁻² d⁻¹). The shown data are arithmetic mean rates in 3° latitude ×3° longitude bins. Empty diamonds and filled circles denote the existing data in the version 1 of the database and the new data added to version 2, respectively.

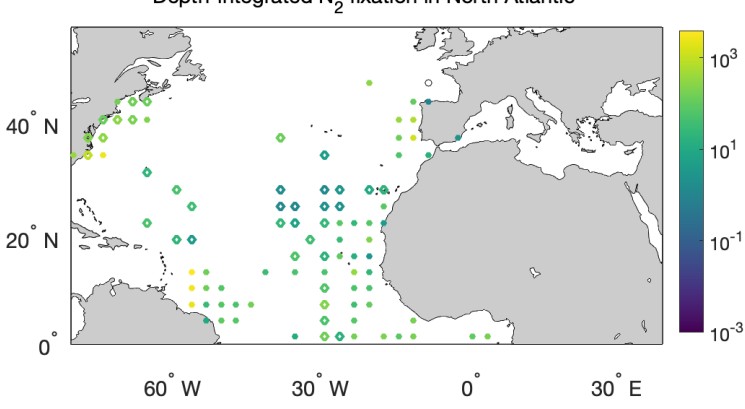

**Figure 13.** Depth-integrated N₂ fixation rates in the North Atlantic Ocean (μmol N m⁻² d⁻¹). The shown data are arithmetic mean rates in 3° latitude ×3° longitude bins. Empty diamonds and filled circles denote the existing data in the version 1 of the database and the new data added to version 2, respectively.

However, when estimating the global marine $N_2$ fixation rate using geometric means, both version 1 and version 2 yielded
similar rates of approximately 50 Tg N $yr^{-1}$ (**Table 9**). The $N_2$ fixation rates in each basin tended to follow a log-normal
distribution (**Fig. 14**), with the geometric mean aligning near the peak of the distribution. In the South Pacific Ocean, as
discussed earlier, version 2 included a substantial number of newly observed high $N_2$ fixation rates, but it also incorporated a
significant number of rates that were much lower than those in version 1 (**Fig. 14c**). This could be partially attributed to
enhanced detection limits in measurements. Consequently, while version 2 yielded a much higher arithmetic mean $N_2$ fixation
rate compared to version 1 for the South Pacific Ocean (**Table 8**), their geometric means remained quite similar (**Table 9**). In
the North Pacific Ocean, for the same reasons, the arithmetic mean $N_2$ fixation rates obtained from both versions were very
close, while the geometric mean from version 1 could be even higher than that from version 2 (**Tables 8 & 9; Fig. 14a**). These
analyses reveal that, despite the similarity in geometric means of $N_2$ fixation rates obtained from both versions of the database,
the higher arithmetic means in version 2 were not coincidental. Instead, they were a direct outcome of the improved
measurement methods and the expanded spatial and temporal coverage of marine $N_2$ fixation over the past decade.
Consequently, previous assessments of the global marine $N_2$ fixation rate were likely underestimated due to the absence of
these new measurements.

**Table 9.** Same as Table 8 but based on the geometric means of $N_2$ fixation rates. The numbers in parentheses are estimated
ranges based on one standard error of log-transformed $N_2$ fixation rates (*see* **Section 2.4**).

| Region | Proportion of zero-value data | | Geometric mean $N_2$ fixation rate ($\mu mol$ N $m^{-2}$ $d^{-1}$) | | Areal sum of $N_2$ fixation rate (Tg N $yr^{-1}$) | |
|---|---|---|---|---|---|---|
| | Version 1 | Version 2 | Version 1 | Version 2 | Version 1 | Version 2 |
| North Atlantic | 0% | 5% | 22 (18–26) | 46 (39–54) | 4.1 (3.3–5.0) | 8.7 (7.4–10.1) |
| South Atlantic | 0% | 25% | 8 (6–10) | 15 (13–17) | 1.1 (0.9–1.3) | 2.3 (1.9–2.7) |
| North Pacific | 3% | 6% | 73 (63–83) | 45 (39–52) | 27.8 (24.2–32.0) | 17.3 (15.1–19.8) |
| South Pacific | 0% | 9% | 52 (45–59) | 51 (43–61) | 16.6 (14.4–19.1) | 18.0 (15.1–21.4) |
| Indian Ocean | ND | 0% | ND | 25 (20–31) | ND | 7.1 (5.7–8.9) |
| Mediterranean Sea | 0% | 3% | NQ | 3 (2–4) | NQ | 0.04 (0.03–0.05) |
| Arctic Ocean | ND | 2% | ND | 14 (11–18) | ND | NQ |
| Southern Ocean | ND | 70% | ND | 4 (1–16) | ND | NQ |
| Global Ocean | | | | | 50 (43–57) | 53 (45–63) |

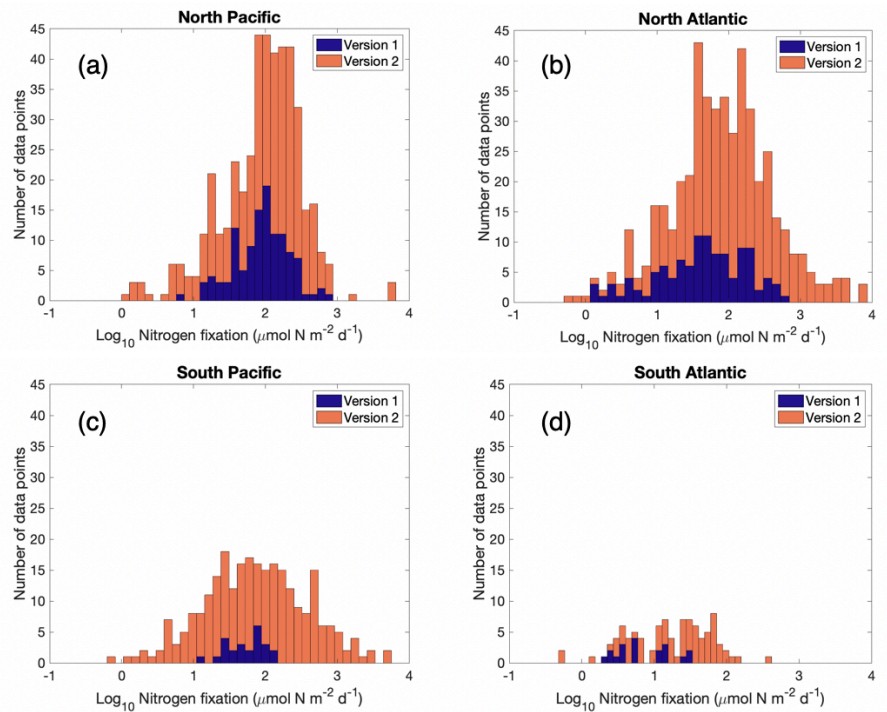

**Figure 14.** Comparison of the distribution of log-transformed $N_2$ fixation rates between the two versions of the database. Note that the zero-value data are not included because of the log-transformation. The comparison is performed for data in (**a**) North Pacific, (**b**) North Atlantic, (**c**) South Pacific, and (**d**) South Atlantic Oceans.

We must emphasize that this calculation simply used average $N_2$ fixation rates in different ocean basins, and therefore can only be considered as a first-order estimate. Furthermore, limited measurements have shown a large range of $N_2$ fixation rates in the Southern Ocean (**Fig. 8**). Considering its vast area, future measurements expanding coverage of $N_2$ fixation rates in the Southern Ocean (but see White et al. 2022) may help to better constrain the contribution of $N_2$ fixation to the N budget of the global ocean.

## 4. Discussion

### 4.1 Comparison of $N_2$ fixation measured using $^{15}N_2$ bubble and dissolution methods

To date, the discrepancy in $N_2$ fixation rates estimated using different $^{15}N_2$ tracer methods remains unclear. As shown above, the volumetric $N_2$ fixation rates obtained by the original $^{15}N_2$ bubble method and the $^{15}N_2$ dissolution method spanned a similar range (**Fig. 1**), while the average rates using the former method were significantly lower than that measured using the latter method (one-tailed Wilcoxon test, $p<0.001$, $n = 2460$ and $1128$). With substantial data accumulated over the past decade, we further compared $N_2$ fixation rates measured using the two methods at close locations and sampling time, although the samples

were not identical. We first binned data collected from the same months, horizontal locations (3° latitude × 3° longitude) and depth intervals (0–5 m, 5–25 m, 25–100 m, and 100–200 m), and calculated the average rates for each method in each bin. The results showed that the original $^{15}N_2$ bubble method produced lower rates than the $^{15}N_2$ dissolution method in 69% of the cases (**Fig. 13**). Furthermore, our analysis employing the generalized additive model (GAM) revealed that the relationship between the rates measured using the original $^{15}N_2$ bubble method and those obtained through the $^{15}N_2$ dissolution method closely

adhered to the 1:1 line, albeit with slightly lower values in the former (**Fig. 15**). Please note that this slightly lower values can still result in significant underestimation in measured $N_2$ fixation rates, because the GAM model was applied in a logarithmic space. It is crucial to reiterate that the rates being compared were derived from different samples, emphasizing the necessity for more future investigations that directly compare the two methods using the same samples with controlled parameters such as temperature, volume of injected $^{15}N_2$ and incubation volume. Despite this limitation, our analysis suggests that the extensive

body of historical marine $N_2$ fixation rate data obtained through the original $^{15}N_2$ bubble method still holds a value, particularly in the examination of spatial and temporal variations in $N_2$ fixation.

We also used the same procedure to compare the $N_2$ fixation rates measured using the acetylene reduction assays and the $^{15}N_2$ tracer methods. However, there were insufficient pairs of data available for reliable comparisons ($n = 16$ for acetylene reduction versus the $^{15}N_2$ dissolution method; $n = 6$ for acetylene reduction versus original $^{15}N_2$ bubble method).

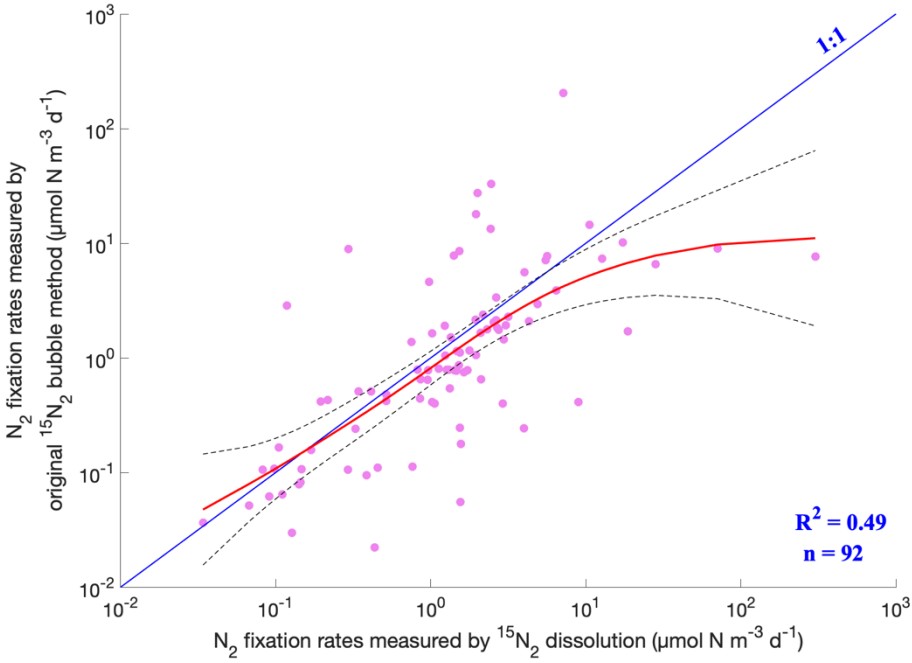

**Figure 15.** Comparison of measured $N_2$ fixation rates using the original $^{15}N_2$ bubble method and the $^{15}N_2$ dissolution method. The pink dots are measurements. The fitted results of the two methods by the generalized additive model (GAM) and confidence intervals are represented by the red solid line and the dashed black lines, respectively. Only the $N_2$ fixation rates measured with incubation periods of 24 hours were included in this analysis.

## 4.2 Comparison between diazotrophic cell counts and *nifH* copies

Whether *nifH* copies can be used to infer diazotrophic abundance and to study diazotrophic biogeography, while some challenges remain in conversion of gene counts to biomass, as a large range in the number of *nifH* copies per diazotrophic cell has been reported (**Table S2**). In version 2, we first converted *Trichodesmium* trichome abundance to cell abundance using the same conversion factor of 100 cells trichome$^{-1}$ as that used in Luo et al. (2012). This conversion resulted in mean and variance of log-10 transformed *Trichodesmium* cell abundance ($10^{6.5\pm1.3}$ cells L$^{-1}$) very similar to that of *Trichodesmium nifH* gene copies ($10^{6.6\pm1.5}$ copies L$^{-1}$) (**Fig. 16a**). More recently, however, a much lower conversion factor of 13.2±2.3 cells trichome$^{-1}$ was suggested for *Trichodesmium* based on larger sample sizes, although a very large range of 1.2–685 cells trichome$^{-1}$ were reported (White et al., 2018). Hence, when a conversion factor of 10 cells trichome$^{-1}$ was applied, the *Trichodesmium nifH* gene copy abundance was an order of magnitude higher than its cell abundance (**Fig. 16a**). This result was within the reported mean *nifH*:cell ratios for *Trichodesmium*, albeit based on sparse samples, on the order of 10 – 100 (**Table S2**). It is worth noting that there have been suggestions that the observed *nifH*:cell ratio for *Trichodesmium* may be overestimated due to methodological limitations (Gradoville et al., 2022). Our analyses underscore the importance of enumerating *Trichodesmium* cells, rather than solely focusing on trichomes, in future studies, as suggested by White et al. (2018). While counting all *Trichodesmium* cells may be impractical, it would be valuable to report the number of cells in random samples of *Trichodesmium* trichomes.

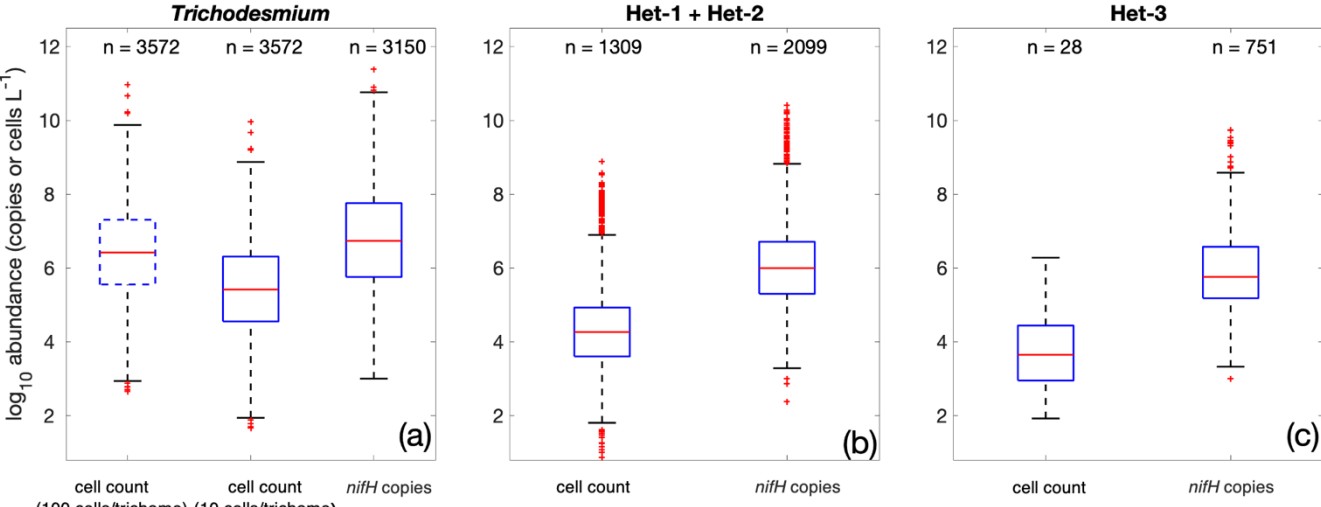

**Figure 16.** Comparison of all cell-count and *nifH* gene copy abundance data in the database. The box plots show the median (central line), 25th and 75th percentiles (upper and lower edges of the boxes), 5th and 95th percentiles (error lines) and outliers (red crosses) of log-10 transformed data. The comparisons are conducted for (**a**) *Trichodesmium*, (**b**) het-1/2, (**c**) het-3. Note that two conversion factors of 10 and 100 cells trichome$^{-1}$ are used for *Trichodesmium*.

The same analyses for heterocystous cyanobacteria showed that the *nifH* gene copy abundances were approximately two orders of magnitude greater than the cell abundances in terms of both mean and distribution (**Fig. 16b, c**). It must be noted that this simple analysis used all the data in our database. The limited *in situ* measurements for identical samples resulted in a mean *nifH*:cell ratio of 76 for heterocystous cyanobacteria (**Table S2**), consistent with our simple analysis.

In contrast, much lower *nifH*:cell ratios (1.51 – 2.58) were derived from regression analysis for heterocystous cyanobacteria and UCYN-B collected in the subtropical North Pacific (Gradoville et al., 2022). Considering these overall scarce measurements and the outcomes of our analysis, it is plausible that there is substantial variability in *nifH*:cell ratios. We expect that future studies, focusing on constraining these ratios and identifying mechanisms underlying variability in these ratios, will contribute to a more comprehensive understanding of the connection between *nifH* gene counts and diazotrophic cell abundance.

The application of qPCR assays for *nifH* based abundance (DNA) and expression (RNA) emerged as a critical step forward in our understanding of the distribution, abundance, and physiology (e.g., expression of *nifH*) of diazotrophs (Short and Zehr, 2005; Zehr and Riemann, 2023). Until then, estimating the abundances of diazotrophs were limited to those that could be identified by microscopy, e.g., *Trichodesmium*, heterocystous cyanobacteria (e.g., *Richelia*, *Calothrix, Anabaena, Nodularia, Aphanizomenon*), and some unicellulars (e.g., *Cyanothece*, later *Crocosphaera*). Thus, qPCR enabled the study of diazotrophic targets (and their activity) without the need to microscopy identify them, which came later as some diazotrophs would (and still) require application of FISH techniques for identification (Biegala and Raimbault, 2008). Additionally, qPCR allowed the study of *in situ* activity (gene expression) by diazotrophs without the need for cultivation. Although beyond the scope of the work presented here, important considerations should be taken into account when using microscopy and qPCR datasets (Table S3), for example, in application to biogeochemical models (Meiler et al., 2023).

## 4.3 Biomass conversion factor

For possible further usage of cell-counted abundance data, here we suggest carbon biomass conversion factors for different diazotrophic groups (**Tables 10 and S4**). Most biomass conversion factors suggested here are the same as those used in Luo et al. (2012), excluding UCYN-A and heterocystous cyanobacteria where new information has become available or additional consideration is necessary. A recent study has discovered a new symbiosis association between the unicellular diazotroph (UCYN-C) and diatom *Epithemia* strains (Schvarcz et al., 2022). However, the conversion factor of UCYN-C could not be updated in this study due to insufficient information on the biovolumes of host cell.

The conversion factor for UCYN-A was updated because it has been found to live symbiotically with haptophyte *Braarudosphaera bigelowii* and relatives (Thompson et al., 2012; Hagino et al., 2013). Because the host and UCYN-A should function together, the host biomass is allocated to UCYN-A. It has been reported that each haptophyte cell hosts one UCYN-

A1 cell (Cornejo-Castillo et al., 2019) or one UCYN-A2 cell (Suzuki et al., 2021). We used the empirically-derived equation (Verity et al., 1992):

$$C = 0.433 \times V^{0.863}, \tag{1}$$

to estimate biomass of UCYN-A and their hosts. The biomass of a UCYN-A1 cell with a diameter of 1 μm and a UCYN-A2 cell with a diameter of 1.6–3.3 μm (Cornejo-Castillo et al., 2019; Martínez-Pérez et al., 2016) equate to 0.2 pg C and 0.8–5.5 pg C, respectively. The biomasses of the host cell for UCYN-A1 or UCYN-A2 is 1.5–2.2 pg C or 6.8–43 pg C according to their reported cell diameters (2–2.3 μm or 3.6–7.3 μm), respectively (Martínez-Pérez et al., 2016; Cornejo-Castillo et al., 2019). Hence, the biomasses of the UCYN-A1 and the UCYN-A2 symbioses are 1.7–2.4 pg C and 7.6–48 pg C, respectively. After normalizing the symbiotic biomass to the number of UCYN cells in each symbiosis (1 for both UCYN-A1 and UCYN-A2), the biomass conversion factors are 1.7–2.4 pg C (UCYN-A1 cell)$^{-1}$ and 7.6–48 pg C (UCYN-A2 cell)$^{-1}$.

Because heterocystous cyanobacteria and their host diatoms form DDAs, similar to UCYN-A, we also suggest allocating the biomass of host diatoms to each associated diazotrophic cell (**Table S4**). The biomasses of heterocystous cells and vegetative cells in *Richelia* filaments were updated according to the cell dimension data reported in Caputo et al. (2019) using the same empirical equation above. The carbon biomass of host diatom cells was calculated using an empirical equation (Menden-Deuer and Lessard, 2000):

$$C = 0.117 \times V^{0.881}, \tag{2}$$

where $C$ is the diatom cell carbon biomass (pg C cell$^{-1}$), and $V$ is the average cell biovolume (μm$^3$) of each diatom genus, for which values from a database (Harrison et al., 2015) were used in this study (**Table S4**). Each host diatom associates with multiple heterocysts. The numbers of *Richelia* heterocysts associated with *Hemiaulus*, *Rhizosoleniae* and *Chaetoceros* were observed to be within the range of 1–2, 1–5 and 3–10 respectively (Villareal et al., 2011; Yeung et al., 2012; Caputo et al., 2019), we selected both the maximum and minimum to do the estimation. The number of vegetative cells in each heterocyst were also updated according to Caputo et al. (2019). Conversion factors for DDAs were estimated by dividing the total biomass of each DDA by the number of associated heterocysts. Changes in the number of *Richelia* in *Rhizosoleniae* (1 or 5) would make a large variation in its conversion factor, possibly due to large host biomass, therefore we keep them both to let users take caution when using this conversion factor. The resulting biomass conversion factors of *Richelia-Hemiaulus* and *Richelia-Chaetoceros* associations were estimated to be 280 (range: 150–1250) and 430 (range: 10–1900) pg C heterocyst$^{-1}$, respectively (**Table S4**), as the number of filaments did not have a large impact on the conversion factors.

It is important to reiterate that these biomass conversion factors are only applicable to cell-count data. Attempting to convert *nifH* gene copies to biomass is not recommended due to significant uncertainties associated with *nifH*:cell, as previously discussed.

**Table 10.** Recommended carbon biomass conversion factors and their likely ranges for diazotrophic groups.

| | Trichodesmium (pg C cell$^{-1}$) | UCYN-A1 (pg C cell$^{-1}$) | UCYN-A2 (pg C cell$^{-1}$) | UCYN-B (pg C cell$^{-1}$) | UCYN-C (pg C cell$^{-1}$) | **Het-1** Richelia-Hemiaulus (pg C heterocyst$^{-1}$) | **Het-2** Richelia-Rhizosolenia (pg C heterocyst$^{-1}$) | **Het-3** Richelia-Chaetoceros (pg C heterocyst$^{-1}$) |
|---|---|---|---|---|---|---|---|---|
| Recommended | 300 | 2 | 30 | 20 | 10 | 350 | 450 (5 heterocyst DDA$^{-1}$) or 1900 (1 heterocyst DDA$^{-1}$) | 50 |
| Likely range | 100−500 | 1−3 | 10-50 | 4−50 | 5−24 | 150−1030 | 19−5700 | 9−300 |

## 5. Conclusions

In this study, we updated the global oceanic diazotrophic database by Luo et al. (2012) by adding new measurements reported in the past decade. Although the spatial coverage of the data was greatly expanded by this effort, the data distribution is still uneven, with most measurements reported from the Pacific and Atlantic Oceans. Using the updated database, the estimation of global oceanic N$_2$ fixation based on arithmetic rates in ocean basins was increased from 74±7 Tg N yr$^{-1}$ to 217±29 Tg N yr$^{-1}$. This change is largely attributable to a new estimate for the Indian Ocean, and a much elevated estimate for the South Pacific Ocean that would account for ~40% of global N$_2$ fixation, this high estimation for the South Pacific Ocean is in line with its qualification as a 'hot spot' for diazotrophy (Messer et al., 2016; Bonnet et al., 2017), partly due to iron fertilization processes in this region (Bonnet et al., 2023). Due to data sparsity, our updated estimation did not include N$_2$ fixation in the Southern and Arctic Oceans. Furthermore, data were more concentrated in surface seawater, and a significant amount of data were measured with incubation periods shorter than a daily cycle (24 h), limiting reliable evaluations of depth-integrated N$_2$ fixation rates. Although this result suggests more balanced N inputs and losses in the global ocean than the previous estimate suggested, large uncertainties still exist. We also compared the N$_2$ fixation rates measured using addition of a bubble of labelled gas or addition of dissolving $^{15}$N$_2$ gases reported at the same location and month (not necessarily in identical samples). The results indicated that the original $^{15}$N$_2$ bubble method produces lower rates than the $^{15}$N$_2$ dissolution method in 69% of the cases. These results reveal that, despite decades of effort, the ocean is still undersampled in terms of the distribution of diazotrophs and N$_2$ fixation rate measurements. Our analyses suggest that prioritizing N$_2$ fixation measurements in the South Pacific Ocean and high northern latitudes can significantly reduce the current uncertainty of N$_2$ fixation rates in the global ocean. Nevertheless, we believe that this updated diazotrophic database, supplemented with enhanced data from the past decade, is timely and can be helpful to scientists studying the marine biogeochemical cycle of N.

**Data availability.**

The database is available in a data repository (https://doi.org/10.6084/m9.figshare.21677687) (Shao et al., 2022)

**Author contributions.**

Y.-W. Luo conceived and designed the structure of the database. Z. Shao, Y. Xu, H. Wang, W. Luo, L. Wang, Y. Huang and Y.-W.Luo collected the data and updated the database. Z. Shao, Y. Xu, H. Wang, S. C. Doney and Y.-W. Luo analyzed data. Other authors contributed the data. Z. Shao, Y. Xu and Y.-W. Luo wrote the first draft of the manuscript, and all authors revised the manuscript.

**Competing interests.**

The authors declare that they have no conflicts of interest.

**Acknowledgments.**

We would like to thank all the scientists and crew who contributed to sample and measure these tremendous amounts of diazotrophic data in the past several decades. We also thank Christopher Somes and an anonymous reviewer for their constructive comments. This work was supported by the National Natural Science Foundation of China (grants 41890802 and 42076153).

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
