# Peer review of "Global oceanic diazotroph database version 2 and elevated estimate of global oceanic N2 fixation"

_Earth System Science Data, 2023_

## Referee Comment (RC1)

**Review ms essd-2023-13**

Shao et al. present an update of the diazotroph database published in 2012
https://essd.copernicus.org/articles/4/47/2012/

The new version adds up data published between 2012 and 2023, including volumetric and depth-integrated $N_2$ fixation rates, diazotroph microscope counts and nifH gene counts. This new version also discusses microscope-nifH count comparisons. While this update is valuable for the community as a tool for comparison and contextualization of diazotrophy studies, it fails to account for many diazotrophy studies published between 2012 and 2023. The text has several misinterpretations that need correction. The new version also includes $N_2$ fixation rates proxied with other methods (ARA). I think this is a major problem, since these rates are not currently solidly comparable and downplay the robustness of the database. The manuscript also eliminates nifH gene counts from non-cyanobacterial diazotrophs (NCDs), which is another major issue since NCDs are considered to be outnumber cyanobacterial diazotrophs in the ocean. Finally, the diazotroph microscopy count versus nifH gene count conversion discussion does not seem appropriate here, since very few of the papers listed have compared these approaches on a same given sample, and the issue has been discussed thoroughly in other publications by specialists.
In all, while I acknowledge the effort and usefulness of this manuscript, I advise major revisions as detailed in the comments below.

L28: N2 gas is not inert to diazotrophs.
L31: The balance between N loss/gains in the ETSP has been widely demonstrated to be false in several publications after that of Deutsch et al., see for example (Knapp et al. 2016; Bonnet et al. 2017).

L35: Only cyanobacterial diazotrophs can be confidently counted by microscopy.

L36: "NifH gene copies"

L40: This issue has been thoroughly discussed in (Gradoville et al. 2022), validating the use of nifH gene counts as a means to quantify diazotrophs.

L42-47: Other sources of unbalance should be briefly mentioned here.

L50: Diazotroph activity was there before, it is our notion of them that increases, the data available.

L56-57: I don't think that the dataset assembled here covers enough studies comparing microscopy and nifH based comparisons, and I strongly recommend removing this sentence and section 4.2 from the manuscript.

L61: The $N_2$ fixation rates from Tang et al. 2019 are based on an ARA-15$N_2$ fixation comparison including only 8 datapoints. This is not robust enough to provide a reliable comparison and downplays the robustness of the 15N2-based rates dataset collected here. I strongly recommend removing these from the database and derived basin-scale and global calculations. These may be mentioned as discussion and the Tang paper cited, but not included for quantitative purposes.

L72: Removing NCDs is an error in my opinion. NCDs have recurrently been shown to be dominant in the ocean (Farnelid et al. 2011; Delmont et al. 2018, 2021; Riemann, Farnelid, and Steward 2010) and may impact N cycling decisively (Riemann et al. 2022; Turk-Kubo et al. 2022). I strongly recommend that any nifH gene counts of NCDs are added. The previous database included Gamma A and Cluster III. I don't see a solid reason to remove NCDs from the database at this stage, as evidence of their importance increases.

L82: Group-specific $N_2$ fixation rates can only be estimated using single-cell approaches. I'm not sure what approach was followed here to derive specific rates, but these can certainly not be estimated with the data

collected here. I would rather recommend the authors to collect all *Trichodesmium,* UCYN-B, DDAs and UCYN-A single-cell rates published, which would be very helpful for the community. See for instance (Foster, Sztejrenszus, and Kuypers 2013; Foster et al. 2011; Benavides et al. 2017; Bonnet et al. 2016; Filella et al. 2022; Krupke et al. 2015; K. Harding et al. 2018; Mills et al. 2020; K. J. Harding et al. 2022; Benavides et al. 2022).

Tables 2 and 4 : Many studies are missing in this table, some include (Benavides et al. 2014, 2021; Saulia et al. 2020; Henke et al. 2018; Bonnet et al. 2018; Gradoville et al. 2017; Moreira-Coello et al. 2017; Wilson et al. 2019). Also, in the table some studies are listed as not including counts of some diazotrophs, which needs correction (e.g. Bombar 2011 and Bonnet 2015, 2019 did have qPCR counts). Please revise all these publications thoroughly and correct accordingly.

L104: The ARA method is rarely used nowadays.

L106-107: The ARA to $N_2$ fixation ratio is highly variable (Mulholland et al. 2006; Benavides et al. 2011; Wilson et al. 2012)

L110: Many other factors affect this difference, including acetylene gas impurity, Bunsen dissolution coefficient, etc.

L112: This is not true. The 15N2 method is much more sensitive, does not require biomass preconcentration (biomass is concentrated during filtration, after the incubation), and requires longer incubations for enough tracer to be detectable in biomass. ARA is usually done in 3-4 h incubations and requires biomass pre-concentration to reach detectable signal (Staal et al. 2007; Benavides et al. 2011).

L120: Wannicke et al. say the opposite of Mohr and Grosskopf.

L123: What White et al. say is that the bubble release method is the most reliable and recommended by the diazotroph research community, with the elimination of rate underestimation benefits overcoming the very unlikely burdens of contamination. This should be corrected in L274-275 as well.

L150: There are 4 UCYN-A sublineages (Farnelid et al. 2016).

L328: UCYN-A has been found in symbiosis with other eukaryotic algae (Zehr et al. 2016)

L370: The first version of the database included all the authors that had contributed to its construction with their seagoing expeditions, laboratory analyses and publications. I humbly find it sad and somewhat unfair that this is not the case in this update.

**References**

Benavides, Mar, N. S. R. Agawin, J. Arístegui, P. Ferriol, and L. J. Stal. 2011. "Nitrogen Fixation by Trichodesmium and Small Diazotrophs in the Subtropical Northeast Atlantic." *Aquatic Microbial Ecology: International Journal.* https://doi.org/10.3354/ame01534.

Benavides, Mar, Hugo Berthelot, Solange Duhamel, Patrick Raimbault, and Sophie Bonnet. 2017. "Dissolved Organic Matter Uptake by Trichodesmium in the Southwest Pacific." *Scientific Reports* 7: 1–6.

Benavides, Mar, Mathieu Caffin, Solange Duhamel, Rachel Ann Foster, Olivier Grosso, Cécile Guieu, France Van Wambeke, and Sophie Bonnet. 2022. "Anomalously High Abundance of *Crocosphaera* in the South Pacific Gyre." *FEMS Microbiology Letters*, April. https://doi.org/10.1093/femsle/fnac039.

Benavides, Mar, L. Conradt, S. Bonnet, Ilana Berman-Frank, S. Barrillon, A. Petrenko, and Andrea Michelangelo Doglioli. 2021. "Fine Scale Sampling Unveils Diazotroph Patchiness in the South Pacific Ocean." *ISME Communications*, 1–6.

Benavides, Mar, Y. Santana-Falcón, N. Wasmund, and J. Arístegui. 2014. "Microbial Uptake and Regeneration of Inorganic Nitrogen off the Coastal Namibian Upwelling System." *Journal of Marine Systems* 140 (PB). https://doi.org/10.1016/j.jmarsys.2014.05.002.

Bonnet, Sophie, Hugo Berthelot, Kendra Turk-Kubo, Véronique Cornet-Barthaux, Sarah Fawcett, Ilana Berman-Frank, Aude Barani, et al. 2016. "Diazotroph Derived Nitrogen Supports Diatom Growth in the South West Pacific: A Quantitative Study Using NanoSIMS." *Limnology and Oceanography*. https://doi.org/10.1002/lno.10300.

Bonnet, Sophie, Mathieu Caffin, Hugo Berthelot, Olivier Grosso, Mar Benavides, Sandra Helias-Nunige, Cécile Guieu, Marcus Stenegren, and Rachel Ann Foster. 2018. "In-Depth Characterization of Diazotroph Activity across the Western Tropical South Pacific Hotspot of N2 Fixation (OUTPACE Cruise)." *Biogeosciences* 15 (13): 4215–32.

Bonnet, Sophie, Mathieu Caffin, Hugo Berthelot, and Thierry Moutin. 2017. "Hot Spot of N2 Fixation in the Western Tropical South Pacific Pleads for a Spatial Decoupling between N2 Fixation and Denitrification." *Proceedings of the National Academy of Sciences*. https://doi.org/10.1073/pnas.1619514114.

Delmont, Tom O., Juan José Pierella Karlusich, Iva Veseli, Jessika Fuessel, A. Murat Eren, Rachel A. Foster, Chris Bowler, Patrick Wincker, and Eric Pelletier. 2021. "Heterotrophic Bacterial Diazotrophs Are More Abundant than Their Cyanobacterial Counterparts in Metagenomes Covering Most of the Sunlit Ocean." *The ISME Journal*, October. https://doi.org/10.1038/s41396-021-01135-1.

Delmont, Tom O., Christopher Quince, Alon Shaiber, Ozcan C. Esen, Sonny T. M. Lee, Sebastian Lucker, and A. Murat Eren. 2018. "Nitrogen-Fixing Populations Of Planctomycetes And Proteobacteria Are Abundant In The Surface Ocean." *Nature Microbiology*. https://doi.org/10.1101/129791.

Farnelid, Hanna, Anders F. Andersson, Stefan Bertilsson, Waleed Abu Al-Soud, Lars H. Hansen, Søren Sørensen, Grieg F. Steward, Åke Hagström, and Lasse Riemann. 2011. "Nitrogenase Gene Amplicons from Global Marine Surface Waters Are Dominated by Genes of Non-Cyanobacteria." *PloS One* 6 (4). https://doi.org/10.1371/journal.pone.0019223.

Farnelid, Hanna, Kendra Turk-Kubo, María Del Carmen Muñoz-Marín, and Jonathan P. Zehr. 2016. "New Insights into the Ecology of the Globally Significant Uncultured Nitrogen-Fixing Symbiont UCYN-A." *Aquatic Microbial Ecology: International Journal* 77 (3): 128–38.

Filella, Alba, Lasse Riemann, France Van Wambeke, Elvira Pulido-Villena, Angela Vogts, Sophie Bonnet, Olivier Grosso, Julia M. Diaz, Solange Duhamel, and Mar Benavides. 2022. "Contrasting Roles of DOP as a Source of Phosphorus and Energy for Marine Diazotrophs." *Frontiers in Marine Science* 9. https://doi.org/10.3389/fmars.2022.923765.

Foster, Rachel A., Marcel M. M. Kuypers, Tomas Vagner, Ryan W. Paerl, Niculina Musat, and Jonathan P. Zehr. 2011. "Nitrogen Fixation and Transfer in Open Ocean Diatom-Cyanobacterial Symbioses." *The ISME Journal*. https://doi.org/10.1038/ismej.2011.26.

Foster, Rachel A., Saar Sztejrenszus, and Marcel M. M. Kuypers. 2013. "Measuring Carbon and N2 Fixation in Field Populations of Colonial and Free-Living Unicellular Cyanobacteria Using Nanometer-Scale Secondary Ion Mass Spectrometry." *Journal of Phycology* 49 (3): 502–16.

Gradoville, Mary R., Deniz Bombar, Byron C. Crump, Ricardo M. Letelier, Jonathan P. Zehr, and Angelicque E. White. 2017. "Diversity and Activity of Nitrogen-Fixing Communities across Ocean Basins." *Limnology and Oceanography* 62 (5): 1895–1909.

Gradoville, Mary R., Mathilde Dugenne, Annette M. Hynes, Jonathan P. Zehr, and Angelicque E. White. 2022. "Empirical Relationship between NifH Gene Abundance and Diazotroph Cell Concentration in the North Pacific Subtropical Gyre." *Journal of Phycology*, October. https://doi.org/10.1111/jpy.13289.

Harding, Katie J., Kendra A. Turk-Kubo, Esther Wing Kwan Mak, Peter K. Weber, Xavier Mayali, and Jonathan P. Zehr. 2022. "Cell-Specific Measurements Show Nitrogen Fixation by Particle-Attached

Putative Non-Cyanobacterial Diazotrophs in the North Pacific Subtropical Gyre." *Nature Communications* 13 (1): 1–10.

Harding, Katie, Kendra A. Turk-Kubo, Rachel E. Sipler, Matthew M. Mills, Deborah A. Bronk, and Jonathan P. Zehr. 2018. "Symbiotic Unicellular Cyanobacteria Fix Nitrogen in the Arctic Ocean." *Proceedings of the National Academy of Sciences*, 201813658.

Henke, Britt A., Kendra A. Turk-Kubo, Sophie Bonnet, and Jonathan P. Zehr. 2018. "Distributions and Abundances of Sublineages of the N2-Fixing Cyanobacterium Candidatus Atelocyanobacterium Thalassa (UCYN-A) in the New Caledonian Coral Lagoon." *Frontiers in Microbiology*. https://doi.org/10.3389/fmicb.2018.00554.

Knapp, Angela N., Karen L. Casciotti, William M. Berelson, Maria G. Prokopenko, and Douglas G. Capone. 2016. "Low Rates of Nitrogen Fixation in Eastern Tropical South Pacific Surface Waters." *Proceedings of the National Academy of Sciences* 113 (16): 4398–4403.

Krupke, Andreas, Wiebke Mohr, Julie Laroche, Bernhard M. Fuchs, Rudolf I. Amann, and Marcel M. M. Kuypers. 2015. "The Effect of Nutrients on Carbon and Nitrogen Fixation by the UCYN-A-Haptophyte Symbiosis." *The ISME Journal* 9 (7): 1635–47.

Mills, Matthew M., Kendra A. Turk-Kubo, Gert L. van Dijken, Britt A. Henke, Katie Harding, Samuel T. Wilson, Kevin R. Arrigo, and Jonathan P. Zehr. 2020. "Unusual Marine Cyanobacteria/Haptophyte Symbiosis Relies on N2 Fixation Even in N-Rich Environments." *The ISME Journal*, no. 3: 2395–2406.

Moreira-Coello, Víctor, Beatriz Mouriño-Carballido, Emilio Marañón, Ana Fernández-Carrera, Antonio Bode, and Marta M. Varela. 2017. "Biological N2 Fixation in the Upwelling Region off NW Iberia: Magnitude, Relevance, and Players." *Frontiers in Marine Science*. https://doi.org/10.3389/fmars.2017.00303.

Mulholland, Margaret R., Peter W. Bernhardt, Cynthia a. Heil, Deborah a. Bronk, and Judith M. O' Neil. 2006. "Nitrogen Fixation and Release of Fixed Nitrogen by Trichodesmium Spp. in the Gulf of Mexico." *Limnology and Oceanography* 51 (4): 1762–76.

Riemann, Lasse, Hanna Farnelid, and Grieg F. Steward. 2010. "Nitrogenase Genes in Non-Cyanobacterial Plankton: Prevalence, Diversity and Regulation in Marine Waters." *Aquatic Microbial Ecology: International Journal* 61 (3): 235–47.

Riemann, Lasse, Eyal Rahav, Uta Passow, Hans-Peter Grossart, Dirk de Beer, Isabell Klawonn, Meri Eichner, Mar Benavides, and Edo Bar-Zeev. 2022. "Planktonic Aggregates as Hotspots for Heterotrophic Diazotrophy: The Plot Thickens." *Frontiers in Microbiology* 0. https://doi.org/10.3389/fmicb.2022.875050.

Saulia, Emmrick, Mar Benavides, Britt Henke, Kendra Turk-Kubo, Haley Cooperguard, Olivier Grosso, Anne Desnues, et al. 2020. "Seasonal Shifts in Diazotrophs Players: Patterns Observed Over a Two-Year Time Series in the New Caledonian Lagoon (Western Tropical South Pacific Ocean)." *Frontiers in Marine Science* 7 (November): 1–11.

Staal, Marc, Sacco Te Lintel Hekkert, Geert Jan Brummer, Marcel Veldhuis, Cor Sikkens, Stefan Persijn, and Lucas J. Stal. 2007. "Nitrogen Fixation along a North – South Transect in the Eastern Atlantic Ocean." *Limnology and Oceanography* 52 (4): 1305–16.

Turk-Kubo, Kendra A., Mary R. Gradoville, Shunyan Cheung, Francisco Cornejo-Castillo, Katie J. Harding, Michael Morando, Matthew Mills, and Jonathan P. Zehr. 2022. "Non-Cyanobacterial Diazotrophs: Global Diversity, Distribution, Ecophysiology, and Activity in Marine Waters." *FEMS Microbiology Reviews*, November. https://doi.org/10.1093/femsre/fuac046.

Wilson, Samuel T., Daniela Böttjer, Matthew J. Church, and David M. Karl. 2012. "Comparative Assessment of Nitrogen Fixation Methodologies, Conducted in the Oligotrophic North Pacific Ocean." *Applied and Environmental Microbiology* 78 (18): 6516–23.

Wilson, Samuel T., Nicholas J. Hawco, E. Virginia Armbrust, Benedetto Barone, Karin M. Björkman, Angela K. Boysen, Macarena Burgos, et al. 2019. "Kīlauea Lava Fuels Phytoplankton Bloom in the North Pacific Ocean." *Science* 365 (6457): 1040–44.

Zehr, Jonathan P., Irina N. Shilova, Hanna M. Farnelid, Maria Del Carmen Muñoz-Maríncarmen, and Kendra A. Turk-Kubo. 2016. "Unusual Marine Unicellular Symbiosis with the Nitrogen-Fixing

Cyanobacterium UCYN-A." *Nature Microbiology* 2 (December). https://doi.org/10.1038/nmicrobiol.2016.214.

---

## Author Comment (AC1)

We have taken all the comments of the reviewers into account in the revision; replies to each of the comments are provided below in blue fonts.

**Reviewer #1:**

Shao et al. present an update of the diazotroph database published in 2012 https://essd.copernicus.org/articles/4/47/2012/

The new version adds up data published between 2012 and 2023, including volumetric and depth-integrated $N_2$ fixation rates, diazotroph microscope counts and nifH gene counts. This new version also discusses microscope-nifH count comparisons. While this update is valuable for the community as a tool for comparison and contextualization of diazotrophy studies, it fails to account for many diazotrophy studies published between 2012 and 2023. The text has several misinterpretations that need correction. The new version also includes N2 fixation rates proxied with other methods (ARA). I think this is a major problem, since these rates are not currently solidly comparable and downplay the robustness of the database. The manuscript also eliminates nifH gene counts from non-cyanobacterial diazotrophs (NCDs), which is another major issue since NCDs are considered to be outnumber cyanobacterial diazotrophs in the ocean. Finally, the diazotroph microscopy count versus nifH gene count conversion discussion does not seem appropriate here, since very few of the papers listed have compared these approaches on a same given sample, and the issue has been discussed thoroughly in other publications by specialists.
In all, while I acknowledge the effort and usefulness of this manuscript, I advise major revisions as detailed in the comments below.

Response: We thank the reviewer for very constructive and thorough comments. We particularly appreciate the reviewer's suggestion to include PIs of the data sources as coauthors, and we have adopted this suggestion. We are pleased to report that more than 80 PIs have agreed to join this collaborative effort. By doing so, we believe that this paper will not only showcase the significant impact of collective research efforts in $N_2$ fixation, but also the completeness of the database and the quality of the paper can be improved substantially. The new coauthors have started to identify missing datasets and to provide additional comments that will further enhance the quality of the paper. Other general comments have also been addressed:

(1)  We have decided not to include the ARA-based data in estimating the global $N_2$ fixation rate, but we have included them in the database for those who are interested in using them.

(2) The NCD data have been added to the database as an additional spreadsheet.

(3) We have decided to keep the comparison of *nifH* gene copies and diazotrophic cell counts in the paper, and we invite you to review our response to the related comments.

Once again, we thank the reviewer for their valuable feedback, which has helped us to improve the quality of our manuscript.

**L28:** $N_2$ gas is not inert to diazotrophs.

Response: We have changed the text to "Dinitrogen (N$_2$) fixation is a process carried out by a group of microorganisms known as diazotrophs. They are capable of converting the N$_2$ gas, which is not usable by most organisms, into bioavailable nitrogen (N)".

**L31:** The balance between N loss/gains in the ETSP has been widely demonstrated to be false in several publications after that of Deutsch et al., see for example (Knapp et al. 2016; Bonnet et al. 2017).

Response: Thanks for the comment. Here, we tried to introduce the general function of nitrogen fixation on the global scale. To avoid misleading, we have revised the text to "and contributes to compensate N loss mechanisms such as denitrification and anammox".

**L35:** Only cyanobacterial diazotrophs can be confidently counted by microscopy.

Response: The text has been revised as "Diazotrophic abundance can be estimated from their *nifH* gene copies using qPCR assays (Church et al., 2005). The abundance of some cyanobacterial diazotrophs can also be directly obtained by counting their cells using microscopes."

**L36:** "NifH gene copies"

Response: Corrected.

**L40:** This issue has been thoroughly discussed in (Gradoville et al. 2022), validating the use of nifH gene counts as a means to quantify diazotrophs.

Response: Gradoville et al. (2022) is a regional study in which all the diazotrophs were sampled in two cruises (June 2017 and April 2018) near the Hawaii Islands or along a transect of several hundred kilometers at fixed depths (5 m and 15 m, respectively). Gradoville et al. (2022) described their study as: "... expeditions which each spanned >200 km (Fig. 1). While limited, this reflects the most geographically extensive field comparison of *nif*H:cell among taxa to date."

Hence, although Gradoville et al. (2022) has shown a strong relationship between *nifH* gene counts and diazotrophic (*Crocosphaera*, *Richelia* and *Calothrix*) abundances, it has not sufficiently indicated that this finding is applicable to diazotrophs sampled in other regions or time. Gradoville et al. (2022) partly attributed the large varieties of *nifH*:cell found in Sargent et al., (2016) and White et al. (2018) to potential methodological issues; but they also concluded that "*nif*H is a useful yet imperfect abundance proxy" and urged "future studies report *nif*H:cell and explore the mechanisms controlling this ratio".

We have therefore decided to keep this sentence in our revised manuscript, followed by an introduction of Gradoville et al. (2022): "However, a recent regional study spanning over 200 km in the North Pacific Subtropical Gyre has revealed a robust and statistically significant

correlation between the abundance of the *nifH* gene and cell counts in the UCYN group B (*Crocosphaera*) and heterocystous cyanobacteria (*Richelia* and *Calothrix*) but not in *Trichodesmium* (Gradoville et al., 2022). Nevertheless, the previously observed wide range of *nif*H:cell ratios could be partly attributed to methodological imperfections (Gradoville et al., 2022), which highlights the need for further investigations in this issue."

**L42-47:** Other sources of unbalance should be briefly mentioned here.

Response: Thanks for the suggestion. The text has been revised as:

"One of possible reasons for this imbalance is inaccurate estimation of global marine $N_2$ fixation due to limited spatio-temporal coverage of measurements and questionable $N_2$ fixation assays (White et al., 2020). Another possible reason is the limited knowledge of ecological niches of $N_2$ fixation. Over the last decade, marine habitats beyond the traditionally recognized, well-stratified oligotrophic tropical and subtropical oceans, such as aphotic waters (Bonnet et al., 2013), coastal areas (Tang et al., 2020), subpolar (Sato et al., 2021; Shiozaki et al., 2018) and even polar regions (Shiozaki et al., 2020; Harding et al., 2018), have demonstrated substantial $N_2$ fixation. Other studies have also suggested that non-cyanobacterial diazotrophs (NCDs) may be significant contributors to marine $N_2$ fixation (Shiozaki et al., 2014; Geisler et al., 2020; Turk-Kubo et al., 2022) and may occupy different niches from cyanobacterial diazotrophs (Shao and Luo, 2022)."

**L50:** Diazotroph activity was there before, it is our notion of them that increases, the data available.

Response: Thanks for the comment. The text has been revised and combined into the above paragraph (see response immediately above).

**L56-57:** I don't think that the dataset assembled here covers enough studies comparing microscopy and nifH based comparisons, and I strongly recommend removing this sentence and section 4.2 from the manuscript.

Response: We thank the reviewer for the comment. We believe it is necessary to include the comparisons of cell counts and *nifH* gene copies in the manuscript for two reasons. First, as discussed in our response to the above comment, the relationship between cell counts and *nifH* gene copies is still debatable. Second, a large number of measurements have been conducted on diazotrophic cell counts and *nifH* gene copies, particularly those of *Trichodesmium* and *Richelia* (Fig 11, n = 2377 vs. 3070 for *Trichodesmium*; 898 vs. 1771 for *Richelia*, with more data to be added in the revised manuscript). These comparisons can reveal the overall distributions of cell counts and *nifH* gene copies in specific diazotrophic groups, providing another angle as a meta-analysis that complements previous studies that have directly compared

*nifH* and cell counts using a limited number of samples.

We have also slightly revised the sentence to more accurately describe our analyses: "We also analyzed the discrepancy in $N_2$ fixation assays and compared the observed ranges of *nifH* gene copies and diazotrophic cell abundance using the data available in the database."

**L61:** The $N_2$ fixation rates from Tang et al. 2019 are based on an ARA-$^{15}N_2$ fixation comparison including only 8 data points. This is not robust enough to provide a reliable comparison and downplays the robustness of the $^{15}N_2$-based rates dataset collected here. I strongly recommend removing these from the database and derived basin-scale and global calculations. These may be mentioned as discussion and the Tang paper cited, but not included for quantitative purposes.

Response: Thanks for the comment. Here, we referred to a diazotroph dataset compiled by Tang et al. (2019) and Tang and Cassar (2019) with historical measurements in 2012-2018. There were other in-situ $N_2$ fixation rates (15 $^{15}N_2$-based and 85 ARA-based measurements) measured by Cassar/Tang's own group (Tang et al., 2019; Tang et al., 2020); these data were also collected into our database. The derived $N_2$ fixation rates in Tang et al. (2019) were not collected into our database.

We reconsidered ARA-based measurements of $N_2$ fixation rates and agreed with the reviewer. We have decided not to include the ARA-based data in estimating the global $N_2$ fixation rate, while keep them in the database for those who are interested in using them.

**L72:** Removing NCDs is an error in my opinion. NCDs have recurrently been shown to be dominant in the ocean (Farnelid et al. 2011; Delmont et al. 2018, 2021; Riemann, Farnelid, and Steward 2010) and may impact N cycling decisively (Riemann et al. 2022; Turk-Kubo et al. 2022). I strongly recommend that any nifH gene counts of NCDs are added. The previous database included Gamma A and Cluster III. I don't see a solid reason to remove NCDs from the database at this stage, as evidence of their importance increases.

Response: Thanks for the comment. One of the reasons why we did not include NCD data was the existence of a comprehensive NCD dataset compiled by Turk-Kubo et al. (2022). We have now obtained the agreement from Turk-Kubo to include her NCD dataset in the database (she has agreed to be a coauthor of the revised manuscript). Additional NCD data published in several recent studies have also been added to the revised database. We then accordingly changed the sentence to:

"A recently compiled NCD dataset (Turk-Kubo et al., 2022) including 7385 *nifH* gene copies of mostly studied phylotype Gamma A (Shao and Luo, 2022) and other phylotypes, and several recently published NCD data (Bonnet et al., 2023; Sato et al., 2022; Reeder et al., 2022; Turk-Kubo et al., 2021; Wen et al., 2022; Moore et al., 2018), were included in the database."

**Line 82:** Group-specific $N_2$ fixation rates can only be estimated using single-cell approaches. I'm not sure what approach was followed here to derive specific rates, but these can certainly not be estimated with the data collected here. I would rather recommend the authors to collect all *Trichodesmium*, UCYN-B, DDAs and UCYN-A single-cell rates published, which would be very helpful for the community. See for instance (Foster, Sztejrenszus, and Kuypers 2013;

Foster et al. 2011; Benavides et al. 2017; Bonnet et al. 2016; Filella et al. 2022; Krupke et al. 2015; K. Harding et al. 2018; Mills et al. 2020; K. J. Harding et al. 2022; Benavides et al. 2022).

Response: Thanks for the comment and we are sorry for the confusing. The "different groups" here referred to different size groups. In the original database of 2012, $N_2$ fixation rates in samples with size >10 μm were assigned to *Trichodesmium* and those of smaller sizes were assigned to UCYN. In the revised database, we have corrected and reported them as $N_2$ fixation rates of size groups > 10 μm and < 10 μm, respectively. In some studies, $N_2$ fixation rates of *Trichodesmium* and heterocystous cyanobacteria were estimated by multiplying their cell abundance with their cell-specific $N_2$ fixation rates; we also collected these diazotrophic group-specific data into the new version of the database.

We agree with the reviewer that the cell-specific $N_2$ fixation rates are important and valuable. The cell-specific $N_2$ fixation rates recommended by the reviewer have been collected into the revised database as a new spreadsheet.

The paragraph has been revised as:

"Same as in the original database, the diazotrophic abundance data in Version 2 were grouped into three taxonomic categories: *Trichodesmium*, UCYN, and heterocystous cyanobacteria. The UCYN abundance data were further grouped into UCYN-A, UCYN-B, and UCYN-C, while heterocystous cyanobacterial abundance was grouped into *Richelia* and *Calothrix*. $N_2$ fixation rates were measured for whole seawater samples, for different size groups (> 10 μm and < 10 μm), or specifically for *Trichodesmium* and heterocystous cyanobacteria. When whole-water $N_2$ fixation rates were not reported, total $N_2$ fixation rates were calculated as the sum of the $N_2$ fixation rates of available groups. Additionally, 392 data of cell-specific $N_2$ fixation rates were also collected to Version 2."

Tables 2 and 4: Many studies are missing in this table, some include (Benavides et al. 2014, 2021; Saulia et al. 2020; Henke et al. 2018; Bonnet et al. 2018; Gradoville et al. 2017; Moreira-Coello et al. 2017; Wilson et al. 2019). Also, in the table some studies are listed as not including counts of some diazotrophs, which needs correction (e.g. Bombar 2011 and Bonnet 2015, 2019 did have qPCR counts). Please revise all these publications thoroughly and correct accordingly.

Response: We thank the reviewer for identifying missing datasets and parameters. We have checked datasets suggested by the reviewer, and have added those parameters collected by this database. We have identified more missing datasets and have added them to the revised database. The new datasets added in the revised database are as follows:

**$N_2$ fixation rates:**

(1) Benavides et al. (2014), Journal of Marine Systems

(2) Benavides et al. (2017), Scientific Reports

(3) Benavides et al. (2021), ISME Comm.

(4) Benavides et al. (2022), ISME J.

(5) Bonnet et al. (2023), ISME J.

(6) Cerdan-Garcia et al. (2022), ISME J.

(7) Foster et al. (2022) , ISME J.

(8) Gradoville et al. (2017), Limnol. & Oceanogr.

(9)  Harding et al. (2022), Nature Comm.

(10) Jiang et al. (2023), J. Geophy. Res.

(11) Kittu et al. (2023), Gobal Biogeochemical Cycles

(12) Landou et al. (2023), Limnol. & Oceanogr.

(13) Messer et al. (2021), PeerJ

(14) Mills et al. (2020), ISME J.

(15) Moreira-Coello et al. (2017), Front. in Mar. Sci.

(16) Rase et al. (2013), Marine Ecology Progress Series

(17) Sato et al. (2022), JGR Biogeosciences

(18) Saulia et al. (2020), Front. in Mar. Sci.

(19) Selden et al. (2021), Limnol. & Oceanogr.

(20) Singh  et al. (2017), Geophy. Res. Let.

(21) Singh  et al. (2019), Continental Shelf Res.

(22) Turk-Kubo  et al. (2021), ISME Comm.

**Cell counts:**

(1)  Estrada et al. (2016), PLOS one

(2)  Mompean et al. (2016), J. of Phyto. Res.

(3) Tenório et al, (2018),  Aquat. Micro. Ecol.

**NifH gene copies:**

(1) Bonnet et al. (2023), ISME J.

(2) Cerdan-Garcia et al., (2021), ISME J.

(3) Jiang et al. (2023) JGR Biogeosciences

(4) Bonnet  et al. (2015), Gobal Biogeochemical Cycles

(5) Cabello  et al. (2020), Journal of Phycology

(6) Messer et al. (2021), PeerJ

(7) Mills et al. (2020), ISME J.

(8) Sato  et al. (2022), J. Geophy. Res.

(9) Saulia et al. (2020), Front. in Mar. Sci.

(10) Selden et al. (2021), Limnol. & Oceanogr.

(11) Selden  et al. (2022), Front. in Mar. Sci.

(12) Turk-Kubo et al. (2021), ISME Comm.

L104: The ARA method is rarely used nowadays

Response: The reviewer was correct. We have revised the texts to:

"The commonly used methods for marine $N_2$ fixation rates include $^{15}N_2$ assimilation and acetylene reduction assay (Mohr et al., 2010; Montoya et al., 1996). In the last decade, most samples were measured using $^{15}N_2$ assimilation methods."

L106-107: The ARA to $N_2$ fixation ratio is highly variable (Mulholland et al. 2006; Benavides et al. 2011; Wilson et al. 2012)

Response: Thanks for this comment. We have added the previously reported range of the conversion factor between acetylene reduction and $N_2$ fixation:

"The acetylene reduction assay estimates gross $N_2$ fixation rates indirectly from the reduction of acetylene to ethylene. Theoretical conversion factors of 3:1 or 4:1 has been used to convert acetylene reduction rates to $N_2$ fixation rates (Postgate, 1998; Capone, 1993; Wilson et al., 2012). However, a wide range of conversion factors from 0.93 to 56 has been reported (e.g., Mague et al., 1974; Graham et al., 1980; Montoya et al., 1996; Capone et al., 2005; Wilson et al., 2012)."

L110: Many other factors affect this difference, including acetylene gas impurity, Bunsen dissolution coefficient, etc.

L112: This is not true. The $^{15}N_2$ method is much more sensitive, does not require biomass preconcentration (biomass is concentrated during filtration, after the incubation), and requires longer incubations for enough tracer to be detectable in biomass. ARA is usually done in 3-4 h incubations and requires biomass pre-concentration to reach detectable signal (Staal et al. 2007; Benavides et al. 2011).

Response: We thank the reviewer for the above two related comments regarding comparing $^{15}N_2$ assimilation and the acetylene reduction assay. We have incorporated the reviewer's comments and corrections and modified the texts as follows:

"Overall, the $^{15}N_2$ assimilation method only measures the fixed N in particulate forms and ignores the N that is fixed but then excreted by diazotrophs during incubation, which, however, can theoretically be counted by the acetylene reduction assays (Mulholland, 2007). Compared to the $^{15}N_2$ assimilation method, the acetylene reduction assay is easier to conduct and needs a shorter incubation time. However, in addition to the uncertainty in converting ethylene production to $N_2$ fixation, the purity of acetylene gas, trace ethylene contamination and the Bunsen gas solubility coefficient of produced ethylene can also affect the accuracy of estimating $N_2$ fixation rates (Giller, 1987; Hardy et al., 1973; Flett et al., 1976; Hyman and Arp, 1987). Acetylene used in the assay can even impact the metabolic activities of diazotrophs (Giller, 1987; Hardy et al., 1973; Flett et al., 1976). Moreover, the acetylene reduction assays need to pre-concentrate cells for signal detection when diazotrophic biomass is low, which can damage cells during filtration and cause underestimated $N_2$ fixation rates (e.g., Capone et al., 2005; Staal et al., 2007; Bhavya et al., 2019; Barthel K-G, 1989). In contrast, the $^{15}N_2$ assimilation method has a higher sensitivity and does not require the cell pre-concentration before incubations."

L120: Wannicke et al. say the opposite of Mohr and Grosskopf.

L123: What White et al. say is that the bubble release method is the most reliable and recommended by the diazotroph research community, with the elimination of rate underestimation benefits overcoming the very unlikely burdens of contamination. This should be corrected in L274-275 as well.

Response: Thanks for pointing out these two mistakes. We have carefully revised whole section:

"The original $^{15}N_2$ assimilation method involved bubbling $^{15}N_2$-labelled gas. However, this method was later found to be inadequate for reaching complete solubility equilibrium over a

short incubation time, resulting in significant underestimations of $N_2$ fixation rates (Mohr et al., 2010; Großkopf et al., 2012). To address this issue, the $^{15}N_2$ dissolution method was employed, which involved pre-preparing $^{15}N_2$-enriched seawater to maintain a constant $^{15}N_2$ %atom throughout the incubation (Mohr et al., 2010), similar to the method described in Glibert and Bronk (1994). However, the $^{15}N_2$ dissolution method may introduce contaminants such as nutrients or trace metals, which can alter the diazotrophic activities and impact the accuracy of $N_2$ fixation measurements (Klawonn et al., 2015). Additionally, the pH and other chemical properties of the inoculum may be altered during its preparation, further affecting the measurement of $N_2$ fixation. Despite these limitations, the $^{15}N_2$ dissolution method remains the predominant assay for measuring $N_2$ fixation rate due to its ability to satisfy the fundamental assumption of constant $^{15}N_2$ %atom over the incubation.

More recently, a modified bubble method, known as the "bubble release method', has been proposed as an alternative to the $^{5}N_2$ dissolution method (Klawonn et al., 2015; Chang et al., 2019; White et al., 2020). This method involves adding $^{15}N_2$ gas to the incubation botttles and mixing for less than 15 minutes to facilitate $^{15}N_2$ equilibration, followed by releasing the gas bubbles and replacing them with $^{15}N_2$-unenriched seawater samples. Compared to the original bubble method, the bubble release method ensures a uniform $^{15}N_2$ %atom throughout the incubation. Moreover, it causes less invasion for the incubation matrix than the $^{15}N_2$ dissolution method and causes less interference with the incubation matrix. However, the agitation of incubation bottles required to stimulate gas dissolution may affect diazotrophs, such as *Trichodesumium* colonies (Wannicke et al., 2018; White et al., 2020). Moreover, the bubble release method results in increased spatial and labor expenditures, thereby impeding its widespread implementation (White et al., 2020). "

The first sentence of 4.1has also been revised as:

"To date, the discrepancy in $N_2$ fixation rates estimated using the original $^{15}N_2$ bubble method, the $^{15}N_2$ dissolution method, and the $^{15}N_2$ bubble release method remains unclear."

L150: There are 4 UCYN-A sublineages (Farnelid et al. 2016).

Response: We have corrected the text as follows:

"Four sublineages of UCYN-A, including UCYN-A1, UCYN-A2, UCYN-A3, and UCYN-A4, have been identified, with the clade UCYN-A1 sharing the same genome as previously targeted UCYN-A (Thompson et al., 2014; Farnelid et al., 2016)."

L328: UCYN-A has been found in symbiosis with other eukaryotic algae (Zehr et al. 2016)

Response: The text has been revised as:

"The conversion factor for UCYN-A is also updated because it has been found to live

symbiotically with prymnesiophytes, coccolithophores or other uncultured eukaryotic algae (Zehr et al., 2016; Thompson et al., 2012)."

L370: The first version of the database included all the authors that had contributed to its construction with their seagoing expeditions, laboratory analyses and publications. I humbly find it sad and somewhat unfair that this is not the case in this update.

Response: We highly value the reviewer's comment and agree proper coauthorship credit is important for all contributors. Initially, our plan to publish an updated global marine diazotrophic database was too simplistic and lacked careful thinking. Since the first global marine diazotrophic database was published in 2012, our group has continuously updated the database with newly published data. In recent years, we have received numerous requests for an updated version of the database, which prompted us to consider publishing it for wider usage.

We have extended an invitation to all PIs to join us as coauthors of the manuscript. Approximately 80 PIs have accepted and have become coauthors of the revised manuscript. We believe that this effort not only provides proper credit all involved contributors but also improves the quality and completeness of the database and accompanying paper.

**Reviewer #2:**

This manuscript by Shao and Xu et al. describes an updated version 2 of the global oceanic diazotroph database. It build upon the previous version by adding additional measurements of marine diazotrophic abundance, $N_2$ fixation rates, microscopic and qPCR-based diazotropic abundance. The spatial coverage significantly improved most notably in the Indian Ocean. The newly revised estimate for global $N_2$ fixation rate is significantly higher (+123 Tg N yr$^{-1}$, almost doubled) when calculating using a standard arithmetic mean, although surprisingly the geometric mean did not significantly change. A brief analysis and discussion of the $^{15}N_2$ bubble vs. dissolution indicated a potential general underestimation from the bubble method particularly at high rates, however noting the comparison of samples were from different times so it is not a formal error analysis (which the authors acknowledge). The database is available to download from the provided link in the abstract.

Overall, I find this to be an important update to the database mainly due to the significant increase in included measurements and spatial coverage. The database is transparent and mostly well described. The analysis and first preliminary quantification of the $^{15}N_2$ bubble vs. dissolution is also an important contribution. Perhaps some additional details/analysis could be provided (see comments below), but additional analyses can also be performed independently by users who download the data for their specific interest. There is one important aspect that needs additional clarification in my view before I would endorse this manuscript for publication (global $N_2$ fixation rate calculation, see below).

-Christopher Somes
GEOMAR Helmholtz Centre for Ocean Research Kiel

Response: We thank Dr. Somes for his positive and constructive comments, which have helped us improve the quality of this paper substantially. Please see our responses below.

**Major Comment: Global N2 fixation calculation description**

Since this paper will likely often be cited for revising the global $N_2$ fixation rate significantly upwards, the description of this calculation should be more transparent and comprehensive:

line 266 (Table 5 caption): "Data are first binned to 3x3 grids…"

This needs to be better described. For example, was there any type of interpolation method used or simple averaging of all measurements in each bin? It would be interesting to know what percentage of bins in each ocean basin has data coverage. How do you define the Southern Ocean region and is that area removed from the other southern regions?

How was the vertical coordinate handled? Is it evenly spaced or according to the depths ranges in Figure 7?

It is not clear to me how the "Areal sum" calculation was made based on the "Mean $N_2$ fixation rate" (Table 5). Does the "Mean $N_2$ fixation" rate include all measurements or only the "Depth-integrated $N_2$ rates", which requires 3 measurements in the vertical? If the vertical coordinate is uneven, do measurements that get binned into a larger volume in larger deeper layers have more weight on the depth-integrated rate than shallower layers?

When calculating the "Areal Sum", do you assume that the "Mean N2 fixation rate" extrapolates across the entire region or do you only consider the area of the bins that have data coverage? For example, the Indian Ocean has about 36% of the bins compared to the South Pacific. Therefore I was expecting a much larger decrease when calculating the Areal Sum relative to the Mean $N_2$ fixation rate for the Indian Ocean compared to the South Pacific. However this relative decrease is quite subtle in Table 5 between these regions. I acknowledge there is no truly perfect way to estimate a global ocean $N_2$ fixation rate with the current coverage, but all of the assumptions and details that go into the calculation should be specifically stated and described.

Response: Here we respond Dr. Somes's general comments regarding the description of calculating the global marine $N_2$ fixation rate.

We followed the procedure used in the previous database paper (Luo et al., 2012) to estimate the global marine $N_2$ fixation rate. However, as reminded by the reviewer, we should describe the method in this paper, which has been added in section 2.2 in the revised manuscript. Please be aware that we have also decided that only the arithmetic means should be used and have removed the geometric means in estimating of the global marine $N_2$ fixation rate (see our response below).

Here are some quick answers to the reviewer's questions:

The data used in the estimation is the depth-integrated $N_2$ fixation rates integrated from surface to the depth of the deepest data (up to 200 m; see section 2.1). The measurements in each vertical profile were linearly interpolated, which was not clearly described in the original manuscript. We have revised the sentence (in Section 2.1) to: "A profile was integrated from sea surface down to the deepest datum measured. The measurements within the profile were interpolated linearly along the depth, with the shallowest datum representing the level between the sea surface and that datum."

The arithmetic mean of the data in each bin was calculated first, and then these means in each basin were averaged further.

The Southern Ocean was defined as the area south of 45°S and was excluded from other basins when calculating the global rates. Additionally, due to very limited data coverage, the $N_2$ fixation rates of the Southern and Arctic Oceans have been excluded from the estimation of global marine $N_2$ fixation.

The percentage of bins with data coverage in each ocean basin have been added in the revised table.

When calculating the areal sum, we extrapolated the mean $N_2$ fixation rate of each basin across the entire basin, i.e., the mean $N_2$ fixation rate was multiplied by the area of each basin. We have listed the areas of every ocean basin in the table.

The description of the methods in calculating global marine $N_2$ fixation rate was added to Section 2.2:

"A first-order estimate of global marine $N_2$ fixation rate was conducted using data from this database. Total $N_2$ fixation rates were estimated for ocean basins including the North Atlantic, South Atlantic, North Pacific, South Pacific, Indian, Arctic, Southern Oceans, and the Mediterranean Sea. The Southern Ocean was defined as the region between 45°S and Antarctica. Due to considerable uncertainties associated with the acetylene reduction method, only $N_2$ fixation rates measured using the $^{15}N_2$ assimilation methods were used in this estimation. To increase data coverage, $N_2$ fixation rates measured using the original $^{15}N_2$ bubble method were included in the estimation, although it is acknowledged that these data may underestimate of the global marine $N_2$ fixation rate. First, the arithmetic means of depth-integrated $N_2$ fixation rates in each 3°×3° bin were calculated. Second, these binned means were further averaged in each ocean basin to obtain the average $N_2$ fixation rate, which was then multiplied by the basin area to estimate the total $N_2$ fixation rate for that basin. Finally, the global marine $N_2$ fixation rate was calculated by summing the basin rates, except for those of the Southern and Arctic Oceans due to limited spatial coverage in these two basins."

The authors do not give much context on interpreting the geometric vs. arithmetic mean despite that it is mentioned multiple times throughout the manuscript and gives a significantly different result. From what I understand, geometric mean is less sensitive to the high-end rates compared to arithmetic mean. Does this mean that most of the increase in the arithmetic mean is driven by newly included high-end rates? It would be valuable to know how much of the large increase in the arithmetic areal sum is driven by additional spatial coverage versus generally higher rate values. I would suggest to include a histogram of the previous version in one of the supplementary figures for comparison. If newly included rate values tend to be significantly higher, it would be interesting to know how much of that may be attributable to growing numbers of the dissolution method compared to bubble method (i.e. based on Figure 10).

Response: Dr. Somes was correct in interpreting geometric versus arithmetic means. As our $N_2$ fixation data were approximately log-normally distributed, their geometric mean is near the most frequently observed rate (i.e., the peaks of the distribution of the log-transformed $N_2$ fixation rates). Meanwhile, high $N_2$ fixation rates do occur and should be included in estimating

global $N_2$ fixation. Hence, the arithmetic means should be used in estimating global $N_2$ fixation if sufficient data have been sampled. However, if the number of samples is small, some occasionally observed high $N_2$ fixation rates can greatly elevate the estimated global rate while we cannot know if these high $N_2$ fixation rates are typical. This was the reason that we c presented both the geometric and arithmetic means of $N_2$ fixation rate in our 2012 paper and in the initial submission of the current manuscript.

With much more measurements becoming available, we have decided to only calculate arithmetic means of $N_2$ fixation in the revised manuscript. By this way, it can avoid the confusion of some readers in choosing proper estimations of total $N_2$ fixation rate of ocean basins and the global ocean. Additionally, $N_2$ fixation rates in most basins approximately follow log-normal distributions (except several data in the Indian Ocean, see below), indicating that most high $N_2$ fixation rates are acceptable.

In the initial version of this manuscript, the increase in the arithmetic-mean-based estimation of global marine $N_2$ fixation, compared to that in Luo et al. (2012), was caused mostly by (1) the nearly doubled rate in the South Pacific Ocean and (2) the high rate in the Indian Ocean for which the estimation of $N_2$ fixation was not made in Luo et al. (2012).

The much higher estimation of $N_2$ fixation rate in the South Pacific Ocean mostly attributes to the high rates sampled in the western South Pacific (new Fig. S9 attached below) where $N_2$ fixation rates were under sampled in the 2012 database. Overall, the $N_2$ fixation rate data in the South Pacific Ocean in the new database were close to log-normal distribution (new Fig. 10a attached below) and we used all of them to estimate the basin-wide rate.

[Figure]

**Figure S9.** Depth-integrated $N_2$ fixation rates in the South Pacific Ocean. Filled circles represent the added data in the new database. Empty diamonds represent the data existing in the original database.

[Figure]

**Figure 10.** Histogram of depth-integrated $N_2$ fixation rates in (a) South Pacific and (b) Indian ocean.

The high estimation of the $N_2$ fixation rate of the Indian Ocean, however, was mainly caused by 14 extremely high measurements of $N_2$ fixation (> 1000 $\mu mol\ N\ m^{-2}\ d^{-1}$) sampled near the coast of western India. These high rates are much higher than all the other measurements in the Indian Ocean (Fig. 10b attached above) and are unlikely typical in the Indian Ocean. We have decided to remove them in our revised estimation, which greatly reducing the estimation of the $N_2$ fixation of the Indian Ocean from 98 Tg N $yr^{-1}$ to 16 Tg N $yr^{-1}$.

**Minor Comments:**

line 84 and data file: Metadata
In the data file, the meta data are titled "Surface …", yet they are associated with a specific depth, so are they really surface? I am used to seeing chlorophyll expressed by volume not area.

Response: Thank you for pointing out the mistakes. In the volumetric spreadsheets, the meta data were measured at the same depths as the diazotrophic data, and the word "surface" has been deleted from their names. Similarly, the chlorophyll concentration in the volumetric datasheets should be in unit of mg $m^{-3}$, which has been corrected in the revised database.

In the depth-integrated datasheet, considering the large vertical variations of environmental parameters and chlorophyll, we collected their near-surface values. We have corrected their names to "Near-surface xx".

lines 127-129: daily vs. daytime vs. nighttime normalization
I am still a little confused about the time normalization with this brief description. If the incubation is only performed during the day, you convert hours to day by 12 hr/day which assumes no rates at night? I see that incubation hours vary a lot and in some cases not a multiple of 12 hours or 1 day. Perhaps you can describe how individual studies typically convert to a daily rate depending on the incubation period. Would it make more sense to multiply by the daytime of each location during the time of sampling instead of assuming 12 hours?

Response: We have changed the way to convert hourly N$_2$ fixation rates to daily rates according to the suggestions of Dr. Somes. The method has now been descried with more details:

"The majority of N$_2$ fixation rates (6766) were reported on a daily basis, while 1097 samples reported hourly N$_2$ fixation rates. We also converted these hourly rates to daily rates. In each sample, hourly N$_2$ fixation rates for daytime and nighttime were first multiplied by the respective local durations of day and night, and were then added to obtain the daily rate. However, 777 samples had hourly N$_2$ fixation measured only during daytime, which could have led to an underestimation of daily N$_2$ fixation because nighttime N$_2$ fixation was not included. It is important to note that diel cycles of N$_2$ fixation vary among samples and/or diazotrophic groups, and thus, substantial errors may be introduced when extrapolating these hourly N$_2$ fixation to daily rates (White et al., 2020)."

Table 5.: "n" is missing in Indian Ocean

Response: Corrected.

Figure 7:
Why do you choose geometric mean over the more commonly used arithmetic mean in this figure? Does it look significantly different if you use arithmetic means?

Response: The general spatial pattern of N$_2$ fixation was similar when using either geometric or arithmetic means, except for some high arithmetic means. Nevertheless, we have discarded the usage of geometric means in global marine N$_2$ fixation rates (see our response above); we then have changed to present arithmetic means of N$_2$ fixation in the revised manuscript (Fig. 7 attached below).

[Figure]

**Figure 7.** N$_2$ fixation rates in version 2 of the database. The panels show **(a)** depth-integrated

data and volumetric data in (**b**) 0–5 m, (**c**) 5–25 m, (**d**) 25–100 m, (**e**) 100-200 m, and (**f**) below 200 m. For a clear demonstration, data are binned to 3° × 3° grids and arithmetic means in each bin are shown. Zero-value data are denoted as black crosses.

I would be interested to see a euphotic vs. aphotic depth-integrated rate. I am curious how much the generally low to moderate rates occurring below 100 meters contribute to the total depth-integrated rate since they can occupy more volume. Perhaps adding a < 100m and >100m panel would be useful? At what depths are the deepest N2 fixation measurements?

Response: We have generated an averaged vertical profile of $N_2$ fixation rates from sea surface to the deepest (4000 m; Hallstrøm et al., 2022) $N_2$ fixation measured (Figure attached below). In the revised manuscript, we will present this figure and will compare and discuss the $N_2$ fixation rates above and below 200 m.

[Figure]

**Figure.** Vertical profile of $N_2$ fixation rates in the global ocean. Blue circles represent the reported $N_2$ fixation rates, and the red circles and error bars are the means and standard errors in depth intervals marked in the y-axis. The x-axis is in a log scale to better show the distribution of low $N_2$ fixation rates.

Section 4.1/Figure 10:
As mentioned above, I think is a useful first investigation into methodological uncertainties on N2 fixation rates. Is there enough data coverage to do a similar analysis for acetylene reduction?

Response: We thank Dr. Somes to recognize the value of our analyses. We compared the $N_2$ fixation rates measured using the $^{15}N_2$ tracer methods and from the acetylene reduction (ARA) method. However, there were too limited pairs of data available (n=28 and 24 for ARA vs. $^{15}N_2$ dissolution method and ARA vs. the original method, respectively) (see Figure S10 attached blow) to be included in the manuscript.

[Figure]

**Figure S10.** Comparison of measured $N_2$ fixation rates using the $^{15}N_2$ tracer methods and acetylene reduction (ARA) assays. (a) The ARA versus the $^{15}N_2$ dissolution; and (b) the ARA versus the original $^{15}N_2$ bubble method. The pink dots are measurements. The fitted results using the generalized additive model (GAM) and confidence intervals are represented by the red solid and the dashed black lines, respectively. The blue lines are the 1:1 ratio of the measurements using the compared methods.

Please be also noted that there were mistakes when pairing $^{15}N_2$ dissolution and $^{15}N_2$ bubbling measurements in the original manuscript, which have been corrected in the revised manuscript (new Fig. 11 attached below). The texts have been revised as follows:

"We also compared mean $N_2$ fixation rates at the same locations (using 3° × 3° grids), depth intervals (as defined in Fig. 7b–f) and months, using the original $^{15}N_2$ bubble method, the $^{15}N_2$ dissolution method, and the acetylene reduction assays, although the samples measured by these methods were not identical. The results showed that, in 68% of cases (**Fig. 11**), the $^{15}N_2$ dissolution method produced higher rates than the original $^{15}N_2$ bubble method. Furthermore, our analysis using the generalized additive model (GAM) indicated that the underestimation by the original $^{15}N_2$ bubble method tended to be exaggerated under high $N_2$ fixation (> ~5 μmol N m$^{-3}$ d$^{-1}$) (**Fig. 11**). This can be explained by the gas equilibrium time (Jayakumar et al., 2017; Mohr et al., 2010). Under low $N_2$ fixation, the original $^{15}N_2$ bubble method can provide sufficient dissolved $^{15}N_2$ regardless of whether the gas reaches equilibrium. However, under

high $N_2$ fixation, the method cannot fulfill the requirement of dissolved $^{15}N_2$, resulting in relatively large underestimations.

We also used the same procedure to compare the $N_2$ fixation rates measured using ARA and the $^{15}N_2$ tracer methods. However, we had insufficient pairs of data available (n = 28 and 24 for ARA versus the $^{15}N_2$ dissolution or the original $^{15}N_2$ bubble method, respectively) for a robust comparison (Fig. S10)."

[revised manuscript text omitted]

---

## Author Response (AR1)

We have taken all the comments of the reviewers into account in the revision; replies to each of the comments are provided below in blue fonts.

In response to one of the reviewers' suggestions, we have carefully considered how to appropriately acknowledge the contributions of the data contributors in our revised manuscript, particularly in relation to this database paper. Furthermore, we have strived to ensure the high quality and completeness of both the database and the accompanying paper. To achieve these goals, we extended an invitation to all lead authors of the published data, inviting them to join us as coauthors of the manuscript.

We are delighted to share that nearly 90 lead authors have accepted our invitation and have become coauthors of the revised manuscript. Their involvement has been invaluable, as they have not only revised the manuscript, but also contributed to data analyses, providing unpublished data, and identifying missing historical datasets. The addition of these co-authors has significantly elevated the overall quality of the database and the manuscript, while also exemplifying the collaborative efforts of the entire diazotroph survey community.

**Reviewer #1:**

Shao et al. present an update of the diazotroph database published in 2012 https://essd.copernicus.org/articles/4/47/2012/

The new version adds up data published between 2012 and 2023, including volumetric and depth-integrated $N_2$ fixation rates, diazotroph microscope counts and nifH gene counts. This new version also discusses microscope-nifH count comparisons. While this update is valuable for the community as a tool for comparison and contextualization of diazotrophy studies, it fails to account for many diazotrophy studies published between 2012 and 2023. The text has several misinterpretations that need correction. The new version also includes N2 fixation rates proxied with other methods (ARA). I think this is a major problem, since these rates are not currently solidly comparable and downplay the robustness of the database. The manuscript also eliminates nifH gene counts from non-cyanobacterial diazotrophs (NCDs), which is another major issue since NCDs are considered to be outnumber cyanobacterial diazotrophs in the ocean. Finally, the diazotroph microscopy count versus nifH gene count conversion discussion does not seem appropriate here, since very few of the papers listed have compared these approaches on a same given sample, and the issue has been discussed thoroughly in other publications by specialists.
In all, while I acknowledge the effort and usefulness of this manuscript, I advise major revisions as detailed in the comments below.

Response: We thank the reviewer for very constructive and thorough comments. We particularly appreciate the reviewer's suggestion to include PIs of the data sources as coauthors, and we have adopted this suggestion. We are pleased to report that nearly 90 lead authors have agreed to join this collaborative effort. By doing so, we believe that this paper will not only showcase the significant impact of collective research efforts in $N_2$ fixation, but also the completeness of the database and the quality of the paper can be improved substantially. The new coauthors have identified missing datasets and provided additional comments that will further enhance the quality of the paper. Other general comments have also been addressed:

(1) We have decided not to include the ARA-based data in estimating the global $N_2$ fixation

rate, but we have included them in the database for those who are interested in using them.

(2) The NCD data have been added to the database as an additional datasheet.

(3) We have decided to keep the comparison of *nifH* gene copies and diazotrophic cell counts in the paper, and we invite you to review our response to the related comments.

Once again, we thank the reviewer for the valuable feedback, which has helped us to improve the quality of our manuscript considerably.

**L28:** $N_2$ gas is not inert to diazotrophs.

Response: We have changed the text to "Dinitrogen ($N_2$) fixation is a process carried out by select prokaryotes (diazotrophs) capable of converting $N_2$ gas, which is not usable by most organisms, into bioavailable nitrogen (N)."

**L31:** The balance between N loss/gains in the ETSP has been widely demonstrated to be false in several publications after that of Deutsch et al., see for example (Knapp et al. 2016; Bonnet et al. 2017).

Response: Thanks for the comment. Here, we tried to introduce the general function of nitrogen fixation on the global scale. To avoid misleading, we have revised the text to " Globally, $N_2$ fixation serves to compensate, at least partially, for fixed N removed via denitrification and anammox (Deutsch et al., 2007; Gruber, 2019)".

**L35:** Only cyanobacterial diazotrophs can be confidently counted by microscopy.

Response: The text has been revised as "Diazotroph abundance can be estimated from *nifH* gene copies using qPCR assays (Church et al., 2005) or droplet digital PCR (ddPCR) (Gradoville et al., 2017). The abundance of some cyanobacterial diazotrophs can also be obtained by counting them directly using microscopy-based techniques and in some cases flow cytometry."

**L36:** "NifH gene copies"

Response: Corrected.

**L40:** This issue has been thoroughly discussed in (Gradoville et al. 2022), validating the use of nifH gene counts as a means to quantify diazotrophs.

Response: Gradoville et al. (2022a) is a regional study in which all the diazotrophs were sampled in two cruises (June 2017 and April 2018) near the Hawaii Islands or along a transect of several hundred kilometers at fixed depths (5 m and 15 m, respectively). Gradoville et al.

(2022a) described their study as: "... expeditions which each spanned >200 km (Fig. 1). While limited, this reflects the most geographically extensive field comparison of *nif*H:cell among taxa to date."

Hence, although Gradoville et al. (2022a) has shown a strong relationship between *nifH* gene counts and diazotrophic (*Crocosphaera*, *Richelia* and *Calothrix*) abundances, it has not sufficiently indicated that this finding is applicable to diazotrophs sampled in other regions or time. Gradoville et al. (2022a) partly attributed the large varieties of *nifH*:cell found in Sargent et al., (2016) and White et al. (2018) to potential methodological issues; but they also concluded that "*nif*H is a useful yet imperfect abundance proxy" and urged "future studies report *nif*H:cell and explore the mechanisms controlling this ratio". Both Drs. Gradoville and White are now coauthors of this paper; they have agreed this interpretion of their paper.

We have therefore decided to keep this sentence in our revised manuscript, followed by an introduction of Gradoville et al. (2022a): " However, a recent regional study spanning over 200 km in the North Pacific Subtropical Gyre has found a statistically significant linear correlation between the abundances of the *nifH* gene and cell counts in the UCYN-B (i.e., *Crocosphaera*) (linear slope = 1.82) and heterocystous cyanobacteria (*Richelia* and *Calothrix*; linear slope from 1.51-2.58) but not in *Trichodesmium* (Gradoville et al., 2022b). A recent discussion highlighted the influence of the uncertainty in gene copy conversion to biomass and the need for further investigations on how to best take advantage of gene copy data for global diazotroph biogeography modelling purposes (Meiler et al., 2022; Zehr and Riemann, 2023); however, there is an agreement that quantifying gene counts is a powerful tool for studying marine diazotroph distributions (Meiler et al., 2023; Zehr and Riemann, 2023). Meiler et al., (2023) proposed a number of topics of study for this field moving forward; Gradoville et al. (2022) concluded that "we hope that future studies report *nifH*:cell and explore the mechanisms controlling this ratio." Both gene based and microscopy cell counts have innate biases, which should be elucidated in future studies. "

**L42-47:** Other sources of unbalance should be briefly mentioned here.

Response: Thanks for the suggestion. The text has been revised as:

"While the overestimation of the N losses cannot be ruled out, one of possible reasons for this imbalance is the inaccurate estimation of global marine $N_2$ fixation due to limited spatio-temporal coverage of rate measurements and the different methods employed in $N_2$ fixation assays (White et al., 2020). Another possible reason is the limited knowledge of ecological niches of $N_2$ fixing organisms. Over the last decade, the realm of marine $N_2$ fixation has been expanded to include numerous non-paradigmatic habitats. Coastal (Mulholland et al., 2012; Bentzon-Tilia et al., 2015; Mulholland et al., 2019; Tang et al., 2020; Turk-Kubo et al., 2021), subpolar (Sato et al., 2021; Shiozaki et al., 2018), and even polar ocean regions (Blais et al.,

2012; Sipler et al., 2017; Harding et al., 2018; Shiozaki et al., 2020) have demonstrated $N_2$ fixation. Notably, $N_2$ fixation in aphotic waters remains debated (Bonnet et al., 2013; Farnelid et al., 2013; Selden et al., 2021; Rahav et al., 2013; Hamersley et al., 2011; Benavides et al., 2018). Other studies have also suggested that NCDs may be significant contributors to marine $N_2$ fixation (Shiozaki et al., 2014; Turk-Kubo et al., 2022; Geisler et al., 2020; Delmont et al., 2021; Karlusich et al., 2021; Bombar et al., 2016; Moisander et al., 2017) and may occupy different niches from cyanobacterial diazotrophs (Shao and Luo, 2022)."

**L50:** Diazotroph activity was there before, it is our notion of them that increases, the data available.

Response: Thanks for the comment. The text has been revised and combined into the above paragraph (see response immediately above).

**L56-57:** I don't think that the dataset assembled here covers enough studies comparing microscopy and nifH based comparisons, and I strongly recommend removing this sentence and section 4.2 from the manuscript.

Response: We thank the reviewer for the comment. We believe it is necessary to include the comparisons of cell counts and *nifH* gene copies in the manuscript for two reasons. First, as discussed in our response to the above comment, the relationship between cell counts and *nifH* gene copies is still an issue undergoing discussions. Second, a large number of measurements have been conducted on diazotrophic cell counts and *nifH* gene copies, particularly those of *Trichodesmium* and *Richelia* (Fig 16, n = 3572 vs. 3098 for *Trichodesmium*; 1309 vs. 1914 for *Richelia*). These comparisons can reveal the overall distributions of cell counts and *nifH* gene copies in specific diazotrophic groups, providing another angle as a meta-analysis that complements previous studies that have directly compared *nifH* and cell counts using a limited number of samples.

We have also slightly revised the texts to more accurately describe our analyses. In the end of Introduction:

"In light of the aforementioned concerns of *nifH*:cell and various $N_2$ fixation methods (*see* **Section 2.3**), we also discuss the significance of employing different methodological approaches to estimate $N_2$ fixation rates and abundance metrics. We use the data available in the database to analyze the discrepancies between $N_2$ fixation rates using $^{15}N_2$ bubble and dissolution methods, and compare the observed ranges of *nifH* gene copies and diazotrophic cell abundance."

At end of the section for analyzing nifH abundance and cell counts, we have also added a full paragraph and a summary table for discussing this issue:

The application of qPCR assays for *nifH* based abundance (DNA) and expression (RNA) emerged as a critical step forward in our understanding of the distribution, abundance, and physiology (e.g., expression of *nifH*) of diazotrophs (Short and Zehr, 2005; Zehr and Riemann, 2023). Until then, estimating the abundances of diazotrophs were limited to those that could be identified by microscopy, e.g., *Trichodesmium*, heterocystous cyanobacteria (e.g., *Richelia, Calothrix, Anabaena, Nodularia, Aphanizomenon*), and some unicellulars (e.g., *Cyanothece*, later *Crocosphaera*). Thus, qPCR enabled the study of diazotrophic targets (and their activity) without the need to microscopy identify them, which came later as some diazotrophs would (and still) require application of FISH techniques for identification (Biegala and Raimbault, 2008). Additionally, qPCR allowed the study of *in situ* activity (gene expression) by diazotrophs without the need for cultivation. Although beyond the scope of the work presented here, important considerations should be taken into account when using microscopy and qPCR datasets (Table S3), for example, in application to biogeochemical models (Meiler et al., 2023).

**Table S3**. Summary of a few considerations for application and interpretation of qPCR and microscopy counting for enumeration and activity (RNA) of diazotrophs.

| Consideration | Comment |
| --- | --- |
| Cell identity | *Microscopy:* Cross-comparison of cell counts can be difficult as training and experience varies. |
| Patchy distribution, low abundance | *Both methods:* collection of samples (volumes, depths) are dependent on logistics; collection strategies can vary: size fractionation, gravity filtration (microscopy), etc.
*Microscopy*: potential to underestimate/overestimate if the minimum number of cells is not enumerated.
*qPCR:* potential to underestimate if targets are below detection of assay (1-10 copies). |
| Dead or moribund cells | *Both methods* do not distinguish vitality, thus potential to overestimate. |
| Primer design | *qPCR*: potential to overestimate if primers cross-react with non-targets; potential to underestimate if primers are too specific to a limited/unknown micro- diversity. |
| Polyploidy | Some bacteria, including data on *Trichodesmium*, *Richelia*, generate multiple genome copies during their life cycle (Sargent et al., 2016; White et al., 2018; Karlush et al., 2021). |
| Gene copy number | Filamentous cyanobacteria (includes heterocystous cyanobacteria) possess a genome copy in each cell; it is not known for all diazotrophs the number of *nifH* copies/cell, often assumed to be one. |
| DNA/RNA Extraction efficiency | Not all targets extract uniformly; RNA is prone to degrade |

**L61:** The $N_2$ fixation rates from Tang et al. 2019 are based on an ARA-$^{15}N_2$ fixation comparison including only 8 data points. This is not robust enough to provide a reliable comparison and downplays the robustness of the $^{15}N_2$-based rates dataset collected here. I strongly recommend removing these from the database and derived basin-scale and global calculations. These may be mentioned as discussion and the Tang paper cited, but not included for quantitative purposes.

Response: Thanks for the comment. Here, we referred to a diazotroph dataset compiled by Tang et al. (2019) and Tang and Cassar (2019) with historical measurements reported by other studies in 2012-2018. There were other in-situ $N_2$ fixation rates (15 $^{15}N_2$-based and 85 ARA-based measurements) measured by Cassar/Tang's own group (Tang et al., 2019; Tang et al., 2020); these data were also collected into our database. The derived $N_2$ fixation rates in Tang et al. (2019) were not collected into our database.

We reconsidered ARA-based measurements of $N_2$ fixation rates and agreed with the reviewer. We have decided not to include the ARA-based data in estimating the global $N_2$ fixation rate, while keep them in the database for those who are interested in using them.

**L72:** Removing NCDs is an error in my opinion. NCDs have recurrently been shown to be dominant in the ocean (Farnelid et al. 2011; Delmont et al. 2018, 2021; Riemann, Farnelid, and Steward 2010) and may impact N cycling decisively (Riemann et al. 2022; Turk-Kubo et al. 2022). I strongly recommend that any nifH gene counts of NCDs are added. The previous database included Gamma A and Cluster III. I don't see a solid reason to remove NCDs from the database at this stage, as evidence of their importance increases.

Response: Thanks for the comment. One of the reasons why we did not include NCD data was the existence of a comprehensive NCD dataset compiled by Turk-Kubo et al. (2022). We have now obtained the agreement from Dr. Turk-Kubo to include her NCD dataset in the database (she has agreed to be a coauthor of the revised manuscript). Additional NCD data published in several recent studies have also been added to the revised database. We then accordingly changed the sentence to:

"Additionally we included a compiled NCD dataset (Turk-Kubo et al., 2022) in the database, which contained 7,919 *nifH* gene copy abundances of primarily the most studied phylotype NCD Gamma A (Shao and Luo, 2022; Langlois et al., 2015), also referred to as 24774A11 (Moisander et al., 2012) and UMB (Bird et al., 2005), as well as other phylotypes, and updated that compilation with 469 additional *nifH* gene copy abundances of NCDs published more recently (Turk-Kubo et al., 2021; Sato et al., 2022; Moore et al., 2018; Reeder et al., 2022; Wen et al., 2022; Bonnet et al., 2023). We also collected 468 cell-specific *in situ* $N_2$ fixation rates and added them to version 2."

**Line 82:** Group-specific $N_2$ fixation rates can only be estimated using single-cell approaches. I'm not sure what approach was followed here to derive specific rates, but these can certainly not be estimated with the data collected here. I would rather recommend the authors to collect all *Trichodesmium*, UCYN-B, DDAs and UCYN-A single-cell rates published, which would be

very helpful for the community. See for instance (Foster, Sztejrenszus, and Kuypers 2013; Foster et al. 2011; Benavides et al. 2017; Bonnet et al. 2016; Filella et al. 2022; Krupke et al. 2015; K. Harding et al. 2018; Mills et al. 2020; K. J. Harding et al. 2022; Benavides et al. 2022).

Response: Thanks for the comment and we are sorry for the confusing. The "different groups" here referred to different size groups. In the original database of 2012, $N_2$ fixation rates in samples with size >10 μm were assigned to *Trichodesmium* and those of smaller sizes were assigned to UCYN. In the revised database, we have corrected and reported them as $N_2$ fixation rates of size groups > 10 μm and < 10 μm, respectively. In some studies, $N_2$ fixation rates of *Trichodesmium* and heterocystous cyanobacteria were estimated by multiplying their cell abundance with their cell-specific $N_2$ fixation rates; we also collected these diazotrophic group-specific data into the new version of the database.

We agree with the reviewer that the cell-specific $N_2$ fixation rates are important and valuable. The cell-specific $N_2$ fixation rates suggested by the reviewer, as well as other identified data sources, have been collected into the revised database as a new datasheet.

The paragraph has been revised as:

" $N_2$ fixation rates were measured for whole seawater samples, for different size fractions (> 10 μm and < 10 μm), or specifically for *Trichodesmium* and heterocystous cyanobacteria. When whole-water $N_2$ fixation rates were not reported, total $N_2$ fixation rates were calculated as the sum of the $N_2$ fixation rates of available groups."

"We also collected 468 cell-specific *in situ* $N_2$ fixation rates and added them to version 2."

Tables 2 and 4: Many studies are missing in this table, some include (Benavides et al. 2014, 2021; Saulia et al. 2020; Henke et al. 2018; Bonnet et al. 2018; Gradoville et al. 2017; Moreira-Coello et al. 2017; Wilson et al. 2019). Also, in the table some studies are listed as not including counts of some diazotrophs, which needs correction (e.g. Bombar 2011 and Bonnet 2015, 2019 did have qPCR counts). Please revise all these publications thoroughly and correct accordingly.

Response: We thank the reviewer for identifying missing datasets and parameters. We have checked datasets suggested by the reviewer, and have added those parameters collected by this database. With help from other coauthors, much more missing datasets (more than 40 publications) have been identified and added to the revised database. We are confident that nearly all the previous published data have now been included in the database.

L104: The ARA method is rarely used nowadays

Response: The reviewer was correct. We have revised the texts to:
"The commonly used methods for marine $N_2$ fixation rates include $^{15}N_2$ tracer methods and acetylene reduction assay (Mohr et al., 2010; Montoya et al., 1996; Capone, 1993). However, in the last decade, the community has turned largely to the use of $^{15}N_2$ tracer methods."

L106-107: The ARA to $N_2$ fixation ratio is highly variable (Mulholland et al. 2006; Benavides et al. 2011; Wilson et al. 2012)

Response: Thanks for this comment. We have added the previously reported range of the conversion factor between acetylene reduction and $N_2$ fixation:

"Theoretical conversion factors of 3:1 or 4:1 have been used to convert acetylene reduction rates to $N_2$ fixation rates (Postgate, 1998; Capone, 1993; Wilson et al., 2012), although a wide range of conversion factors from 0.93 to 56 have been reported (e.g., Mague et al., 1974; Graham et al., 1980; Montoya et al., 1996; Capone et al., 2005; Mulholland et al., 2006; Wilson et al., 2012)."

L110: Many other factors affect this difference, including acetylene gas impurity, Bunsen dissolution coefficient, etc.

L112: This is not true. The $^{15}N_2$ method is much more sensitive, does not require biomass preconcentration (biomass is concentrated during filtration, after the incubation), and requires longer incubations for enough tracer to be detectable in biomass. ARA is usually done in 3-4 h incubations and requires biomass pre-concentration to reach detectable signal (Staal et al. 2007; Benavides et al. 2011).

Response: We thank the reviewer for the above two related comments regarding comparing $^{15}N_2$ assimilation and the acetylene reduction assay. We have incorporated the reviewer's comments and corrections and modified the texts as follows:

"When using the $^{15}N_2$ tracer method, samples are incubated in seawater with $^{15}N_2$; the $^{15}N/^{14}N$ ratio of particulate nitrogen is measured at the beginning and at end of the incubation to calculate the $N_2$ fixation rate (Capone and Montoya, 2001). Most measurements using the $^{15}N_2$ tracer method only counted the fixed N in particulate forms and ignored the N that was fixed but then excreted by diazotrophs in form of dissolved organic N (DON) during incubation, which could theoretically be counted by the acetylene reduction assays (Mulholland, 2007). In some studies using the $^{15}N_2$ tracer method, this missing N was counted by also measuring the $^{15}N$ enrichment in DON (Berthelot et al., 2017; Benavides et al., 2013a; Berthelot et al., 2015; Benavides et al., 2013b).

Compared to the $^{15}N_2$ tracer method, the acetylene reduction assay needs a shorter incubation time. However, in addition to the uncertainty in converting ethylene production to $N_2$ fixation, the purity of acetylene gas, trace ethylene contamination, and the Bunsen gas solubility coefficient of produced ethylene can also affect the accuracy of estimated $N_2$ fixation rates (Hyman and Arp, 1987; Breitbarth et al., 2004; Kitajima et al., 2009). Acetylene used in the assay can even impact the metabolic activities of diazotrophs (Giller, 1987; Hardy et al., 1973; Flett et al., 1976; Staal et al., 2001). Moreover, the acetylene reduction assay needs to pre-concentrate cells for signal detection when diazotrophic biomass is low, which may lead to underestimated $N_2$ fixation rates by perturbing cells during concentration and filtration (e.g., Capone et al., 2005; Barthel et al., 1989; Staal et al., 2007). In recent years, the acetylene reduction assay has undergone significant advancements. The sensitivity of ethylene detection has been improved by utilizing a reduced gas analyzer (Wilson et al., 2012) and by using highly purified acetylene gas to minimize the ethylene background (Kitajima et al., 2009). However, the preparation of high-purity acetylene with low level of ethylene contamination remains a challenge. More recently, a new method named Flow-through incubation Acetylene Reduction Assays by Cavity ring-down laser Absorption Spectroscopy (FARACAS) has been introduced for high-frequency measurements of aquatic $N_2$ fixation (Cassar et al., 2018). This method involves continuous flow-through incubations and spectral monitoring of the acetylene reduction to ethylene. By employing short-duration flow-through incubations without cell preconcentration, potential artifacts are minimized. This approach also allows for near real-time

estimates, enabling adaptive sampling strategies."

L120: Wannicke et al. say the opposite of Mohr and Grosskopf.

L123: What White et al. say is that the bubble release method is the most reliable and recommended by the diazotroph research community, with the elimination of rate underestimation benefits overcoming the very unlikely burdens of contamination. This should be corrected in L274-275 as well.

Response: Thanks for pointing out these two mistakes. We have carefully revised whole section:

"The original $^{15}N_2$ tracer method involved addition of a known volume of $^{15}N_2$-labelled bubbles to the incubation bottle (named *original $^{15}N_2$ bubble method* hereafter). However, this method was later found to underestimate rates because $N_2$ gas solubility is low and tracer additions take a long time to equilibrate (Mohr et al., 2010; Großkopf et al., 2012; Jayakumar et al., 2017). To address this issue, the *$^{15}N_2$ dissolution method* has been employed, which involves pre-preparing $^{15}N_2$-enriched seawater to maintain a constant $^{15}N_2$ atom% enrichment throughout the incubation (Mohr et al., 2010), similar to the method described in Glibert and Bronk (1994). However, the $^{15}N_2$ dissolution method does not always yield higher $N_2$ fixation rates than the original $^{15}N_2$ bubble method (Table S4 in Großkopf et al., 2012; Saulia et al., 2020); it is still not conclusive what control the magnitude of the underestimation (if it exists) by the original $^{15}N_2$ bubble method. Compared to the original $^{15}N_2$ bubble method, the $^{15}N_2$ dissolution method is more susceptible to the introduction of contaminants (e.g., metals) during the preparation of the $^{15}N_2$ inoculum due to its more complex process, which can alter the diazotrophic activities and abundance, thereby impacting the accuracy of $N_2$ fixation measurements. (Dabundo et al., 2014; Klawonn et al., 2015). For example, Needoba et al. (2007) reported that a low but detectable amount of $Fe^{3+}$ contamination can be measured when protecting the needle of the gas-tight syringe with a commercially available tubing. Additionally, pH and other chemical properties of the inoculum may be altered during its preparation, further affecting the measurements of $N_2$ fixation. Despite these limitations, the $^{15}N_2$ dissolution method remains the predominant assay for measuring $N_2$ fixation rate due to its ability to satisfy the fundamental assumption of constant $^{15}N_2$ atom% enrichment over the incubation period.

More recently, a modified $^{15}N_2$ bubble method, known as the *$^{15}N_2$ bubble release method*, has been proposed as an alternative to the $^{15}N_2$ dissolution method (Klawonn et al., 2015; Chang et al., 2019; Selden et al., 2019). This method involves adding $^{15}N_2$ gas to the incubation bottles and mixing for a brief period (~15 min) to facilitate $^{15}N_2$ equilibration, then removing the gas bubble. Compared to the original $^{15}N_2$ bubble method, the $^{15}N_2$ bubble release method ensures a uniform $^{15}N_2$ atom% enrichment throughout the incubation. Moreover, it causes less interference with the incubation matrix than the $^{15}N_2$ dissolution method. However, the slow and gentle rocking of incubation bottles required to stimulate gas dissolution has been suggested

to negatively affect diazotrophs, although no robust studies have yet been performed to assess this criticism (Wannicke et al., 2018; White et al., 2020). Moreover, the $^{15}N_2$ bubble release method requires a handling step and additional costs for preparing tracers may not be allowed (White et al., 2020). Ultimately White et al. (2020) "advise employing either the dissolution or bubble release method, whichever is best suited to the specific research objectives and logistical constraints" with additional recommendations on the need for determination of detection limits for all rate measurements."

The first sentence of 4.1has also been revised as:

"To date, the discrepancy in $N_2$ fixation rates estimated using different $^{15}N_2$ tracer methods remains unclear."

L150: There are 4 UCYN-A sublineages (Farnelid et al. 2016).

Response: We have corrected the text as follows:

"Four sublineages of UCYN-A, including UCYN-A1, UCYN-A2, UCYN-A3, and UCYN-A4, have been identified (Thompson et al., 2014; Farnelid et al., 2016)."

L328: UCYN-A has been found in symbiosis with other eukaryotic algae (Zehr et al. 2016)

Response: The text has been revised as:

"The conversion factor for UCYN-A is also updated because it has been found to live symbiotically with haptophyte *Braarudosphaera bigelowii* and relatives (Thompson et al., 2012; Hagino et al., 2013). "

L370: The first version of the database included all the authors that had contributed to its construction with their seagoing expeditions, laboratory analyses and publications. I humbly find it sad and somewhat unfair that this is not the case in this update.

Response: We highly value the reviewer's comment and agree proper coauthorship credit is important for all contributors. Initially, our plan to publish an updated global marine diazotrophic database was too simplistic and lacked careful thinking. Since the first global marine diazotrophic database was published in 2012, our group has continuously updated the database with newly published data. In recent years, we have received numerous requests for an updated version of the database, which prompted us to consider publishing it for wider usage.

We have extended an invitation to all lead authors to join us as coauthors of the manuscript. Please refer the texts at the beginning of this file for our detailed response to this suggestion.

**Reviewer #2:**
This manuscript by Shao and Xu et al. describes an updated version 2 of the global oceanic diazotroph database. It build upon the previous version by adding additional measurements of

marine diazotrophic abundance, $N_2$ fixation rates, microscopic and qPCR-based diazotropic abundance. The spatial coverage significantly improved most notably in the Indian Ocean. The newly revised estimate for global $N_2$ fixation rate is significantly higher (+123 Tg N $yr^{-1}$, almost doubled) when calculating using a standard arithmetic mean, although surprisingly the geometric mean did not significantly change. A brief analysis and discussion of the $^{15}N_2$ bubble vs. dissolution indicated a potential general underestimation from the bubble method particularly at high rates, however noting the comparison of samples were from different times so it is not a formal error analysis (which the authors acknowledge). The database is available to download from the provided link in the abstract.

Overall, I find this to be an important update to the database mainly due to the significant increase in included measurements and spatial coverage. The database is transparent and mostly well described. The analysis and first preliminary quantification of the $^{15}N_2$ bubble vs. dissolution is also an important contribution. Perhaps some additional details/analysis could be provided (see comments below), but additional analyses can also be performed independently by users who download the data for their specific interest. There is one important aspect that needs additional clarification in my view before I would endorse this manuscript for publication (global $N_2$ fixation rate calculation, see below).

-Christopher Somes
GEOMAR Helmholtz Centre for Ocean Research Kiel

Response: We thank Dr. Somes for his positive and constructive comments, which have helped us improve the quality of this paper substantially. Please see our responses below.

**Major Comment: Global N2 fixation calculation description**

Since this paper will likely often be cited for revising the global $N_2$ fixation rate significantly upwards, the description of this calculation should be more transparent and comprehensive:

line 266 (Table 5 caption): "Data are first binned to 3x3 grids…"

This needs to be better described. For example, was there any type of interpolation method used or simple averaging of all measurements in each bin? It would be interesting to know what percentage of bins in each ocean basin has data coverage. How do you define the Southern Ocean region and is that area removed from the other southern regions?

How was the vertical coordinate handled? Is it evenly spaced or according to the depths ranges in Figure 7?

It is not clear to me how the "Areal sum" calculation was made based on the "Mean $N_2$ fixation rate" (Table 5). Does the "Mean $N_2$ fixation" rate include all measurements or only the "Depth-integrated $N_2$ rates", which requires 3 measurements in the vertical? If the vertical coordinate is uneven, do measurements that get binned into a larger volume in larger deeper layers have more weight on the depth-integrated rate than shallower layers?

When calculating the "Areal Sum", do you assume that the "Mean N2 fixation rate" extrapolates across the entire region or do you only consider the area of the bins that have data coverage? For example, the Indian Ocean has about 36% of the bins compared to the South Pacific. Therefore I was expecting a much larger decrease when calculating the Areal Sum

relative to the Mean $N_2$ fixation rate for the Indian Ocean compared to the South Pacific. However this relative decrease is quite subtle in Table 5 between these regions. I acknowledge there is no truly perfect way to estimate a global ocean $N_2$ fixation rate with the current coverage, but all of the assumptions and details that go into the calculation should be specifically stated and described.

Response: Here we respond Dr. Somes's general comments regarding the description of calculating the global marine $N_2$ fixation rate.

We followed the procedure used in the previous database paper (Luo et al., 2012) to estimate the global marine $N_2$ fixation rate. However, as reminded by the reviewer, we should describe the method in this paper, which has been added in section 2.4 in the revised manuscript.

Here are some quick answers to the reviewer's questions:

The data used in the estimation is the depth-integrated $N_2$ fixation rates integrated from surface to the depth of the deepest data (up to 200 m; see section 2.1). The measurements in each vertical profile were linearly interpolated, which was not clearly described in the original manuscript. We have revised the sentence (in Section 2.1) to: " The measurements within a profile were first interpolated linearly with depth, with the shallowest datum representing the level between the sea surface and the depth of that datum. The profile was then integrated from the sea surface to the deepest recorded measurement. Most vertical profiles of $N_2$ fixation rates were measured within the euphotic zone, with a few studies extending measurements to several hundred meters or deeper. In these cases, we only integrated to the deepest data point above 200 m, taking into account the scarcity of aphotic $N_2$ fixation measurements in the global ocean and their controversial contribution to the global budget (Benavides et al., 2018). As a result, it was possible that certain measurements below the euphotic zone but above 200 m were included in the integration. However, these measurements would typically have minimal impact on the depth-integrated $N_2$ fixation rates due to their low rates and limited vertical extent in this range."

The arithmetic and geome mean of the data in each bin was calculated first, and then these means in each basin were averaged further.

The Southern Ocean was defined as the area south of 45°S and was excluded from other basins when calculating the global rates. Additionally, due to very limited data coverage, the $N_2$ fixation rates of the Southern and Arctic Oceans have been excluded from the estimation of global marine $N_2$ fixation.

The percentage of bins with data coverage in each ocean basin have been added in the revised table.

When calculating the areal sum, we extrapolated the mean $N_2$ fixation rate of each basin across the entire basin, i.e., the mean $N_2$ fixation rate was multiplied by the area of each basin. We have listed the areas of every ocean basin in the table.

The description of the methods in calculating global marine $N_2$ fixation rate was added to Section 2.4:

"The estimation of the global marine $N_2$ fixation rate involved four steps. First, we calculated

the arithmetic or geometric means of depth-integrated $N_2$ fixation rates within each 3° latitude × 3° longitude bin. Second, these mean values were further averaged using either arithmetic or geometric methods to determine the mean $N_2$ fixation rates for different ocean basins, which included the North Atlantic, South Atlantic, North Pacific, South Pacific, Indian, Arctic, Southern Oceans, and the Mediterranean Sea. Third, we multiplied the arithmetic or geometric mean of each basin by its respective area to estimate the total $N_2$ fixation rate for that specific basin, except when there was insufficient spatial coverage available. Finally, we obtained the global marine $N_2$ fixation rate by summing up the individual rates calculated for each basin, with the errors associated with basin rates propagated properly (Glover et al., 2011).

In the first two steps, the geometric means were derived from positive $N_2$ fixation rates ($NF_+$): if $\mu$ and $SE$ represented the mean and standard error of $\ln(NF_+)$, respectively, the geometric mean was $e^\mu$. The confidence interval for the geometric mean, based on the standard error, ranged between $e^\mu / e^{SE}$ and $e^\mu \cdot e^{SE}$ (Thomas, 1979). To address the issue of not including zero-value $N_2$ fixation rates, we adjusted the geometric means by multiplying them with the percentage of zero-value data within each 3° × 3° bin (in the first step) or within each basin (in the second step)."

The authors do not give much context on interpreting the geometric vs. arithmetic mean despite that it is mentioned multiple times throughout the manuscript and gives a significantly different result. From what I understand, geometric mean is less sensitive to the high-end rates compared to arithmetic mean. Does this mean that most of the increase in the arithmetic mean is driven by newly included high-end rates? It would be valuable to know how much of the large increase in the arithmetic areal sum is driven by additional spatial coverage versus generally higher rate values. I would suggest to include a histogram of the previous version in one of the supplementary figures for comparison. If newly included rate values tend to be significantly higher, it would be interesting to know how much of that may be attributable to growing numbers of the dissolution method compared to bubble method (i.e. based on Figure 10).

Response: Dr. Somes was correct in interpreting geometric versus arithmetic means. As our $N_2$ fixation data were approximately log-normally distributed, their geometric mean is near the most frequently observed rate (i.e., the peaks of the distribution of the log-transformed $N_2$ fixation rates). Meanwhile, high $N_2$ fixation rates do occur and should be included in estimating global $N_2$ fixation. Hence, the arithmetic means should be used in estimating global $N_2$ fixation if sufficient data have been sampled. However, if the number of samples is small, some occasionally observed high $N_2$ fixation rates can greatly elevate the estimated global rate while we cannot know if these high $N_2$ fixation rates are typical. This was the reason that we presented both the geometric and arithmetic means of $N_2$ fixation rate.

In the revised manuscript, the increase in the arithmetic-mean-based estimation of global marine $N_2$ fixation, compared to that in Luo et al. (2012), was caused mostly by (1) the much higher estimation for the South Pacific Ocean and North Atlantic Ocean and (2) the estimation for the Indian Ocean for which the estimation of $N_2$ fixation was not made in Luo et al. (2012).

We also conducted additional analyses of data distributions to explore reasons why the

arithmetic means can be much higher using version 2 compared to using version 1 of the database, and found that version 2 considerably extends both the left and right tails of the data distribution. We concluded the previous assessments of the global marine $N_2$ fixation rate were likely underestimated due to the absence of these new measurements.

The text has been revised and new figures have been produced:

"The substantial increase was mostly driven by notable changes in the South Pacific, North Atlantic, and Indian Oceans. In the South Pacific Ocean, numerous high $N_2$ fixation rates were observed in the western subtropical region over the past decade (**Fig. 12**), resulting in a substantial increase of 68±23 Tg N $yr^{-1}$ in the estimated $N_2$ fixation rate for this basin (**Table 8**). It is worth noting that these newly recorded measurements in the western subtropics of the South Pacific Ocean might even be underestimated since most of them were obtained using the original $^{15}N_2$ bubble method. In the North Atlantic Ocean, the estimated $N_2$ fixation rate also experienced an increase of 30±9 Tg N $yr^{-1}$ for (**Table 8**), without any discernible pattern regarding the locations of the new high $N_2$ fixation measurements (**Fig. 13**). Furthermore, in the Indian Ocean, the improved data coverage in version 2 (**Fig. 8a**) supported the estimation of an $N_2$ fixation rate of 35±14 Tg N $yr^{-1}$ for this basin (**Table 8**), which was not possible to calculate using version 1 due to insufficient data availability.

However, when estimating the global marine $N_2$ fixation rate using geometric means, both version 1 and version 2 yielded similar rates of approximately 50 Tg N $yr^{-1}$ (**Table 9**). The $N_2$ fixation rates in each basin tended to follow a log-normal distribution (**Fig. 14; Table S2**), with the geometric mean aligning near the peak of the distribution. In the South Pacific Ocean, as discussed earlier, version 2 included a substantial number of newly observed high $N_2$ fixation rates, but it also incorporated a significant number of rates that were much lower than those in version 1 (**Fig. 14c**). This could be partially attributed to enhanced detection limits in measurements. Consequently, while version 2 yielded a much higher arithmetic mean $N_2$ fixation rate compared to version 1 for the South Pacific Ocean (**Table 8**), their geometric means remained quite similar (**Table 9**). In the North Pacific Ocean, for the same reasons, the arithmetic mean $N_2$ fixation rates obtained from both versions were very close, while the geometric mean from version 1 could be even higher than that from version 2 (**Tables 8 & 9; Fig. 14a**). These analyses reveal that, despite the similarity in geometric means of $N_2$ fixation rates obtained from both versions of the database, the higher arithmetic means in version 2 were not coincidental. Instead, they were a direct outcome of the improved measurement methods and the expanded spatial and temporal coverage of marine $N_2$ fixation over the past decade. Consequently, previous assessments of the global marine $N_2$ fixation rate were likely

underestimated due to the absence of these new measurements."

[Figure]

**Figure 12.** Depth-integrated N₂ fixation rates in the South Pacific Ocean (μmol N m⁻² d⁻¹). The shown data are arithmetic mean rates in 3° latitude ×3° longitude bins. Empty diamonds and filled circles denote the existing data in the version 1 of the database and the new data added to version 2, respectively.

[Figure]

**Figure 13.** Depth-integrated N₂ fixation rates in the North Atlantic Ocean (μmol N m⁻² d⁻¹). The shown data are arithmetic mean rates in 3° latitude ×3° longitude bins. Empty diamonds and filled circles denote the existing data in the version 1 of the database and the new data added to version 2, respectively.

[Figure]

**Figure 14.** Comparison of the distribution of log-transformed $N_2$ fixation rates between the two versions of the database. Note that the zero-value data are not included because of the log-transformation. The comparison is performed for data in (**a**) North Pacific, (**b**) North Atlantic, (**c**) South Pacific, and (**d**) South Atlantic Oceans.

**Minor Comments:**

line 84 and data file: Metadata
In the data file, the meta data are titled "Surface …", yet they are associated with a specific depth, so are they really surface? I am used to seeing chlorophyll expressed by volume not area.

Response: Thank you for pointing out the mistakes. In the volumetric spreadsheets, the meta data were measured at the same depths as the diazotrophic data, and the word "surface" has been deleted from their names. Similarly, the chlorophyll concentration in the volumetric datasheets should be in unit of mg m$^{-3}$, which has been corrected in the revised database.

In the depth-integrated datasheet, considering the large vertical variations of environmental parameters and chlorophyll, we collected their near-surface values. We have corrected their names to "Near-surface xx".

lines 127-129: daily vs. daytime vs. nighttime normalization
I am still a little confused about the time normalization with this brief description. If the incubation is only performed during the day, you convert hours to day by 12 hr/day which assumes no rates at night? I see that incubation hours vary a lot and in some cases not a multiple of 12 hours or 1 day. Perhaps you can describe how individual studies typically convert to a daily rate depending on the incubation period. Would it make more sense to multiply by the daytime of each location during the time of sampling instead of assuming 12 hours?

Response: We have divided the $N_2$ fixation rate data into two spreadsheets based on the incubation period (24 hrs or < 24 hrs), only the $N_2$ fixation rates with incubation period of 24 hours were used in this estimation or method comparision. The method has now been descried with more details:

"The majority of $N_2$ fixation rates (9,405) were measured with incubation periods of 24 hours and were reported as daily rates. In contrast, 2,416 samples were incubated for less than 24 hours and hourly $N_2$ fixation rates were reported. Diel cycles of $N_2$ fixation vary among samples and/or diazotrophic groups, and substantial errors may be introduced when extrapolating $N_2$ fixation rates incubated for less than 24 hours to daily rates (White et al., 2020). Therefore, the $N_2$ fixation rates measured with incubation periods of less than 24 hours were collected into separated datasheets in our database and were not used in further analyses within this study. Please note that the incubation periods of whole diurnal cycles (e.g., 24, 48, or 72 hours) were used in Konno et al. (2010). The samples in Yogev et al. (2011) were incubated between 24 to 30 hours. The reported daily $N_2$ fixation rates by these two studies were also included in the 24-hour datasheets and were used in our estimation of the global marine $N_2$ fixation rate (see below)."

Table 5.: "n" is missing in Indian Ocean

Response: Corrected.

Figure 7:
Why do you choose geometric mean over the more commonly used arithmetic mean in this figure? Does it look significantly different if you use arithmetic means?

Response: The general spatial pattern of $N_2$ fixation was similar when using either geometric or arithmetic means, except for some high arithmetic means. In order to demonstrate these high values in the global ocean, we then have changed to present arithmetic means of $N_2$ fixation in the revised manuscript (Fig. 7 attached below).

[Figure]

Figure 8. N$_2$ fixation rates in the version 2 of the database. The panels show (a) depth-integrated data and volumetric data in (b) 0–5 m, (c) 5–25 m, (d) 25–100 m, (e) 100-200 m, and (f) below 200 m. For a clear demonstration, arithmetic mean N$_2$ fixation rates in 3° latitude × 3° longitude bins are shown. Zero-value data are denoted as black empty circles. Only rates measured with incubation periods of 24 hours are included.

I would be interested to see a euphotic vs. aphotic depth-integrated rate. I am curious how much the generally low to moderate rates occurring below 100 meters contribute to the total depth-integrated rate since they can occupy more volume. Perhaps adding a < 100m and >100m panel would be useful? At what depths are the deepest N2 fixation measurements?

Response: We have generated an averaged vertical profile of N$_2$ fixation rates from sea surface to the deepest (4000 m; Hallstrøm et al., 2022) N$_2$ fixation measured (**Figure R1** attached below). Using the average vertical profiles of this figure, the total N$_2$ fixation below 200 m would be 2.5 times magnitude of that above 200 m.

However, the contribution of aphotic to the global budget has been discussed elsewhere (Benavides et al., 2018). This paper recoganized that the scarce N$_2$ fixation measurements in the dark ocean prevented a reliable estimate, which was still true after 5 years as shown in the figure below. We then decided not to include the estimate of aphotic N$_2$ fixation in our paper, considering it could be somehow unreliable and beyond scope of this paper.

[Figure]

**Figure R1.** Vertical profile of $N_2$ fixation rates in the global ocean. Blue circles represent the reported $N_2$ fixation rates, and the red circles and error bars are the means and standard errors in depth intervals marked in the y-axis. The x-axis is in a log scale to better show the distribution of low $N_2$ fixation rates.

Section 4.1/Figure 10:
As mentioned above, I think is a useful first investigation into methodological uncertainties on N2 fixation rates. Is there enough data coverage to do a similar analysis for acetylene reduction?

Response: We thank Dr. Somes to recognize the value of our analyses. We compared the $N_2$ fixation rates measured using the $^{15}N_2$ tracer methods and from the acetylene reduction (ARA) method. However, there were too limited pairs of data available (n=16 and 6 for ARA vs. $^{15}N_2$ dissolution method and ARA vs. the original method, respectively) to be included in the manuscript (see **Figure R2** attached blow for your reference).

[Figure]

**Figure R2.** Comparison of measured $N_2$ fixation rates using the $^{15}N_2$ tracer methods and acetylene reduction (ARA) assays. (a) The ARA versus the $^{15}N_2$ dissolution; and (b) the ARA versus the original $^{15}N_2$ bubble method. The pink dots are measurements. The fitted results using the generalized additive model (GAM) and confidence intervals are represented by the red solid line and the dashed black lines, respectively. The blue lines is the 1:1 ratio of the measurements using the compared methods.

Please be also noted that there were mistakes when pairing $^{15}N_2$ dissolution and $^{15}N_2$ bubbling measurements in the original manuscript, which have been corrected in the revised manuscript (new **Fig. 13** attached below). The texts have been revised as follows:

[revised manuscript text omitted]